# AGENT INSTRUCTS LARGE LANGUAGE MODELS TO BE GENERAL ZERO-SHOT REASONERS

## ABSTRACT

We introduce a method to improve the zero-shot reasoning abilities of large language models on general language understanding tasks. Specifically, we build an autonomous agent to instruct the reasoning process of large language models. We show this approach further unleashes the zero-shot reasoning abilities of large language models to more tasks. We study the performance of our method on a wide set of datasets spanning generation, classification, and reasoning. We show that our method generalizes to most tasks and obtains state-of-the-art zero-shot performance on 20 of the 29 datasets that we evaluate. For instance, our method boosts the performance of state-of-the-art large language models by a large margin, including Vicuna-13b (13.3%), Llama-2-70b-chat (23.2%), and GPT-3.5 Turbo (17.0%). Compared to zero-shot chain of thought, our improvement in reasoning is striking, with an average increase of 10.5%. With our method, Llama-2-70b-chat outperforms zero-shot GPT-3.5 Turbo by 10.2%.

## 1 INTRODUCTION

Large language models (LLMs) (Brown et al., 2020; Wang & Komatsuzaki, 2021; Zhang et al., 2022a; Smith et al., 2022; Chowdhery et al., 2022; Hoffmann et al., 2022; Scao et al., 2023; OpenAI, 2023; Anil et al., 2023; Touvron et al., 2023a;b; Penedo et al., 2023) have significantly advanced the state-of-the-art on a wide range of language understanding tasks. These models have led to widespread deployment and adoption in applications (Araci, 2019; Huang et al., 2020; Bolton et al., 2022; Wu et al., 2023; Driess et al., 2023; Huang et al., 2023). In particular, the emerging capabilities of LLMs such as complex reasoning (Wei et al., 2022b; Wang et al., 2023c; Kıcıman et al., 2023) have made them the subject of research in recent years. Among these, zero-shot reasoning (Kojima et al., 2022; Wan et al., 2023) has drawn substantial public interest and achieved promising results in specific task domains. However, the reasoning ability of LLMs on general tasks remains unclear.

In this paper, we improve the zero-shot reasoning abilities of LLMs on general language understanding tasks. To solve a task, we build an agent to instruct the reasoning process of LLMs for the task (Figure 1). More specifically, our autonomous agent generates task-specific instructions to better align the chain of thought reasoning process of LLMs with each task. We refer to this approach as zero-shot agent instructed reasoning (AgentInstruct). The basic idea of our approach is motivated by two lines of work. First, language agents (Yao et al., 2023b; Shinn et al., 2023; Park et al., 2023; Wang et al., 2023a; Xi et al., 2023) have been developed to automatically complete a task. Instead of completing the task, our agent produces instructions on how to complete the task. We enable this by adapting an existing agent to access a wide range of task-relevant knowledge on the web, given basic task information such as the name of the dataset and several input examples. As a result, the agent synthesizes high-quality step-by-step instructions for tasks verified by the web resources. We follow the recent design of agents where an LLM is used to produce the plans for this process. Second, zero-shot chain of thought (CoT) reasoning of LLMs has obtained promising results on tasks such as arithmetic reasoning (Kojima et al., 2022; Wang et al., 2023b). Standard in-context zero-shot learning prompts an LLM to directly output predictions without task examples. In contrast, CoT decomposes a task into intermediate steps and solving each will lead to the final output. We further align the CoT reasoning steps with a particular task by prompting with the task-specific agent instructions. The design of zero-shot AgentInstruct is important: We generalize the zero-shot reasoning abilities of LLMs to more tasks with the combination of task-specific instructions from a language agent and task-specific reasoning of LLMs.

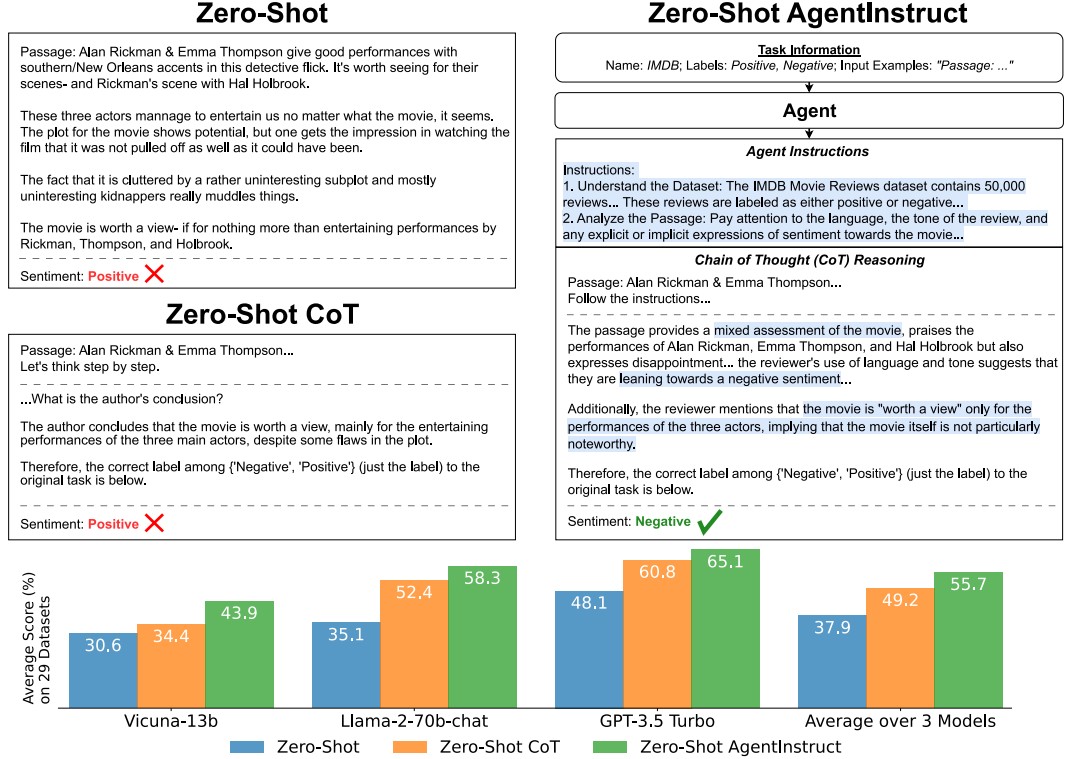

Figure 1: Summary of our approach and results. Top: Our zero-shot AgentInstruct generalizes the zero-shot reasoning abilities of large language models to a wide set of language understanding tasks including generation, classification, and reasoning. Our agent produces task-specific instructions to instruct the reasoning process of large language models to solve a task. Both the agent instructions and task-specific reasoning process are highlighted. Bottom: Performance of zero-shot AgentInstruct compared with standard zero-shot and zero-shot chain of thought (CoT). Zero-shot AgentInstruct improves the performance of three large language models substantially on the 29 datasets we evaluate.

We empirically evaluate the zero-shot reasoning abilities of LLMs on a wide set of language understanding tasks across 29 datasets (including 53 subsets), spanning generation, classification, and reasoning. Zero-shot AgentInstruct obtains state-of-the-art performance on 20 datasets. We conduct our evaluation on three state-of-the-art LLMs, namely, Vicuna (Chiang et al., 2023), Llama-2-chat (Touvron et al., 2023b), and GPT-3.5 Turbo (OpenAI, 2022). We show that zero-shot AgentInstruct boosts the performance of these models by 17.8% on average. When compared to zero-shot CoT, the overall performance improvement is significant (6.5%), and in particular, the improvement in reasoning is substantial with an average increase of 10.5%, leading to the best performance on 10 out of 12 reasoning tasks. Notably, Llama-2-70b-chat with zero-shot AgentInstruct outperforms standard zero-shot GPT-3.5 Turbo by an average of 10.2%. We hope the results help foster future research on further unlocking the zero-shot reasoning abilities of large foundation models and exploring the broader usage of agents.

## 2 APPROACH

We present zero-shot AgentInstruct in this section. To solve a task, zero-shot AgentInstruct employs an agent to instruct an LLM to reason toward the final prediction. Intuitively, humans often rely on specific instructions to more effectively guide their thought process as they work towards a solution to a problem. For instance, to understand the sentiment in the movie reviews, instructions such as "*1. Understand the Dataset: ... Movie Reviews dataset ... 2. Analyze the Passage: Pay attention to ... the tone of the review ...*" help humans decompose the problem into task-specific reasoning steps and solve each to deliver the final answer (Figure 1). Zero-shot AgentInstruct follows this intuition.

**Agent Instructions** Instead of handwriting task-specific instructions, we build an agent to automate the process. The intuition is that an agent is able to synthesize high-quality instructions with access to a wide range of existing task knowledge on the web. We design our agent based on ReAct (Yao et al., 2023b). This is motivated by the recent developments of language agents for task solving. Our agent highlights two features (Figure 2): (i) Instruction generation. Our agent follows ReAct which uses an LLM to propose a series of thoughts. The agent then receives observations and takes actions following the thoughts. Different from ReAct which aims to directly solve the task, our agent outputs step-by-step instructions on how to solve the task. The major advantage of this is that we only need to generate instructions once per dataset instead of running the agent on all dataset instances. We use GPT-4 (OpenAI, 2023) as the de-

Figure 2: An example of our agent producing instructions for a classification dataset IMDB. The task-specific instructions are highlighted.

fault agent. Once the action is `finish`, the corresponding output is our task-specific instructions. (ii) Action space. We constrain our action space to contain two types of actions that support the instruction generation: (a) `ask_about_dataset[string]`, which returns the top relevant web pages containing information about the dataset. To do this, we construct and utilize a question answering API as a tool. This API is meant to answer questions about the document by interfacing with a vector database storing the web pages to provide an answer. (b) `finish[instructions]`, which finishes the instruction generation with the task-specific instructions. As shown in Figure 2, to produce the instructions, our agent takes basic dataset information such as the name of the dataset (e.g., IMDB), a few input-only examples (examples without ground truth labels), and the set of output labels for the dataset (if applicable, else the type of dataset like generation) as its input. Using the task knowledge from the web, our agent forms observations (e.g., "*Observation 1: ... labeled as either positive or negative ...*") and thoughts (e.g., "*Thought 2: ... creating instructions ...*") which trigger the agent to perform actions, such as the `finish` action to output the task-specific instructions.

**Chain of Thought Reasoning** Chain of thought (CoT) (Wei et al., 2022b; Kojima et al., 2022) prompts LLMs to break down the task into intermediate reasoning steps that lead to the final answer. Unlike zero-shot CoT which uses a fixed prompt "Let's think step by step", we prepend our task-specific agent instructions to the input to prompt the LLMs to optimize their reasoning processes for the task. LLMs will then follow our task-specific instructions to decompose the task into a chain of more specific intermediate steps to solve the task. As shown in Figure 1, the agent instructions "... *Pay attention to ... explicit or implicit expressions of sentiment towards the movie ...*" are the key to producing the critical reasoning path "... *the movie is worth a view only for the performances of the three actors ...*", which leads to the correct prediction where standard zero-shot and zero-shot CoT fail. We follow zero-shot CoT, which consists of a reasoning extraction prompt to produce the intermediate reasoning steps, and an answer extraction prompt to collect the answers. For simplicity of implementation, we replace zero-shot CoT's fixed prompt with zero-shot AgentInstruct's task-specific instructions in the reasoning extraction prompt.

AgentInstruct is zero-shot as there are no task-specific examples involved in the pipeline. It enjoys several unique properties: (i) Zero-shot AgentInstruct is a new way to improve the zero-shot reasoning of LLMs. Zero-shot AgentInstruct decouples the language agent and reasoning process of LLMs, which helps zero-shot AgentInstruct generalize to more tasks. Additionally, the agent instructions provide more task-specific controls to the reasoning paths of LLMs, which benefits human alignment and improves the safety of LLMs. (ii) Our agent instructions are customized for different tasks and verified by existing task knowledge. For each task, different LLMs use the same set of instructions, and we find the instructions transfer well among these models. This is important in practice as the agent LLMs are often more powerful and costly than the reasoning LLMs, so our approach is a cost-effective alternative to using agents directly. (iii) By providing task-specific instructions, chain of thought reasoning abilities are further generalized to more tasks beyond reasoning tasks. We show general language understanding tasks such as generation and classification also benefit from chain of thought reasoning. To better embrace the emerging reasoning capabilities of LLMs, our task-specific instructions align the reasoning process with a particular task better than general or fixed instructions.

## 3 EXPERIMENT

We show that zero-shot AgentInstruct successfully improves the zero-shot reasoning abilities of LLMs, namely, Vicuna (Chiang et al., 2023), Llama-2-chat (Touvron et al., 2023b), and GPT-3.5 Turbo (OpenAI, 2022), by a large margin on average. We evaluate zero-shot AgentInstruct on an exhaustive selection of 29 benchmarking datasets containing 53 subsets. As shown in Figure 3, each dataset is either a generation or classification task and a portion of the datasets in each category are also reasoning tasks. The datasets consist of all HELM core scenarios from Liang et al. (2023), as well as the reasoning datasets from Kojima et al. (2022). The details of the experimental setup, including datasets and models, are described in Appendix A.

### 3.1 MAIN RESULTS

Results are shown in Figure 5. We compare zero-shot AgentInstruct to standard zero-shot and zero-shot CoT. We focus our analysis on three models: Vicuna-13b, Llama-2-70b-chat, and GPT-3.5 Turbo.

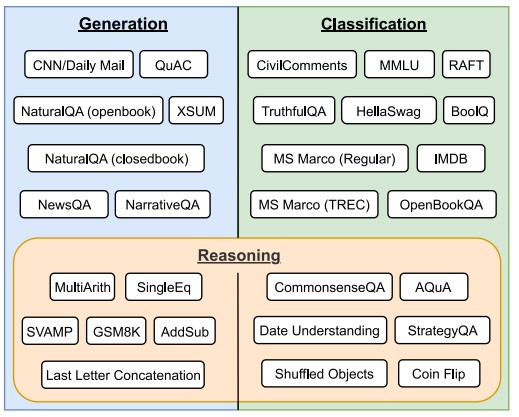

Figure 3: Datasets for generation (blue), classification (green), and reasoning (orange). Reasoning contains generation and classification tasks.

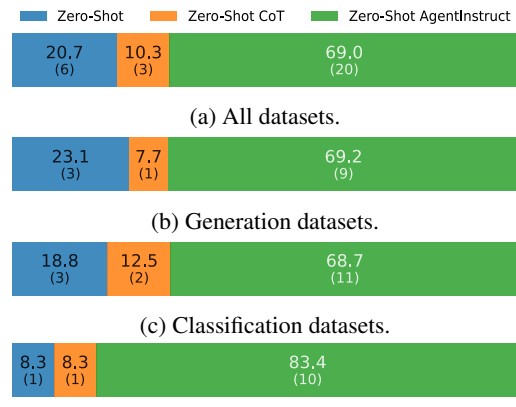

(a) All datasets.

(b) Generation datasets.

(c) Classification datasets.

(d) Reasoning datasets.

Figure 4: Winning rate (%) between zero-shot, zero-shot CoT, and zero-shot AgentInstruct based on the average results over three models.

We first compare zero-shot AgentInstruct to standard zero-shot prompting (Figure 1). The zero-shot prompt design follows Liang et al. (2023). On each model, zero-shot AgentInstruct wins on the majority of datasets, with no less than a 13.0% increase on average. Average performance versus the zero-shot setting is best on Llama-2-70b-chat with a 23.2% improvement. Figure 5a and Figure 5b show the results for generation and classification tasks respectively. On average, with zero-shot AgentInstruct, the three models beat the zero-shot setup by 23.1% for generation and 13.5% for classification. We hypothesize that generation datasets generally require more specific instructions than classification datasets, as the model does not know the best format of the generation output unless it has sufficient task information. This shows that our agent is able to instruct the reasoning process to improve the final outputs for different tasks, and zero-shot AgentInstruct is able to generalize LLMs' reasoning abilities across tasks. Note that we only run the agent 53 times, resulting in 53 agent generated instructions, as we evaluate on 53 subsets. With zero-shot AgentInstruct, we also observe a large margin of improvement for zero-shot performance across different models. Significantly, Llama-2-70b-chat beats the performance of zero-shot GPT-3.5 Turbo by 10.2% on average across all datasets. This indicates our agent instructions are the key to improving the reasoning performance of LLMs.

The most immediate comparison to zero-shot AgentInstruct is zero-shot CoT, as zero-shot AgentInstruct uses task-specific instructions instead of a fixed manual instruction. On average, across all three models, zero-shot AgentInstruct beats zero-shot CoT by 6.5%, with the largest growth being Vicuna-13b at 9.5%. On both generation and classification datasets, across three models, zero-shot AgentInstruct wins by 5.0% and 7.8% on each category respectively. This suggests that zero-shot AgentInstruct is able to generalize the zero-shot reasoning abilities of LLMs to both generation and classification tasks, and optimize the performance of specific tasks.

In particular, we look into the performance of reasoning tasks (Figure 5c). Of our three models, the average difference between zero-shot AgentInstruct and the zero-shot setting on reasoning tasks is 31.3%, whereas the difference between zero-shot AgentInstruct and zero-shot CoT is 10.5%. This shows that our task-specific instructions are more helpful for LLMs to break down tasks into more specific intermediate reasoning steps compared to the task-agnostic instructions in zero-shot CoT, which leads to improved final predictions.

Overall, zero-shot AgentInstruct wins on 9 of the 13 generation datasets, 11 of the 16 classification datasets, and 10 of the 12 reasoning datasets (Figure 4). See Appendix B for additional results and analysis, including results on individual datasets and subsets.

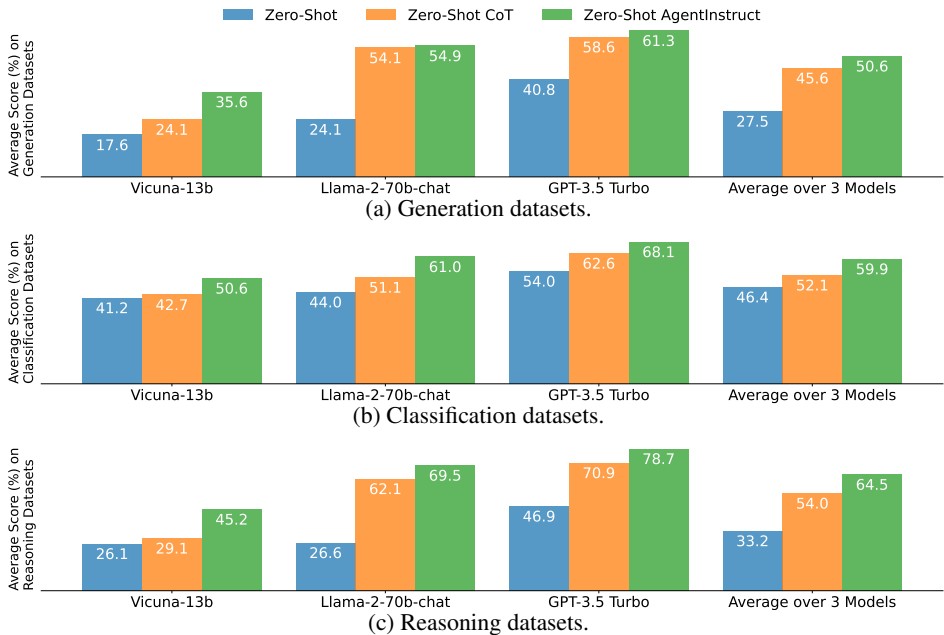

Figure 5: Results on Vicuna-13b, Llama-2-70b-chat, and GPT-3.5 Turbo across tasks. Top: generation. Middle: classification. Bottom: reasoning.

## 3.2 ABLATION STUDY

|  | AddSub | IMDB | NarrativeQA |
|---|---|---|---|
| Zero-Shot AgentInstruct | **79.5** | **94.0** | **65.0** |
| w/o Agent Instructions | 73.2 | 89.0 | 62.3 |
| w/o Input Examples | 72.4 | 88.0 | 60.1 |
| w/o Labels | 74.9 | 93.8 | 63.9 |
| w/o GPT-4 | 75.2 | 92.6 | 63.5 |

Table 1: Ablation over different facets of zero-shot AgentInstruct with Llama-2-70b-chat.

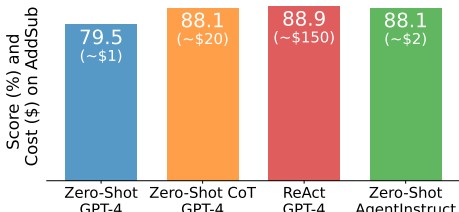

Figure 6: Comparison on GPT-4 using zero-shot, zero-shot CoT, ReAct, and zero-shot AgentInstruct on AddSub.

We examine how different components of zero-shot AgentInstruct impact its zero-shot reasoning performance. Results are shown in Table 1 on AddSub (reasoning), IMDB (classification), and NarrativeQA (generation). We use Llama-2-70b-chat for reasoning. The four settings examine the importance of agent instructions in zero-shot AgentInstruct. Descriptions of each setting are as follows: (i) w/o Agent Instructions: We compare the zero-shot AgentInstruct methodology to zero-shot CoT. (ii) w/o Input Examples: We remove the input-only examples from the input to the agent. (iii) w/o Labels: We remove the description of the labels from the input to the agent. (iv) w/o GPT-4: We use GPT-3.5 Turbo, instead of GPT-4, as the agent to generate instructions. The

results suggest that all components of zero-shot AgentInstruct are effective in providing high-quality instructions and eliciting high-quality reasoning steps before making predictions. See Appendix C.3 for further descriptions of each setting.

Next, we focus our analysis on GPT-4. We test the following methods: zero-shot, zero-shot CoT, ReAct (Yao et al., 2023b), and zero-shot AgentInstruct. Figure 6 shows the performance of each method on AddSub. Zero-shot AgentInstruct outperforms zero-shot GPT-4 by 8.6% and ties the performance of zero-shot CoT GPT-4 for approximately one-tenth of the cost. Zero-shot AgentInstruct is using GPT-3.5 Turbo for the CoT reasoning. This indicates that both task-specific instructions and CoT reasoning help improve the zero-shot performance of LLMs. Though ReAct narrowly outperforms zero-shot AgentInstruct, it costs nearly 100 times more since the zero-shot AgentInstruct agent is run only once per dataset rather than per instance. Each run of our agent to generate instructions cost less than $1. We also add results on IMDB, namely zero-shot GPT-4, zero-shot CoT GPT-4, and zero-shot AgentInstruct GPT-4 (the reasoning step also uses GPT-4), scoring 87.4, 96.1, and 96.6 respectively. This result implies that decoupling the agent instruction generation and reasoning steps further unleashes the zero-shot reasoning abilities of LLMs, and that zero-shot AgentInstruct is a cost-effective alternative to using agents directly, resulting in only a minimal performance loss.

## 3.3 TRUNCATED CONTEXT LENGTH

In Figure 7, we test how zero-shot AgentInstruct performs on various context lengths. We artificially reduce the context length of Llama-2-70b-chat from 4,000 (maximum context length) to 2,048, 1,024, 768, and 640. The results suggest that the performance of zero-shot AgentInstruct is worse for models with smaller context lengths. While the impact is minimal on AddSub and IMDB, performance on NarrativeQA steeply declines below a context length of 2,048, which is due to truncation. This is because the instances of NarrativeQA are much longer, with an average length of 855 tokens compared to the instances of AddSub and IMDB, with average lengths of 330 tokens and 48 tokens respectively.

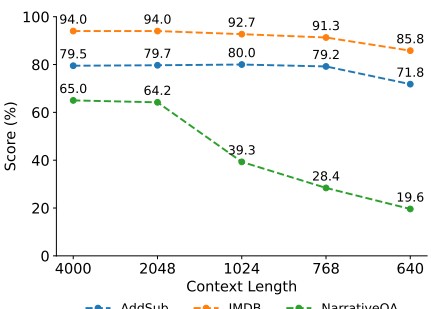

Figure 7: Truncating context lengths Llama-2-70b-chat with zero-shot AgentInstruct on AddSub, IMDB, and NarrativeQA.

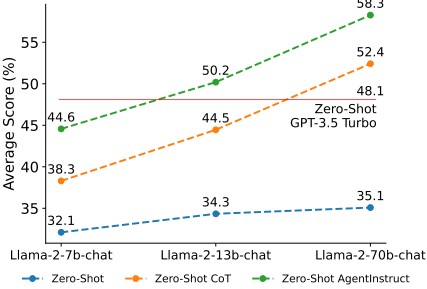

Figure 8: Model scaling results of zero-shot, zero-shot CoT, and zero-shot AgentInstruct with Llama-2-chat on all datasets.

## 3.4 MODEL SCALING

As it is often the case that larger models substantially improve the reasoning capabilities of LLMs (Wei et al., 2022b), we test zero-shot AgentInstruct's performance on models of various sizes. Specifically, we test on three Llama-2-chat models with 7 billion, 13 billion, and 70 billion parameters. Figure 8 shows the average score across all 29 datasets for zero-shot, zero-shot CoT, and zero-shot AgentInstruct. These results confirm that the average performance of all three methods increases with model size. Each time model size increases, zero-shot CoT and zero-shot AgentInstruct show consistent gains of around 6%, while zero-shot has smaller gains near 2%. This is because reasoning steps are best produced with more powerful models. In fact, at just 13b parameters, Llama-2-13b-chat with zero-shot AgentInstruct surpasses the performance of zero-shot GPT-3.5 Turbo by over 2%. Additionally, zero-shot AgentInstruct's superiority over zero-shot and zero-shot CoT appears independent of model size.

| CoT Reasoning | Prompt | Quasi-Exact Match (%) |
|---|---|---|
| Reasoning extraction | **Follow the instructions to generate an explanation that reasons towards the correct answer to the task above. End the explanation with the correct answer.\n\nExplanation:** | **79.5** |
| | Let's think step-by-step according to the instructions. End with the correct answer. Explanation: | 77.5 |
| | Let's think step-by-step according to the instructions. First, | 75.2 |
| | Use the instructions to guide you towards your answer.\nExplanation: | 79.0 |
| | Explanation: | 78.5 |
| Answer extraction | **Therefore, the answer to the task is below. Give the answer in the shortest form possible that will still be correct. \nAnswer:** | 79.5 |
| | Therefore, the answer to the task is below.\nAnswer: | **79.7** |
| | Therefore, the answer is | 79.2 |
| | Answer: | 77.7 |

Table 2: Prompt sensitivity analysis of chain of thought reasoning of zero-shot AgentInstruct with Llama-2-70b-chat on AddSub. The default prompts are highlighted. Higher scores are better.

## 3.5 MANUAL PROMPT SENSITIVITY

Zero-shot AgentInstruct has two manual prompts in the CoT reasoning step: (1) the reasoning extraction prompt, which asks for intermediate reasoning steps, and (2) the answer extraction prompt, which collects the final answer. To test the sensitivity of each prompt, we vary a single prompt while keeping the default zero-shot AgentInstruct prompt for the other. Results are shown in Table 2 based on Llama-2-70b-chat on AddSub. Overall, zero-shot AgentInstruct's performance does not appear particularly sensitive to changes in the manual prompts, suggesting that the methodology behind zero-shot AgentInstruct is robust. Additional prompt sensitivity experiments are conducted in Appendix C.2.

## 3.6 ERROR ANALYSIS

To investigate the reasons for errors made by zero-shot AgentInstruct, we select 25 samples from AddSub, IMDB, and NewsQA respectively where zero-shot AgentInstruct results in incorrect predictions on Llama-2-70b-chat. We define incorrect predictions as those with a quasi-exact match or F1 score less than 1. Table 3 shows our error analysis. The most common error across datasets is incorrect reasoning, i.e., not correctly reasoning through the problem when applying the accurate agent instructions. For example, on AddSub, zero-shot AgentInstruct chooses the wrong operation due to a misleading verb. On IMDB,

|  | Error Type | Percentage |
|---|---|---|
| Reasoning | Incorrect reasoning | 32.0 |
| | Not factual | 12.0 |
| Answer | Ambiguity | 22.7 |
| | Invalid label | 14.7 |
| | Short answer | 10.6 |
| | Incorrect format | 8.0 |

Table 3: Error analysis for zero-shot AgentInstruct Llama-2-70b-chat on AddSub, IMDB, and NewsQA.

zero-shot AgentInstruct misreads the sentiment due to emphasizing words describing the movie, not the review. See Figure 9 for an example of incorrect reasoning on IMDB. The answer ambiguity is another main source of errors. For example, in a review where the reviewer clearly enjoyed the movie even though the reviewer acknowledged it was a typical bad movie, our prediction is "Positive" while the ground truth is "Negative" on IMDB. For many errors, either the instructions are taken too literally or partially ignored. As larger models become better at reasoning, such errors should be minimized. More thorough error analysis and full examples for each error category are in Appendix C.1.

## 3.7 CASE STUDY

Next, we analyze the quality of the CoT reasoning steps when predictions are correct. On three datasets (AddSub, IMDB, and NewsQA), we randomly select 25 examples from each dataset with a perfect quasi-exact match or F1 score. We find that the reasoning capabilities are further enhanced by the combination of effective agent instructions and the task-specific reasoning process of LLMs. An example is in Figure 10.

Passage: As an avid Disney fan, I was not to tally impressed by this movie, certainly not motivated to catch it in the theaters. I am, however, so glad that I caught it on DVD and watched the special features. You MUST check out the "Moose commentary" The enjoyment I got from this commentary completely made up for the tepid reaction I had to the film itself.

**CoT Reasoning:** Based on the language used in the passage, it is clear that the reviewer has a positive sentiment towards the movie. The reviewer uses positive adjectives such as "enjoyment" to describe their experience with the movie...

Answer: Positive ✗

Passage: This is a good movie. Terrance Stamp is great, the music is sweet, Carol White is very believeable. The only thing that marred this was the shakey acting of Carol's first husband, but if you can get past that, you're OK. Donovan provides some of the most languid, mellow, bittersweet lyrics from the 60s.

**CoT Reasoning:** The reviewer mentions that the movie is "good" and that it has a "sweet" sound track. They also mention that the acting by Carol White is "believeable". However, they also mention that the acting by Carol's first husband is "shakey."

Answer: Positive ✓

Figure 9: An incorrect reasoning example for Llama-2-70b-chat with zero-shot AgentInstruct on IMDB for error analysis. Here, the model mistook the love of the commentary as the love of the movie (highlighted).

Figure 10: Case study example for Llama-2-70b-chat with zero-shot AgentInstruct on IMDB. Here, the answer is correct and the task-specific reasoning is helpful for finding the answer (highlighted).

### 3.8 COMPARISON TO RELATED METHODS

**Few-Shot** We compare Llama-2-70b-chat zero-shot AgentInstruct results with few-shot results on AddSub, IMDB, and NarrativeQA in Figure 11. Surprisingly, zero-shot AgentInstruct reaches competitiveness with few-shot prompting. Zero-shot AgentInstruct, without any few-shot examples, outperforms few-shot performance on AddSub and NarrativeQA, by over 4.3% and 23.7% respectively, and loses by 0.7% on IMDB. Ideally, all the information encoded in the few-shot examples can be presented in a clear manner within the agent instructions to better utilize the reasoning capabilities of models. As shown, zero-shot AgentInstruct has the potential to reach or even beat few-shot performance.

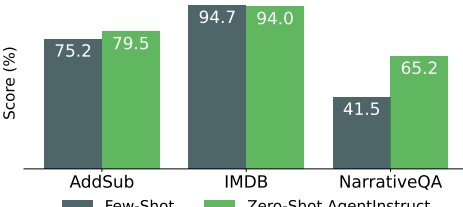

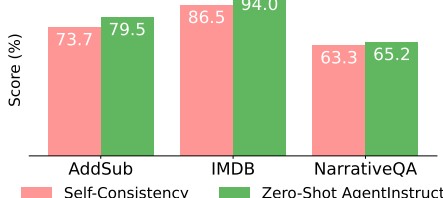

Figure 11: Comparison between zero-shot AgentInstruct and few-shot on Llama-2-70b-chat on AddSub, IMDB, and NarrativeQA.

Figure 12: Comparison between zero-shot AgentInstruct and zero-shot self-consistency on Llama-2-70b-chat.

**Self-Consistency** Finally, we compare zero-shot AgentInstruct results with self-consistency (Wang et al., 2023c) results of Llama-2-70b-chat on AddSub, IMDB, and NarrativeQA in Figure 12. We adapt self-consistency to the zero-shot setting as follows: We sample three responses using a temperature of 0.7, top-$k$ sampling with $k = 40$, and a randomly generated seed for each request. After cleaning the output, we use a majority vote to determine the consensus answer, choosing at random to break ties. On AddSub, IMDB, and NarrativeQA, zero-shot AgentInstruct outperforms self-consistency by 5.8%, 7.5%, and 1.9% respectively. Besides, zero-shot AgentInstruct is more computationally efficient than self-consistency as there is no need for sampling the reasoning paths multiple times.

## 4 RELATED WORK

Large language models, such as GPT-4 (OpenAI, 2023), GPT-3 (Brown et al., 2020), PaLM (Chowdhery et al., 2022), PaLM-2 (Anil et al., 2023), BLOOM (Scao et al., 2023), OPT (Zhang et al., 2022a), LLaMA (Touvron et al., 2023a), Llama-2 (Touvron et al., 2023b), and many others (Radford et al., 2019; Wang & Komatsuzaki, 2021; Black et al., 2021; Smith et al., 2022; Hoffmann et al., 2022; Penedo et al., 2023) have shown remarkable performance on natural language processing

(NLP) tasks. Following the pretraining phase, additional finetuning enables models (e.g., FLAN (Wei et al., 2022a), FLAN-T5 (Chung et al., 2022), InstructGPT (Ouyang et al., 2022)) to better align with human instructions to complete tasks. Moreover, instruction-tuning has enabled LLMs (e.g., GPT-3.5 Turbo (OpenAI, 2022), Llama-2-chat (Touvron et al., 2023b), Self-Instruct (Wang et al., 2023d), Alpaca (Taori et al., 2023), Vicuna (Chiang et al., 2023), Koala (Geng et al., 2023)) to better engage with users through single-turn or multi-turn dialogue. Zero-shot AgentInstruct builds on these instruction-following language models, enabling better zero-shot reasoning abilities through the use of agent instructions.

Language agents (Yao et al., 2023b; Shinn et al., 2023; Xu et al., 2023; Park et al., 2023; Zhou et al., 2023b; Andreas, 2022; Wang et al., 2023a; Xi et al., 2023; Sumers et al., 2023; Chan et al., 2023) have recently emerged due to the task planning capabilities of LLMs. Given a task that is often demonstrated in natural language, these agents aim to complete the task directly. Unlike existing agents, the goal of our agent is to generate task-specific instructions on how to complete the given task, decoupling the agent planning and reasoning steps. Besides reaching competitive performance with using agents directly, our design turns out to be more cost-effective.

Finetuning is an effective method for generating higher-quality responses from LLMs on downstream tasks (Liu et al., 2019; Howard & Ruder, 2018). As model scale increases, finetuning becomes less practical, which lightweight tuning methods (such as prefix tuning (Li & Liang, 2021), prompt learning (Lester et al., 2021; Liu et al., 2023), LoRA (Hu et al., 2022)) have tried to solve. Even with such methods, in-context prompting techniques have gained attention as an alternative. Few-shot learning, which involves providing a few examples demonstrating the task before prompting the models during inference, is often effective on a range of tasks (Brown et al., 2020; Dong et al., 2023). Chain of thought (CoT) prompting (Wei et al., 2022b) involves generating a series of intermediate reasoning steps, which can dramatically increase the performance of LLMs on complex reasoning tasks. While this reasoning behavior is traditionally learned from few-shot demonstrations, Kojima et al. (2022) extends CoT prompting to the zero-shot setting. More recently, new approaches such as self-consistency (Wang et al., 2023c), plan-and-solve prompting (Wang et al., 2023b), tree of thought (Yao et al., 2023a), and graph of thought (Besta et al., 2023) have further improved the reasoning quality. On the other hand, zero-shot AgentInstruct focuses on the zero-shot setup. It generalizes the reasoning abilities of LLMs to more tasks by utilizing task-specific instructions generated from our agent to better align the reasoning process with a particular task.

NLP benchmarking datasets provide a standardized interface to evaluate LLMs on specific downstream tasks. Common benchmarks (e.g., HELM (Liang et al., 2023), MMLU (Hendrycks et al., 2021), and many others (Wang et al., 2019a;b; Rajpurkar et al., 2016; Srivastava et al., 2023; Zhong et al., 2023; Suzgun et al., 2023; Chen et al., 2021)) have become part of the standard evaluation of LLMs. We benchmark our method on 29 datasets, including the core scenarios from HELM (Liang et al., 2023) and the reasoning datasets from Kojima et al. (2022). We also include results from these benchmarks as task-specific model results for comparison purposes. Besides reasoning tasks, our method generalizes to general language understanding benchmark tasks including generation and classification. For reasoning tasks, our method outperforms existing zero-shot approaches (Brown et al., 2020; Kojima et al., 2022) by a large margin.

## 5 CONCLUSION

Our work proposes a new way of improving the zero-shot reasoning abilities of large language models on general language understanding tasks. We build an agent to instruct the reasoning process of LLMs. Our agent automatically generates task-specific instructions for a wide set of tasks. The instructions are used to guide LLMs to reason better across these tasks to make high-quality predictions. Our method is zero-shot so no input-output examples are required to solve the task. Our results confirm the overall efficacy of our approach, leading to substantial improvements across various NLP tasks spanning generation, classification, and reasoning. Average score enhancements of 13.3%, 23.2%, and 17.0% are achieved over the standard zero-shot setting across 29 datasets for Vicuna-13b, Llama-2-70b-chat, and GPT-3.5 Turbo respectively. Our method wins on 20 of the 29 datasets used for evaluation. We believe zero-shot AgentInstruct's style of human-understandable reasoning, along with its utilization of an autonomous agent with access to a wide range of dataset knowledge, can replace more traditional styles of zero or few-shot prompting as models become equipped with stronger reasoning capabilities.

ETHICS STATEMENT

We hereby acknowledge that all of the co-authors of this work are aware of the provided *ICLR Code of Ethics* and honor the code of conduct. Our method is built on LLMs, for which the risks and potential harms are discussed in Brown et al. (2020); OpenAI (2023); Touvron et al. (2023b). A concern is LLMs' tendency to generate non-factual responses with a high degree of confidence. Our approach is advantageous in this regard, as guiding the model to output step-by-step reasoning leading to its answer offers researchers the opportunity to analyze cases of erroneous outputs. Moreover, our method shows notable improvement on safety benchmarks such as CivilComments and TruthfulQA, suggesting that the use of instructions to ground the reasoning along a task-specific path can reduce the risk of harmful outputs. However, better performance on benchmarks does not guarantee safe outputs, which requires future research from the community.

REPRODUCIBILITY STATEMENT

The source code is available at `https://anonymous.4open.science/r/AgentInstruct_ICLR2024`. Our datasets are based on existing benchmarks (Liang et al., 2023; Kojima et al., 2022). We provide specific model details in Appendix A.2 and additional dataset information in Appendix A.4.

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

## CHANGES FROM INITIAL ICLR SUBMISSION

- Clarified that zero-shot AgentInstruct is only run once per dataset and the cost of generating our instructions.

- Clarified that AgentInstruct is zero-shot as we do not use any examples that have ground truth labels.

- Clarified the action space and the question answering API tool we used.

- Clarified the difference between zero-shot CoT and zero-shot AgentInstruct.

- Added IMDB results on zero-shot, zero-shot CoT, and zero-shot AgentInstruct run on GPT-4.

- Added further descriptions of the different settings of Table 1.

## A  EXPERIMENTAL SETUP

### A.1  DATASETS

We evaluate zero-shot AgentInstruct on a wide selection of datasets consisting of all HELM core scenarios (Liang et al., 2023) and all the reasoning datasets used to benchmark zero-shot CoT (Kojima et al., 2022). The datasets are listed in Table 4, along with their categorizations. See Figure 13 for general details and A.4 for the collection and processing of each dataset in more detail.

### A.2  MODELS

We focus on the following large language models:

- Vicuna-13b (Chiang et al., 2023): A finetuned version of LLaMA-13b (Touvron et al., 2023a) which was trained on an additional 70 thousand user-shared conversations from ShareGPT (ShareGPT, 2023). Version 1.1 weight deltas were used.

- Llama-2-chat (Touvron et al., 2023b): A family of chat models based on Llama-2. Llama-2-chat was further trained using supervised finetuning, reward modeling, and reinforcement learning from human feedback, and comes in three sizes: 7b, 13b, and 70b. We focus on the 70b model, but share results from the 7b and 13b models in Figure 8 and Appendix B.

- GPT-3.5 Turbo (OpenAI, 2022): A finetuned version of GPT-3.5 which was further trained using reinforcement learning from human feedback. Specifically, the 0301 version was used.

### A.3  IMPLEMENTATION DETAILS

#### A.3.1  ZERO-SHOT AGENTINSTRUCT DETAILS

**Agent Instructions**    Before generating the instructions using zero-shot AgentInstruct, we query our question answering (QA) API built based on the Bing Web Search API to retrieve five relevant websites, each of which is added to a vector database. Our QA API is a retrieval question answering system built from that database, which is used as the only tool in our main agent. Since the documents are long, they are added to a vector database so they do not expand past the context length of the LLM powering the agent. We implement our agent based on LangChain's ReAct implementation (Chase, 2022; Yao et al., 2023b). To initiate instruction generation, we build the agent input based on basic dataset information including the name of the dataset, labels (if applicable, else type of dataset like generation), and a few input-only examples without ground truth labels. We use GPT-4 (OpenAI, 2023) inside our agent. We use the default temperature of 0.3 and the snapshot gpt-4-0613 when generating instructions. Given the temperature, if rerun it is not guaranteed to generate the same outputs each time, nor is there any guarantee that the same steps of our pipeline are reached. In the case that instructions were not returned, we reran the pipeline until there were valid instructions. We also reran the pipeline on a select number of datasets (NaturalQA (openbook), GSM8K) where the name of the dataset was unhelpful. In this case, we use an alias for the name of the dataset.

| Dataset | Task Category | Metric | Number of Instances |
|---|---|---|---|
| AddSub (Koncel-Kedziorski et al., 2016) | Generation, Reasoning | QEM | 395 |
| AQuA (Ling et al., 2017) | Classification, Reasoning | EM | 254 |
| BoolQ (Clark et al., 2019) | Classification | QPEM | 1,000 |
| CivilComments (Koh et al., 2021; Borkan et al., 2019) | Classification | QPEM | 6,688 |
| All | Classification | QPEM | 765 |
| Black | Classification | QPEM | 735 |
| Christian | Classification | QPEM | 733 |
| Female | Classification | QPEM | 730 |
| LGBTQ | Classification | QPEM | 735 |
| Male | Classification | QPEM | 748 |
| Muslim | Classification | QPEM | 760 |
| Other Religions | Classification | QPEM | 767 |
| White | Classification | QPEM | 715 |
| CNN/Daily Mail (See et al., 2017) | Generation | ROUGE-2 | 466 |
| Coin Flip (Wei et al., 2022b) | Classification, Reasoning | QEM | 500 |
| CommonsenseQA (Talmor et al., 2019) | Classification, Reasoning | EM | 1,000 |
| Date Understanding (Srivastava et al., 2023) | Classification, Reasoning | QEM | 369 |
| GSM8K (Cobbe et al., 2021) | Generation, Reasoning | QEM | 1,000 |
| HellaSwag (Zellers et al., 2019) | Classification | EM | 1,000 |
| IMDB (Maas et al., 2011) | Classification | QEM | 1,000 |
| Last Letter Concatenation (Wei et al., 2022b) | Generation, Reasoning | QEM | 500 |
| MMLU (Hendrycks et al., 2021) | Classification | EM | 514 |
| Abstract Algebra | Classification | EM | 100 |
| College Chemistry | Classification | EM | 100 |
| Computer Security | Classification | EM | 100 |
| Econometrics | Classification | EM | 114 |
| US Foreign Policy | Classification | EM | 100 |
| MS MARCO (Regular) (Bajaj et al., 2016) | Classification | RR@10 | 1,000 |
| MS MARCO (TREC) (Bajaj et al., 2016) | Classification | NDCG@10 | 43 |
| MultiArith (Roy & Roth, 2015) | Generation, Reasoning | QEM | 600 |
| NarrativeQA (Kočiský et al., 2018) | Generation | F1 | 355 |
| NaturalQuestions (closed-book) (Kwiatkowski et al., 2019) | Generation | F1 | 1,000 |
| NaturalQuestions (open-book) | Generation | F1 | 1,000 |
| NewsQA (Trischler et al., 2017) | Generation | F1 | 1,000 |
| OpenBookQA (Mihaylov et al., 2018) | Classification | EM | 500 |
| QuAC (Choi et al., 2018) | Generation | F1 | 1,000 |
| RAFT (Alex et al., 2021) | Classification | QEM | 440 |
| ADE Corpus v2 | Classification | QEM | 40 |
| Banking 77 | Classification | QEM | 40 |
| Neurips Impact Statement | Classification | QEM | 40 |
| One Stop English | Classification | QEM | 40 |
| Overruling | Classification | QEM | 40 |
| Semiconductor Org Types | Classification | QEM | 40 |
| Systematic Review Inclusion | Classification | QEM | 40 |
| Tai Saftey Research | Classification | QEM | 40 |
| Terms of Service | Classification | QEM | 40 |
| Tweet Eval Hate | Classification | QEM | 40 |
| Twitter Complaints | Classification | QEM | 40 |
| Shuffled Objects (Srivastava et al., 2023) | Classification, Reasoning | QEM | 2,750 |
| Five Objects | Classification, Reasoning | QEM | 1,000 |
| Seven Objects | Classification, Reasoning | QEM | 1,000 |
| Three Objects | Classification, Reasoning | QEM | 750 |
| SingleEq (Koncel-Kedziorski et al., 2016) | Generation, Reasoning | QEM | 508 |
| StrategyQA (Geva et al., 2021; Srivastava et al., 2023) | Classification, Reasoning | QEM | 458 |
| SVAMP (Patel et al., 2021) | Generation, Reasoning | QEM | 1,000 |
| TruthfulQA (Lin et al., 2022) | Classification | EM | 654 |
| XSUM (Narayan et al., 2018) | Generation | ROUGE-2 | 518 |

Table 4: Overview of each dataset used.

**Chain of Thought Reasoning**   The reasoning extraction prompt of the chain of thought reasoning pipeline is in Table 13. The answer extraction prompts for generation, classification (excluding multi-choice), and classification (multi-choice) datasets are in Tables 14, 15, and 16 respectively. The prompts for reasoning tasks follow the corresponding prompts of either generation or classification based on their general categorization. We take the output of the answer extraction prompt as the answer, applying parsing as dictated by the dataset. Note that for generation, we encouraged shorter answers in the answer extraction prompt due to the nature of the datasets we used.

```
You will be provided instructions for completing a task followed by a
task to complete.

Instructions:
{Agent Instructions}

{Test Instance}
Follow the instructions to generate an explanation that reasons towards
the correct answer to the task above. End the explanation with the
correct answer.

Explanation:
```

Figure 13: Reasoning extraction prompt for the CoT reasoning of zero-shot AgentInstruct.

```
{Reasoning Extraction Prompt}{Reasoning Extraction Prompt Output}

Therefore, the answer to the task is below. Give the answer in
the shortest form possible that will still be correct.
```

Figure 14: Answer extraction prompt for the CoT reasoning of zero-shot AgentInstruct on a generation dataset. Generation datasets that also belong to the reasoning category use the same prompt.

```
{Reasoning Extraction Prompt}{Reasoning Extraction Prompt Output}

Therefore, the correct label among {labels} (just the label) to
the original task is below.
```

Figure 15: Answer extraction prompt for the CoT reasoning of zero-shot AgentInstruct on a classification (excluding multi-choice) dataset. Classification (excluding multi-choice) datasets that also belong to the reasoning category use the same prompt.

Inference requests on Vicuna and Llama-2-chat were submitted from HELM to a TorchServe API running on a local cluster containing 2 nodes, each with 8x NVIDIA RTX A6000 GPUs. As a general rule, models were loaded at the highest precision while still allowing each GPU to contain an entire copy of the full model. Llama-2-7b-chat, Llama-2-13b-chat, and Vicuna-13b were run in FP16, while Llama-2-70b-chat was quantized to NF4. NaturalQuestions (open-book), NewsQA, and QuAC runs with Llama-2-70b-chat required 2 GPUs per worker in order to handle longer sequence lengths. Inference requests on GPT-3.5 Turbo were submitted to OpenAI's API.

```
{Reasoning Extraction Prompt}{Reasoning Extraction Prompt Output}

Therefore,  the  correct  multiple  choice  label  (just  the  letter)
to the task is below.
```

Figure 16: Answer extraction prompt for the CoT reasoning of zero-shot AgentInstruct on a classification (multi-choice) dataset. Classification (multi-choice) datasets that also belong to the reasoning category use the same prompt.

For Llama-2-chat runs, we prepend the necessary special tokens, including an empty system message. Also, we include a special designation [\INST] during the reasoning extraction prompt to signify the transition from User to Assistant roles. Initial experimentation with formatting revealed that Llama-2-chat was very sensitive to this designation, and would often fail to generate thoughtful reasoning steps if omitted during runs with zero-shot AgentInstruct or standard zero-shot CoT. We made no such User/Assistant distinction before the answer extraction prompts. For Vicuna and GPT-3.5 Turbo runs, we made no User/Assistant distinctions either, effectively treating all model input as part of the User role. Generally, we use each dataset's default generation parameters outlined in HELM. All inference was done using a temperature of 0.0, except on datasets that involved summarization, where a temperature of 0.3 was used. For the reasoning extraction prompt, we request a maximum of 512 new tokens, and for the answer extraction prompt, the number of tokens requested is specific to each dataset.

### A.3.2 EVALUATION DETAILS

Each dataset uses one of the following accuracy metrics:

- Exact Match (EM): A binary indicator that awards correctness if and only if the output exactly matches one or more of the correct references. We primarily use this metric for classification, in particular for multi-choice classification tasks.

- Quasi-Exact Match (QEM): A more lenient version of Exact Match which awards correctness if the output text matches the correct reference after both have been normalized (i.e., after removing extra white space, articles, punctuation, and capitalization) (Choi et al., 2018). This metric is primarily used for classification tasks, as well as some generation tasks.

- Quasi-Prefix Exact Match (QPEM): A more lenient version of Quasi-Exact Match that awards correctness if the normalized output begins with the normalized reference. Hence, "Yes, Elmendorf" would be marked correct if the reference was "Yes". The metric is useful on several classification tasks where the model has a tendency to generate additional tokens beyond that of the correct answer.

- F1: Instead of checking for an exact match, this metric will award partial correctness for any string overlap (Rajpurkar et al., 2016). The metric is used for a number of question answering tasks with open-ended generation.

- ROUGE-2: This metric scores correctness based on bigram overlap (Lin, 2004). We use ROUGE-2 as the accuracy measure for generation tasks involving summarization.

- RR@10: This metric is used for MS MARCO (Regular) and it uses the reciprocal rank of the first relevant document.

- NDCG@10: This metric is used for MS MARCO (TREC) to assess the normalized discounted cumulative gain using the set of graded rankings (Järvelin & Kekäläinen, 2002).

Prior to evaluation, we often perform some post-processing to clean the generated text. On mathematical generation datasets, we parse out the first number in the output and use that for our final answer. Similarly, on classification (multi-choice) datasets, we grab the first letter in the output. Appendix A.4 gives a compact description of each dataset, as well as the accuracy metric used, and any initial or post-processing done. These choices followed HELM or other relevant papers.

In general, we report evaluation results using the default accuracy metric outlined in HELM. For datasets with various subsets, we report the macro-average accuracy score across all subsets. Prompts, parsing techniques, and instructions were tested using 50 samples on most datasets to ensure the explanations and parsing were valid. Note that we are including these 50 samples in the full (up to) 1,000 sample evaluation, so there is a possibility of bias, especially regarding some of the smaller datasets. Note that all experiments done in this paper are the result of a single run on our pipeline. We only generate instructions once and run inference once as well due to the cost of running our models on all datasets with a high amount of samples.

## A.4 Additional Information About Each Dataset

### A.4.1 AddSub

**Description** AddSub (Koncel-Kedziorski et al., 2016) is a generation dataset containing addition and subtraction math word problems. We also classify AddSub as a reasoning dataset because of its inclusion in Kojima et al. (2022). The dataset contains 395 test instances.

**Implementation Details** AddSub is a generation dataset. Specifically, we prompt for an answer that is at most 10 tokens. During few-shot runs, we use the 5 in-context learning examples extracted from the math word problem instances presented in the chain of thought paper (Wei et al., 2022b). Responses undergo a post-processing step where the first number in the output is extracted for evaluation. The evaluation metric used is quasi-exact match.

**Examples** An example is shown in Figure 18.

**Comparison Methods** The few-shot results on the GPT-4 model achieved the state-of-the-art (Zhao et al., 2023).

**Results** The results are shown in Table 5.

### A.4.2 AQuA

**Description** AQuA (Ling et al., 2017) is a classification (multi-choice) dataset containing various math word problems. Because it is in Kojima et al. (2022), we consider it a reasoning dataset. The dataset contains 254 test instances.

**Implementation Details** We prompt for an answer that is at most 5 tokens. Responses undergo a post-processing step where the first letter in the output is extracted for evaluation. The evaluation metric used is exact match.

**Examples** An example is shown in Figure 19.

**Comparison Methods** The few-shot results on the GPT-4 model achieved the state-of-the-art (Zheng et al., 2023).

**Results** The results are shown in Table 5.

### A.4.3 BoolQ

**Description** BoolQ (Clark et al., 2019) is a classification benchmark where yes/no questions are asked about a given passage. We sample 1,000 test instances following HELM.

**Implementation Details** BoolQ is a core scenario in HELM (Liang et al., 2023), and we use the implementation parameters outlined in HELM. Specifically, we prompt for an answer that is at most 5 tokens. Quasi-prefix exact match is used for evaluation.

**Examples** An example is shown in Figure 20.

**Comparison Methods**    The result of finetuned ST-MoE-32B achieved the state-of-the-art (Zoph et al., 2022).

**Results**    The results are shown in Table 5.

### A.4.4    CIVILCOMMENTS

**Description**    Civil Comments (Koh et al., 2021; Borkan et al., 2019) is a classification dataset to identify toxicity in online comments. The dataset is split by demographic (All, Black, Christian, Female, LGBTQ, Male, Muslim, Other religions, White). We sample 1,000 test instances from each subset following HELM.

**Implementation Details**    CivilComments is a core scenario in HELM, and we use the implementation parameters outlined in HELM. We prompt for an answer that is at most 5 tokens. Following HELM, we report the macro-average accuracy of each subset below with respect to the toxic/non-toxic classes, as well as the average accuracy across all subsets. We found it necessary to map yes/no outputs to true/false during zero-shot runs. Quasi-prefix exact match is used for evaluation.

**Examples**    Examples are shown in Figures 21, 22, 23, 24, 25, 26, 27, 28, 29.

**Comparison Methods**    The few-shot results on the GPT-3.5 Turbo model achieved the state-of-the-art (Liang et al., 2023).

**Results**    The results are shown in Table 5.

### A.4.5    CNN/DAILY MAIL

**Description**    CNN/Daily Mail (See et al., 2017) is a generation dataset where the goal is to summarize a selection of news articles. The dataset contains 466 test instances.

**Implementation Details**    CNN/Daily Mail is a core scenario in HELM. We generate a summary of no more than 128 tokens and use a temperature of 0.3. The ROUGE-2 metric is used for evaluation.

**Examples**    An example is shown in Figure 30.

**Comparison Methods**    The result of the PEGASUS$_{\text{LARGE}}$ (HugeNews) model achieved the state-of-the-art (Zhang et al., 2020).

**Results**    The results are shown in Table 5.

### A.4.6    COIN FLIP

**Description**    Coin Flip (Wei et al., 2022b) is a classification task where the objective is to determine the state of a coin after a number of successive flips. We consider Coin Flip a reasoning task because it is included in Kojima et al. (2022). The dataset contains 500 test instances.

**Implementation Details**    We recreated the Coin Flip dataset following the details in Kojima et al. (2022). We generate a response of no more than 4 tokens and use quasi-exact match for evaluation.

**Examples**    An example is shown in Figure 31.

**Comparison Methods**    The result of the GPT-3.5 (text-davinci-002) model achieved the state-of-the-art (Zhang et al., 2022b).

**Results**    The results are shown in Table 5.

### A.4.7 COMMONSENSEQA

**Description**  CommonsenseQA (Talmor et al., 2019) is a classification (multi-choice) dataset focusing on common sense. Because it is in Kojima et al. (2022), we consider it a reasoning dataset. We sample 1,000 instances from the test set mirroring the existing implementation in HELM.

**Implementation Details**  Though CommonsenseQA was already implemented into HELM, we added a separate implementation to better align with Kojima et al. (2022). We use the joint multiple-choice adapter in HELM rather than the separate multiple-choice adapter. We generate a response of no more than 5 tokens, and grab the first letter from the output and then use exact match for evaluation.

**Examples**  An example is shown in Figure 32.

**Comparison Methods**  The result of the DeBERTaV3-large model achieved the state-of-the-art (Xu et al., 2022).

**Results**  The results are shown in Table 5.

### A.4.8 DATE UNDERSTANDING

**Description**  Date Understanding (Srivastava et al., 2023; Suzgun et al., 2023) is a classification (multi-choice) dataset from the BIG-bench benchmark (Srivastava et al., 2023), which aims to assess pretrained language models' capacity to comprehend dates through inquiries about specific days' dates. The dataset contains 369 test instances.

**Implementation Details**  We use the Date Understanding instances from Suzgun et al. (2023). We generate up to 60 tokens, though in practice, the generation ends much quicker when it hits a stop token: either "\n" or ")". As with other classification multi-choice, we extract the first letter from the response. Quasi-exact match is used as the evaluation metric.

**Examples**  An example is shown in Figure 33.

**Comparison Methods**  The few-shot CoT results on the PaLM 2 model achieved the state-of-the-art (Anil et al., 2023).

**Results**  The results are shown in Table 5.

### A.4.9 GSM8K

**Description**  GSM8K (Cobbe et al., 2021) is a generation dataset focused on mathematical reasoning. Because it is in Kojima et al. (2022), we consider it a reasoning dataset. We sample 1,000 test instances following HELM.

**Implementation Details**  GSM8K is included in HELM as a target scenario. We leverage this implementation, which generates up to 400 new tokens. Responses undergo a post-processing step where the first number in the output is extracted for evaluation. The evaluation metric used is quasi-exact match.

**Examples**  An example is shown in Figure 34.

**Comparison Methods**  The result of the GPT-4 code interpreter achieved the state-of-the-art (Zhou et al., 2023a).

**Results**  The results are shown in Table 5.

### A.4.10 HELLASWAG

**Description** HellaSwag (Zellers et al., 2019) is a classification (multi-choice) dataset comprised of 70,000 multiple-choice questions about grounded situations, originating from ActivityNet and wikiHow domains, with human-verified adversarial incorrect answers and one correct answer aimed at studying grounded commonsense inference. We sample 1,000 test instances following HELM.

**Implementation Details** We use the implementation of HellaSwag in HELM, but use the joint adapter to mimic a standard multiple-choice question, rather than the separate adapter which scores each reference separately using the sentence probability normalized by the sentence length. We generate only a single token. During zero-shot and zero-shot CoT runs, we include the following instructions from HELM: "The following are multiple choice questions (with answers) about common sense." The exact match metric is used for evaluation. Like with other classification (multi-choice) datasets, we parse out the first letter from the response for evaluation.

**Examples** An example is shown in Figure 35.

**Comparison Methods** The few-shot results on the GPT-4 model achieved the state-of-the-art (OpenAI, 2023).

**Results** The results are shown in Table 5.

### A.4.11 IMDB

**Description** IMDB (Maas et al., 2011) is a classification dataset comprising 50,000 movie reviews, with 25,000 for training and 25,000 for testing, focused on binary sentiment classification. We sample 1,000 test instances following HELM.

**Implementation Details** IMDB is a core scenario in HELM, and we follow their implementation. We generate up to 5 new tokens. During few-shot runs, we sampled 5 in-context examples from the IMDB training set. Quasi-exact match is used for evaluation.

**Examples** An example is shown in Figure 36.

**Comparison Methods** The result of the ERNIE-DOC-Large model achieved the state-of-the-art (Ding et al., 2021).

**Results** The results are shown in Table 5.

### A.4.12 LAST LETTER CONCATENATION

**Description** Last Letter Concatenation (Wei et al., 2022b) is a generation task where the objective is to concatenate the last letters of a series of two words. Because it is in Kojima et al. (2022), we consider it a reasoning dataset. We generate full names by randomly concatenating names from the top one-thousand first and last names from name census data (https://namecensus.com/), as referenced in Wei et al. (2022b). The dataset contains 500 test instances.

**Implementation Details** We recreated the Last Letter Concatenation dataset following the details in Kojima et al. (2022); Wei et al. (2022b). Unlike Kojima et al. (2022) which does concatenation on 4 words, we only do so on 2. We generate a response of no more than 4 tokens. We noticed that quasi-exact match unfairly penalized zero-shot CoT runs, so for all runs we extract the answer directly from the generated text, grabbing only the first 2 letters after stripping away white space and punctuation.

**Examples** An example is shown in Figure 37.

**Comparison Methods** The few-shot CoT results on the PaLM-540B model achieved the state-of-the-art (Wei et al., 2022b).

**Results**  The results are shown in Table 5.

### A.4.13  MMLU

**Description**  MMLU (Hendrycks et al., 2021) is a classification (multi-choice) benchmark comprised of 57 diverse subjects, ranging from elementary to professional levels, encompassing STEM, humanities, social sciences, and specialized fields. Following HELM, we focus on 5 subjects (Abstract Algebra, College Chemistry, Computer Security, Econometrics, and US Foreign Policy) totaling 514 test instances.

**Implementation Details**  MMLU is a core HELM scenario, and we use HELM's implementation of the dataset. We prompt for a single token. HELM's implementation includes the following instructions, where "{subject}" is replaced with the subject of the dataset: "The following are multiple choice questions (with answers) about {subject}." We prepend these instructions to instances during zero-shot and zero-shot CoT runs. We parse out the first letter from the response, and the exact match metric is used for evaluation.

**Examples**  Examples are shown in Figures 38, 39, 40, 41, 42.

**Comparison Methods**  The few-shot results on the Palmyra X (43B) model achieved the state-of-the-art (Liang et al., 2023).

**Results**  The results are shown in Table 5.

### A.4.14  MS MARCO (REGULAR)

**Description**  MS MARCO (Bajaj et al., 2016; Nogueira et al., 2020; MacAvaney et al., 2021) is a classification dataset containing real Bing search queries and nearly 9 million web passages, annotated for relevance. It consists of two tracks: the Regular track uses binary RR@k metric, while the TREC track employs graded annotations with NDCG as the main metric, featuring more extensively annotated queries and passages with relevance grades. We sample 1,000 test instances for the Regular track following HELM.

**Implementation Details**  We use the default HELM implementation for MS MARCO (Regular) and evaluate using RR@10.

**Examples**  An example is shown in Figures 43.

**Comparison Methods**  The few-shot results on the Cohere Command beta (52.4B) model achieved the state-of-the-art (Liang et al., 2023).

**Results**  The results are shown in Table 5.

### A.4.15  MS MARCO (TREC)

**Description**  This is the TREC track of MS MARCO (Bajaj et al., 2016; Nogueira et al., 2020; MacAvaney et al., 2021) as introduced in Appendix A.4.14. The TREC track contains 43 instances.

**Implementation Details**  We use the default HELM implementation for MS MARCO (TREC) and evaluate using NDCG@10.

**Examples**  An example is shown in Figure 44.

**Comparison Methods**  The few-shot results on the Cohere Command beta (52.4B) model achieved the state-of-the-art (Liang et al., 2023).

**Results**  The results are shown in Table 5.

### A.4.16 MULTIARITH

**Description**  MultiArith (Roy & Roth, 2015) is a generation dataset containing a comprehensive collection of arithmetic word problems, encompassing various mathematical operations and ranging in complexity. Because it is in Kojima et al. (2022) as a reasoning task, we consider it a reasoning dataset. The dataset contains 600 test instances.

**Implementation Details**  We prompt for up to 10 new tokens. As with other math generation datasets, we extract the first number from the generated text and use that as the final answer. Quasi-exact match is used for evaluation.

**Examples**  An example is shown in Figure 45.

**Comparison Methods**  The result of the self-consistency strategy on the GPT-3 (code-davinci-002) model achieved the state-of-the-art (Wang et al., 2023c).

**Results**  The results are shown in Table 5.

### A.4.17 NARRATIVEQA

**Description**  NarrativeQA (Kočiský et al., 2018) is a generation dataset focused on reading comprehension where the model answers a question about a passage. The dataset is comprised of 355 test instances.

**Implementation Details**  We use the implementation of NarrativeQA in HELM. We generate up to 100 new tokens, and use F1 score for evaluation. For few-shot runs, we sampled 5 in-context examples from the training set.

**Examples**  An example is shown in Figure 46.

**Comparison Methods**  The few-shot results on the Llama 2 (70B) model achieved the state-of-the-art (Liang et al., 2023).

**Results**  The results are shown in Table 5.

### A.4.18 NATURALQUESTIONS (CLOSED-BOOK)

**Description**  NaturalQuestions (Kwiatkowski et al., 2019) is a generation dataset comprised of anonymized real search queries posed to Google. We focus on two subsets: closed-book where the model is not given any external context, and open-book long answer, where a passage is given prior to the question. We sample 1,000 test instances following HELM.

**Implementation Details**  We use the implementation of the Natural Questions dataset within HELM. Up to 300 new tokens are generated, and we use F1 score as the evaluation metric.

**Examples**  An example is shown in Figure 47.

**Comparison Methods**  The few-shot results on the Llama 2 (70B) model achieved the state-of-the-art on the NaturalQuestions (closed-book) dataset (Liang et al., 2023).

**Results**  The results are shown in Table 5.

### A.4.19 NATURALQUESTIONS (OPEN-BOOK)

**Description**  This is the open-book subset of NaturalQuestions (Kwiatkowski et al., 2019) as introduced in Appendix A.4.18.

**Implementation Details**  We use the same implementation as introduced in Appendix A.4.18.

**Examples**   An example is shown in Figure 48.

**Comparison Methods**   The few-shot results on the text-davinci-003 model achieved the state-of-the-art on the NaturalQuestions (open-book) dataset (Liang et al., 2023).

**Results**   The results are shown in Table 5.

### A.4.20   NEWSQA

**Description**   NewsQA (Trischler et al., 2017) is a generation dataset that contains human-generated question-answer pairs sourced from more than 10,000 CNN news articles. We sample 1,000 test instances following HELM.

**Implementation Details**   We followed the instructions in HELM to collect and process the NewsQA dataset. We generate up to 50 tokens. F1 score is used as the evaluation metric.

**Examples**   Because of restrictive licensing, we cannot publish an example.

**Comparison Methods**   The result of the SpanBERT model achieved the state-of-the-art (Joshi et al., 2020).

**Results**   The results are shown in Table 5.

### A.4.21   OPENBOOKQA

**Description**   OpenBookQA (Mihaylov et al., 2018) is a classification (multi-choice) dataset focused on commonsense knowledge utilization. It contains 500 test instances.

**Implementation Details**   We follow the implementation of OpenbookQA in HELM, but use the joint adapter to mimic a standard multiple-choice question, rather than the separate adapter which scores each reference separately using the sentence probability normalized by the sentence length. We generate only a single token. During zero-shot and zero-shot CoT runs, we include the following instructions from HELM: "The following are multiple choice questions (with answers) about common sense." The first letter from the response is parsed out and the exact match metric is used for evaluation.

**Examples**   An example is shown in Figure 49.

**Comparison Methods**   The result of the fine-tuning method with the self-consistency prompting method on the PaLM-540B model achieved the state-of-the-art (Huang et al., 2022).

**Results**   The results are shown in Table 5.

### A.4.22   QUAC

**Description**   QuAC (Choi et al., 2018) is a generation dataset that facilitates information-seeking dialogue modeling by presenting interactive conversations between a student asking open-ended questions and a teacher providing short text excerpts from a hidden Wikipedia passage. There are 1,000 test instances following HELM.

**Implementation Details**   We use the default QuAC implementation in HELM. Responses are limited to 100 tokens, and F1 is used to measure accuracy.

**Examples**   An example is shown in Figure 50.

**Comparison Methods**   The result of the FLOWQA model achieved the state-of-the-art (Huang et al., 2019).

**Results**   The results are shown in Table 5.

### A.4.23   RAFT

**Description**   RAFT (Alex et al., 2021) is a classification benchmark designed to assess language models' performance across diverse domains. Specifically, we focus on 11 subsets, containing 40 test instances each.

**Implementation Details**   We use the default implementation of RAFT in HELM. Responses are limited to 30 tokens. For zero-shot and zero-shot CoT runs, we prepend subset-specific task instructions provided in HELM. We report the average score across the 11 subsets using quasi-exact match as the evaluation metric.

**Examples**   Examples are shown in Figures 51, 52, 53, 54, 55, 56, 57, 58, 59, 60, 61.

**Comparison Methods**   The few-shot results on the GPT-3.5 Turbo model achieved the state-of-the-art (Liang et al., 2023).

**Results**   The results are shown in Table 5.

### A.4.24   SHUFFLED OBJECTS

**Description**   Shuffled Objects (Srivastava et al., 2023; Suzgun et al., 2023) is a classification (multi-choice) dataset of the BIG-bench benchmark. In this dataset, objects initially owned by different people are swapped through pairwise trades, and the model's goal is to accurately determine which person ends up with a particular object. There are 3 subtasks (three objects, five objects, and seven objects) in this dataset. The five and seven objects subtasks each contain 1000 instances, and the three object subtask contains 750 test instances.

**Implementation Details**   We use the Shuffled Objects instances from Suzgun et al. (2023). We generate up to 60 tokens, though in practice, the generation ends much quicker when it hits a stop token: either "\n" or ")". Unlike Kojima et al. (2022), we report results on the three, five, and seven object cases, instead of only the three object case. As with other classification (multi-choice) datasets, we parse out the first letter in the response. Quasi-exact match is used as the evaluation metric.

**Examples**   Examples are shown in Figures 63, 64, 65.

**Comparison Methods**   The few-shot CoT results on the PaLM 2 model achieved the state-of-the-art (Anil et al., 2023).

**Results**   The results are shown in Table 5.

### A.4.25   SINGLEEQ

**Description**   SingleEq (Koncel-Kedziorski et al., 2016) is a generation dataset containing math word problems requiring the use of single mathematical equations. Because the dataset is included in Kojima et al. (2022), we consider it a reasoning task in addition to generation. The dataset contains 508 test instances.

**Implementation Details**   We followed the instructions in Kojima et al. (2022) to collect the SingleEQ dataset. We limit generated responses to 10 tokens. Like other math generation datasets, we parse out the first number from the response, and then use quasi-exact match to measure accuracy.

**Examples**   An example is shown in Figure 62.

**Comparison Methods**   The few-shot results on the GPT-4 model achieved the state-of-the-art (Zhao et al., 2023).

**Results**   The results are shown in Table 5.

### A.4.26   STRATEGYQA

**Description**   StrategyQA (Geva et al., 2021) is a yes/no classification dataset, though we follow the BIG-bench implementation which treats instances as multiple-choice questions with two options (A. Yes, B. No). Because it is in Kojima et al. (2022), we consider it a reasoning dataset. The dataset contains 458 test instances.

**Implementation Details**   We use the StragetyQA implementation through BIG-bench. As a result, our implementation varies from that in Kojima et al. (2022) in that the instances are formulated as a classification (multi-choice) task, rather than an open-ended classification task. We prompt for no more than 64 new tokens. We extract the first letter from the response, and use exact match to measure accuracy.

**Examples**   An example is shown in Figure 66.

**Comparison Methods**   The few-shot CoT results with self-consistency on the PaLM 2 model achieved the state-of-the-art (Anil et al., 2023).

**Results**   The results are shown in Table 5.

### A.4.27   SVAMP

**Description**   SVAMP (Patel et al., 2021) is a generation dataset focused on mathematical reasoning containing 1,000 test instances. Because it is in Kojima et al. (2022), we also consider it a reasoning dataset.

**Implementation Details**   We prompt for at most 10 tokens. Like other math generation datasets, we parse the first number out of the response, and use quasi-exact match for evaluation.

**Examples**   An example is shown in Figure 67.

**Comparison Methods**   The few-shot results on the GPT-4 model achieved the state-of-the-art (Zhao et al., 2023).

**Results**   The results are shown in Table 5.

### A.4.28   TRUTHFULQA

**Description**   TruthfulQA (Lin et al., 2022) is a classification (multi-choice) task designed to test LLMs' propensity to dispense harmful information. The dataset contains 654 test instances.

**Implementation Details**   We use the implementation of TruthfulQA in HELM, which prompts for only a single token. We extract the first letter and use exact match for evaluation.

**Examples**   An example is shown in Figure 68.

**Comparison Methods**   The few-shot results on the Palmyra X (43B) model achieved the state-of-the-art (Liang et al., 2023).

**Results**   The results are shown in Table 5.

### A.4.29   XSUM

**Description**   XSUM (Narayan et al., 2018) is a generation dataset focused on summarization. The dataset contains 518 test instances.

**Implementation Details**  XSUM is a core scenario in HELM. We generate a summary of no more than 128 tokens and use a temperature of 0.3. The ROUGE-2 metric is used for evaluation.

**Examples**  An example is shown in Figure 69.

**Comparison Methods**  The result of the PEGASUS$_{LARGE}$ (HugeNews) model achieved the state-of-the-art (Zhang et al., 2020).

**Results**  The results are shown in Table 5.

## B  ADDITIONAL RESULTS

Table 5 presents the results of zero-shot, zero-shot CoT, and zero-shot AgentInstruct on Vicuna-13b, Llama-2-70b-chat, and GPT-3.5 Turbo, as well as the smaller 7b and 13b sizes of Llama-2-chat, across 29 datasets and 53 subsets. Also, scores of task-specific models are included for comparison. If there are multiple subsets, the score of task-specific models is only given for all subsets on average, not for individual subsets. Figure 17 plots the results for each dataset on Vicuna-13b, Llama-2-70b-chat, and GPT-3.5 Turbo, and Table 6 gives the average for each of the five models on each task category. Similar to our three main models, Llama-2-7b-chat and Llama-2-13b-chat with zero-shot AgentInstruct outperform both zero-shot and zero-shot CoT, on average, in each category of tasks.

| Dataset | Subset | Task-Specific Model | Model | Results | | |
|---|---|---|---|---|---|---|
| | | | | Zero-Shot(%) | Zero-Shot CoT(%) | Zero-Shot AgentInstruct(%) |
| AddSub (Koncel-Kedziorski et al., 2016) | - | 95.7 (Zhao et al., 2023) | Vicuna-13b | 10.9 | 29.1 | 60.8 |
| | | | Llama-2-7b-chat | 50.4 | 56.7 | 63.8 |
| | | | Llama-2-13b-chat | 52.7 | 52.4 | 65.1 |
| | | | Llama-2-70b-chat | 36.2 | 73.2 | 79.5 |
| | | | GPT-3.5 Turbo | 78.7 | 79.0 | 88.1 |
| AQuA (Ling et al., 2017) | - | 79.9 (Zheng et al., 2023) | Vicuna-13b | 25.6 | 16.5 | 24.0 |
| | | | Llama-2-7b-chat | 20.9 | 1.2 | 22.0 |
| | | | Llama-2-13b-chat | 18.9 | 24.0 | 25.2 |
| | | | Llama-2-70b-chat | 19.7 | 35.8 | 37.8 |
| | | | GPT-3.5 Turbo | 16.5 | 60.2 | 57.9 |
| BoolQ (Clark et al., 2019) | - | 92.4 (Zoph et al., 2022) | Vicuna-13b | 77.2 | 71.0 | 74.2 |
| | | | Llama-2-7b-chat | 77.5 | 76.4 | 72.8 |
| | | | Llama-2-13b-chat | 79.9 | 76.1 | 76.1 |
| | | | Llama-2-70b-chat | 78.5 | 83.2 | 84.6 |
| | | | GPT-3.5 Turbo | 74.0 | 82.1 | 83.6 |
| | Average across all 9 subsets | 68.4 (Liang et al., 2023) | Vicuna-13b | 56.1 | 51.2 | 60.6 |
| | | | Llama-2-7b-chat | 57.7 | 54.3 | 58.1 |
| | | | Llama-2-13b-chat | 66.1 | 52.0 | 64.7 |
| | | | Llama-2-70b-chat | 66.9 | 57.4 | 65.5 |
| | | | GPT-3.5 Turbo | 45.0 | 64.4 | 67.5 |
| | All | - | Vicuna-13b | 60.3 | 51.6 | 63.6 |
| | | | Llama-2-7b-chat | 65.3 | 58.8 | 59.6 |
| | | | Llama-2-13b-chat | 63.9 | 59.1 | 59.0 |
| | | | Llama-2-70b-chat | 64.8 | 61.4 | 70.2 |
| | | | GPT-3.5 Turbo | 52.1 | 67.7 | 68.8 |
| | Black | - | Vicuna-13b | 55.0 | 51.4 | 61.1 |
| | | | Llama-2-7b-chat | 54.7 | 51.1 | 64.3 |
| | | | Llama-2-13b-chat | 71.1 | 52.7 | 68.1 |
| | | | Llama-2-70b-chat | 71.9 | 58.6 | 68.2 |
| | | | GPT-3.5 Turbo | 44.5 | 66.7 | 63.0 |
| | Christian | - | Vicuna-13b | 59.5 | 54.4 | 61.4 |
| | | | Llama-2-7b-chat | 62.0 | 58.7 | 66.6 |
| | | | Llama-2-13b-chat | 66.4 | 48.4 | 62.8 |
| | | | Llama-2-70b-chat | 65.1 | 56.2 | 72.7 |
| | | | GPT-3.5 Turbo | 53.2 | 67.5 | 74.7 |
| CivilComments (Koh et al., 2021; Borkan et al., 2019) | Female | - | Vicuna-13b | 55.1 | 48.3 | 65.1 |
| | | | Llama-2-7b-chat | 56.3 | 54.6 | 57.1 |
| | | | Llama-2-13b-chat | 68.6 | 52.6 | 71.0 |
| | | | Llama-2-70b-chat | 72.2 | 55.8 | 55.1 |
| | | | GPT-3.5 Turbo | 47.1 | 67.0 | 71.2 |

| Dataset | Subset | Task-Specific Model | Model | Results | | |
|---|---|---|---|---|---|---|
| | | | | Zero-Shot(%) | Zero-Shot CoT(%) | Zero-Shot AgentInstruct(%) |
| | LGBTQ | - | Vicuna-13b | 55.1 | 49.0 | 57.3 |
| | | | Llama-2-7b-chat | 53.8 | 56.6 | 56.1 |
| | | | Llama-2-13b-chat | 63.4 | 49.5 | 64.0 |
| | | | Llama-2-70b-chat | 68.9 | 56.7 | 64.7 |
| | | | GPT-3.5 Turbo | 41.6 | 62.3 | 65.6 |
| | Male | - | Vicuna-13b | 57.5 | 51.0 | 54.4 |
| | | | Llama-2-7b-chat | 59.4 | 54.9 | 55.6 |
| | | | Llama-2-13b-chat | 67.1 | 55.7 | 60.8 |
| | | | Llama-2-70b-chat | 66.8 | 56.3 | 66.7 |
| | | | GPT-3.5 Turbo | 49.2 | 66.0 | 67.7 |
| | Muslim | - | Vicuna-13b | 52.6 | 51.8 | 60.0 |
| | | | Llama-2-7b-chat | 55.3 | 52.7 | 53.2 |
| | | | Llama-2-13b-chat | 63.5 | 52.3 | 67.0 |
| | | | Llama-2-70b-chat | 64.3 | 55.7 | 65.4 |
| | | | GPT-3.5 Turbo | 40.4 | 61.0 | 68.2 |
| | Other Religions | - | Vicuna-13b | 57.9 | 52.1 | 63.6 |
| | | | Llama-2-7b-chat | 60.6 | 51.7 | 58.9 |
| | | | Llama-2-13b-chat | 69.0 | 46.5 | 68.1 |
| | | | Llama-2-70b-chat | 67.3 | 56.3 | 67.0 |
| | | | GPT-3.5 Turbo | 41.2 | 60.9 | 68.4 |
| | White | - | Vicuna-13b | 51.7 | 51.8 | 58.6 |
| | | | Llama-2-7b-chat | 52.0 | 49.9 | 51.0 |
| | | | Llama-2-13b-chat | 61.8 | 51.6 | 61.7 |
| | | | Llama-2-70b-chat | 60.9 | 59.4 | 59.2 |
| | | | GPT-3.5 Turbo | 36.1 | 60.3 | 60.2 |
| CNN/Daily Mail (See et al., 2017) | - | 21.5 (Zhang et al., 2020) | Vicuna-13b | 15.0 | 12.3 | 12.6 |
| | | | Llama-2-7b-chat | 14.1 | 13.8 | 13.4 |
| | | | Llama-2-13b-chat | 14.2 | 12.5 | 13.0 |
| | | | Llama-2-70b-chat | 14.4 | 13.4 | 13.4 |
| | | | GPT-3.5 Turbo | 17.4 | 15.8 | 15.0 |
| Coin Flip (Wei et al., 2022b) | - | 99.9 (Zhang et al., 2022b) | Vicuna-13b | 49.0 | 27.0 | 52.4 |
| | | | Llama-2-7b-chat | 0.0 | 15.8 | 62.2 |
| | | | Llama-2-13b-chat | 0.0 | 18.8 | 65.0 |
| | | | Llama-2-70b-chat | 0.0 | 5.6 | 99.4 |
| | | | GPT-3.5 Turbo | 32.6 | 61.8 | 95.2 |
| CommonsenseQA (Talmor et al., 2019) | - | 91.2 (Xu et al., 2022) | Vicuna-13b | 60.4 | 48.0 | 61.8 |
| | | | Llama-2-7b-chat | 60.5 | 16.4 | 54.3 |
| | | | Llama-2-13b-chat | 63.1 | 60.0 | 62.6 |
| | | | Llama-2-70b-chat | 71.6 | 62.4 | 72.8 |
| | | | GPT-3.5 Turbo | 77.2 | 72.1 | 74.1 |
| Date Understanding (Srivastava et al., 2023) | - | 91.2 (Anil et al., 2023) | Vicuna-13b | 63.2 | 50.8 | 60.8 |
| | | | Llama-2-7b-chat | 38.8 | 46.4 | 46.4 |
| | | | Llama-2-13b-chat | 45.2 | 57.2 | 56.4 |
| | | | Llama-2-70b-chat | 56.8 | 68.4 | 70.8 |
| | | | GPT-3.5 Turbo | 35.6 | 70.8 | 71.6 |
| GSM8K (Cobbe et al., 2021) | - | 97.0 (Zhou et al., 2023a) | Vicuna-13b | 2.3 | 14.7 | 19.3 |
| | | | Llama-2-7b-chat | 3.1 | 30.0 | 21.9 |
| | | | Llama-2-13b-chat | 5.2 | 32.6 | 31.6 |
| | | | Llama-2-70b-chat | 2.2 | 54.6 | 51.6 |
| | | | GPT-3.5 Turbo | 8.4 | 77.8 | 73.4 |
| HellaSwag (Zellers et al., 2019) | - | 95.3 (OpenAI, 2023) | Vicuna-13b | 45.5 | 39.9 | 49.9 |
| | | | Llama-2-7b-chat | 41.5 | 42.8 | 33.5 |
| | | | Llama-2-13b-chat | 59.7 | 43.0 | 46.8 |
| | | | Llama-2-70b-chat | 65.0 | 61.9 | 60.6 |
| | | | GPT-3.5 Turbo | 75.8 | 57.1 | 73.5 |
| IMDB (Maas et al., 2011) | - | 97.1 (Ding et al., 2021) | Vicuna-13b | 5.5 | 69.5 | 94.2 |
| | | | Llama-2-7b-chat | 29.9 | 77.7 | 90.1 |
| | | | Llama-2-13b-chat | 41.2 | 78.4 | 90.6 |
| | | | Llama-2-70b-chat | 43.5 | 89.0 | 94.0 |
| | | | GPT-3.5 Turbo | 84.2 | 88.9 | 93.9 |
| Last Letter Concatenation (Wei et al., 2022b) | - | 99.4 (Wei et al., 2022b) | Vicuna-13b | 2.4 | 3.0 | 18.8 |
| | | | Llama-2-7b-chat | 0.2 | 3.2 | 25.4 |
| | | | Llama-2-13b-chat | 1.8 | 3.4 | 31.8 |
| | | | Llama-2-70b-chat | 3.0 | 3.4 | 35.6 |
| | | | GPT-3.5 Turbo | 4.8 | 3.4 | 99.8 |
| | Average across all 5 subsets | 60.9 (Liang et al., 2023) | Vicuna-13b | 44.1 | 44.2 | 44.5 |
| | | | Llama-2-7b-chat | 41.5 | 37.8 | 35.1 |
| | | | Llama-2-13b-chat | 46.8 | 45.8 | 46.6 |
| | | | Llama-2-70b-chat | 53.3 | 52.2 | 50.3 |

| Dataset | Subset | Task-Specific Model | Model | Results | | |
|---|---|---|---|---|---|---|
| | | | | Zero-Shot(%) | Zero-Shot CoT(%) | Zero-Shot AgentInstruct(%) |
| | | | GPT-3.5 Turbo | 58.6 | 62.3 | 58.9 |
| MMLU (Hendrycks et al., 2021) | Abstract Algebra | - | Vicuna-13b | 31.0 | 30.0 | 29.0 |
| | | | Llama-2-7b-chat | 25.0 | 24.0 | 18.0 |
| | | | Llama-2-13b-chat | 30.0 | 30.0 | 33.0 |
| | | | Llama-2-70b-chat | 37.0 | 28.0 | 35.0 |
| | | | GPT-3.5 Turbo | 38.0 | 50.0 | 36.0 |
| | College Chemistry | - | Vicuna-13b | 37.0 | 37.0 | 29.0 |
| | | | Llama-2-7b-chat | 26.0 | 30.0 | 26.0 |
| | | | Llama-2-13b-chat | 31.0 | 35.0 | 29.0 |
| | | | Llama-2-70b-chat | 39.0 | 41.0 | 32.0 |
| | | | GPT-3.5 Turbo | 43.0 | 50.0 | 55.0 |
| | Computer Security | - | Vicuna-13b | 60.0 | 60.0 | 63.0 |
| | | | Llama-2-7b-chat | 57.0 | 56.0 | 48.0 |
| | | | Llama-2-13b-chat | 66.0 | 64.0 | 68.0 |
| | | | Llama-2-70b-chat | 64.0 | 73.0 | 72.0 |
| | | | GPT-3.5 Turbo | 75.0 | 80.0 | 78.0 |
| | Econometrics | - | Vicuna-13b | 23.7 | 29.8 | 31.6 |
| | | | Llama-2-7b-chat | 30.7 | 22.8 | 25.4 |
| | | | Llama-2-13b-chat | 27.2 | 28.1 | 27.2 |
| | | | Llama-2-70b-chat | 39.5 | 43.0 | 40.4 |
| | | | GPT-3.5 Turbo | 53.0 | 44.7 | 44.7 |
| | US Foreign Policy | - | Vicuna-13b | 69.0 | 64.0 | 70.0 |
| | | | Llama-2-7b-chat | 69.0 | 56.0 | 58.0 |
| | | | Llama-2-13b-chat | 80.0 | 72.0 | 76.0 |
| | | | Llama-2-70b-chat | 87.0 | 76.0 | 72.0 |
| | | | GPT-3.5 Turbo | 84.0 | 87.0 | 81.0 |
| MS MARCO (Bajaj et al., 2016) | Regular | 47.2 (Liang et al., 2023) | Vicuna-13b | 11.7 | 14.0 | 19.0 |
| | | | Llama-2-7b-chat | 17.5 | 10.6 | 19.2 |
| | | | Llama-2-13b-chat | 7.7 | 12.4 | 18.8 |
| | | | Llama-2-70b-chat | 15.6 | 13.2 | 20.4 |
| | | | GPT-3.5 Turbo | 26.0 | 20.5 | 30.0 |
| | TREC | 76.2 (Liang et al., 2023) | Vicuna-13b | 27.9 | 31.0 | 34.6 |
| | | | Llama-2-7b-chat | 32.8 | 26.3 | 33.3 |
| | | | Llama-2-13b-chat | 20.9 | 31.1 | 38.9 |
| | | | Llama-2-70b-chat | 37.8 | 28.7 | 33.5 |
| | | | GPT-3.5 Turbo | 47.7 | 38.1 | 54.1 |
| MultiArith (Roy & Roth, 2015) | - | 100.0 (Wang et al., 2023c) | Vicuna-13b | 3.8 | 33.7 | 56.3 |
| | | | Llama-2-7b-chat | 4.8 | 60.5 | 65.2 |
| | | | Llama-2-13b-chat | 7.3 | 73.0 | 76.2 |
| | | | Llama-2-70b-chat | 5.8 | 88.3 | 88.2 |
| | | | GPT-3.5 Turbo | 50.3 | 96.7 | 96.5 |
| NarrativeQA (Kočiský et al., 2018) | - | 77.0 (Liang et al., 2023) | Vicuna-13b | 40.3 | 45.9 | 41.4 |
| | | | Llama-2-7b-chat | 38.8 | 55.6 | 61.1 |
| | | | Llama-2-13b-chat | 50.0 | 61.4 | 64.1 |
| | | | Llama-2-70b-chat | 45.9 | 62.3 | 65.0 |
| | | | GPT-3.5 Turbo | 49.1 | 67.7 | 65.2 |
| NaturalQuestions (Kwiatkowski et al., 2019) | NaturalQuestions (closed-book) | 45.8 (Liang et al., 2023) | Vicuna-13b | 14.6 | 18.3 | 27.5 |
| | | | Llama-2-7b-chat | 15.3 | 21.2 | 27.3 |
| | | | Llama-2-13b-chat | 18.7 | 26.7 | 33.2 |
| | | | Llama-2-70b-chat | 17.0 | 33.1 | 35.0 |
| | | | GPT-3.5 Turbo | 23.5 | 40.6 | 39.7 |
| | NaturalQuestions (open-book) | 77.0 (Liang et al., 2023) | Vicuna-13b | 39.3 | 47.2 | 55.5 |
| | | | Llama-2-7b-chat | 43.9 | 56.3 | 59.4 |
| | | | Llama-2-13b-chat | 43.3 | 61.7 | 64.1 |
| | | | Llama-2-70b-chat | 38.4 | 57.7 | 65.9 |
| | | | GPT-3.5 Turbo | 41.5 | 63.9 | 42.7 |
| NewsQA (Trischler et al., 2017) | - | 73.6 (Joshi et al., 2020) | Vicuna-13b | 37.2 | 27.2 | 33.7 |
| | | | Llama-2-7b-chat | 44.0 | 40.7 | 49.0 |
| | | | Llama-2-13b-chat | 40.9 | 45.5 | 51.5 |
| | | | Llama-2-70b-chat | 39.5 | 47.1 | 50.7 |
| | | | GPT-3.5 Turbo | 30.9 | 52.0 | 50.2 |
| OpenBookQA (Mihaylov et al., 2018) | - | 94.4 (Huang et al., 2022) | Vicuna-13b | 61.6 | 54.8 | 57.6 |
| | | | Llama-2-7b-chat | 56.8 | 57.0 | 56.2 |
| | | | Llama-2-13b-chat | 61.6 | 63.8 | 62.0 |
| | | | Llama-2-70b-chat | 73.8 | 75.0 | 70.4 |
| | | | GPT-3.5 Turbo | 81.6 | 78.6 | 79.2 |
| QuAC (Choi et al., 2018) | - | 64.1 (Huang et al., 2019) | Vicuna-13b | 37.5 | 17.9 | 16.8 |
| | | | Llama-2-7b-chat | 40.8 | 18.0 | 13.4 |
| | | | Llama-2-13b-chat | 42.5 | 19.3 | 14.0 |

| Dataset | Subset | Task-Specific Model | Model | Results | | |
|---|---|---|---|---|---|---|
| | | | | Zero-Shot(%) | Zero-Shot CoT(%) | Zero-Shot AgentInstruct(%) |
| | | | Llama-2-70b-chat | 46.8 | 15.6 | 16.6 |
| | | | GPT-3.5 Turbo | 38.6 | 33.1 | 34.7 |
| | Average across all 11 subsets | 76.8 (Liang et al., 2023) | Vicuna-13b | 19.1 | 54.5 | 51.4 |
| | | | Llama-2-7b-chat | 21.8 | 39.1 | 47.3 |
| | | | Llama-2-13b-chat | 17.3 | 43.0 | 56.6 |
| | | | Llama-2-70b-chat | 6.8 | 19.8 | 57.0 |
| | | | GPT-3.5 Turbo | 68.2 | 78.0 | 68.9 |
| | ADE Corpus v2 | - | Vicuna-13b | 5.0 | 77.5 | 75.0 |
| | | | Llama-2-7b-chat | 0.0 | 60.0 | 45.0 |
| | | | Llama-2-13b-chat | 62.5 | 75.0 | 57.5 |
| | | | Llama-2-70b-chat | 0.0 | 30.0 | 37.5 |
| | | | GPT-3.5 Turbo | 82.5 | 82.5 | 80.0 |
| | Banking 77 | - | Vicuna-13b | 15.0 | 12.5 | 12.5 |
| | | | Llama-2-7b-chat | 20.0 | 15.0 | 2.0 |
| | | | Llama-2-13b-chat | 2.5 | 10.0 | 2.5 |
| | | | Llama-2-70b-chat | 7.5 | 17.5 | 12.5 |
| | | | GPT-3.5 Turbo | 45.0 | 65.0 | 45.0 |
| | Neurips Impact Statement | - | Vicuna-13b | 0.0 | 75.0 | 70.0 |
| | | | Llama-2-7b-chat | 7.5 | 7.5 | 45.0 |
| | | | Llama-2-13b-chat | 0.0 | 47.5 | 72.5 |
| | | | Llama-2-70b-chat | 0.0 | 0.0 | 90.0 |
| | | | GPT-3.5 Turbo | 20.0 | 87.5 | 90.0 |
| | One Stop English | - | Vicuna-13b | 17.5 | 25.0 | 22.5 |
| | | | Llama-2-7b-chat | 27.5 | 32.5 | 25.0 |
| | | | Llama-2-13b-chat | 17.5 | 27.5 | 37.5 |
| | | | Llama-2-70b-chat | 0.0 | 32.5 | 37.5 |
| | | | GPT-3.5 Turbo | 35.0 | 35.0 | 25.0 |
| | Overruling | - | Vicuna-13b | 20.0 | 75.0 | 87.5 |
| | | | Llama-2-7b-chat | 2.5 | 55.0 | 77.5 |
| | | | Llama-2-13b-chat | 0.0 | 37.5 | 80.0 |
| | | | Llama-2-70b-chat | 0.0 | 0.0 | 85.0 |
| | | | GPT-3.5 Turbo | 95.0 | 97.5 | 95.0 |
| RAFT (Alex et al., 2021) | Semiconductor Org Types | - | Vicuna-13b | 30.0 | 25.0 | 40.0 |
| | | | Llama-2-7b-chat | 42.5 | 57.5 | 67.5 |
| | | | Llama-2-13b-chat | 0.0 | 52.5 | 90.0 |
| | | | Llama-2-70b-chat | 32.5 | 30.0 | 92.5 |
| | | | GPT-3.5 Turbo | 92.5 | 95.0 | 95.0 |
| | Systematic Review Inclusion | - | Vicuna-13b | 22.5 | 42.5 | 2.5 |
| | | | Llama-2-7b-chat | 65.0 | 40.0 | 15.0 |
| | | | Llama-2-13b-chat | 25.0 | 57.5 | 12.5 |
| | | | Llama-2-70b-chat | 15.0 | 2.5 | 7.5 |
| | | | GPT-3.5 Turbo | 85.0 | 95.0 | 12.5 |
| | Tai Safety Research | - | Vicuna-13b | 55.0 | 57.5 | 55.0 |
| | | | Llama-2-7b-chat | 57.5 | 60.0 | 45.0 |
| | | | Llama-2-13b-chat | 77.5 | 57.5 | 57.5 |
| | | | Llama-2-70b-chat | 20.0 | 80.0 | 60.0 |
| | | | GPT-3.5 Turbo | 77.5 | 72.5 | 65.0 |
| | Terms of Service | - | Vicuna-13b | 45.0 | 72.5 | 47.5 |
| | | | Llama-2-7b-chat | 17.5 | 40.0 | 55.0 |
| | | | Llama-2-13b-chat | 0.0 | 20.0 | 50.0 |
| | | | Llama-2-70b-chat | 0.0 | 15.0 | 35.0 |
| | | | GPT-3.5 Turbo | 85.0 | 90.0 | 85.0 |
| | Tweet Eval Hate | - | Vicuna-13b | 0.0 | 70.0 | 70.0 |
| | | | Llama-2-7b-chat | 0.0 | 32.5 | 67.5 |
| | | | Llama-2-13b-chat | 0.0 | 40.0 | 80.0 |
| | | | Llama-2-70b-chat | 0.0 | 0.0 | 85.0 |
| | | | GPT-3.5 Turbo | 47.5 | 50.0 | 80.0 |
| | Twitter Complaints | - | Vicuna-13b | 0.0 | 67.5 | 82.5 |
| | | | Llama-2-7b-chat | 0.0 | 30.0 | 57.5 |
| | | | Llama-2-13b-chat | 5.0 | 47.5 | 82.5 |
| | | | Llama-2-70b-chat | 0.0 | 10.0 | 85.0 |
| | | | GPT-3.5 Turbo | 85.0 | 87.5 | 85.0 |
| | Average across all 3 subsets | 79.3 (Anil et al., 2023) | Vicuna-13b | 20.0 | 20.1 | 18.0 |
| | | | Llama-2-7b-chat | 13.5 | 21.5 | 23.5 |
| | | | Llama-2-13b-chat | 7.6 | 22.7 | 23.9 |
| | | | Llama-2-70b-chat | 8.0 | 38.7 | 28.9 |
| | | | GPT-3.5 Turbo | 25.1 | 39.1 | 44.3 |
| Shuffled Objects (Srivastava et al., 2023) | Five Objects | - | Vicuna-13b | 13.6 | 18.0 | 16.8 |
| | | | Llama-2-7b-chat | 17.6 | 17.2 | 22.0 |

| Dataset | Subset | Task-Specific Model | Model | Results | | |
| --- | --- | --- | --- | --- | --- | --- |
| | | | | Zero-Shot(%) | Zero-Shot CoT(%) | Zero-Shot AgentInstruct(%) |
| | | | Llama-2-13b-chat | 4.0 | 18.8 | 21.6 |
| | | | Llama-2-70b-chat | 6.4 | 40.0 | 28.0 |
| | | | GPT-3.5 Turbo | 18.8 | 35.6 | 34.8 |
| | Seven Objects | - | Vicuna-13b | 15.2 | 15.6 | 10.8 |
| | | | Llama-2-7b-chat | 10.8 | 14.4 | 12.8 |
| | | | Llama-2-13b-chat | 6.8 | 16.8 | 16.8 |
| | | | Llama-2-70b-chat | 2.8 | 24.4 | 17.2 |
| | | | GPT-3.5 Turbo | 16.4 | 20.8 | 41.2 |
| | Three Objects | - | Vicuna-13b | 31.2 | 26.8 | 26.4 |
| | | | Llama-2-7b-chat | 12.0 | 32.8 | 35.6 |
| | | | Llama-2-13b-chat | 12.0 | 32.4 | 33.2 |
| | | | Llama-2-70b-chat | 14.8 | 51.6 | 41.6 |
| | | | GPT-3.5 Turbo | 40.0 | 60.8 | 56.8 |
| SingleEq (Koncel-Kedziorski et al., 2016) | - | 98.8 (Zhao et al., 2023) | Vicuna-13b | 11.0 | 29.1 | 63.4 |
| | | | Llama-2-7b-chat | 44.3 | 66.1 | 63.6 |
| | | | Llama-2-13b-chat | 46.9 | 51.0 | 65.0 |
| | | | Llama-2-70b-chat | 31.9 | 78.1 | 78.5 |
| | | | GPT-3.5 Turbo | 77.4 | 90.4 | 90.6 |
| StrategyQA (Geva et al., 2021) | - | 90.4 (Anil et al., 2023) | Vicuna-13b | 57.0 | 49.1 | 55.0 |
| | | | Llama-2-7b-chat | 53.5 | 55.2 | 52.4 |
| | | | Llama-2-13b-chat | 56.3 | 63.1 | 61.1 |
| | | | Llama-2-70b-chat | 60.9 | 68.6 | 67.2 |
| | | | GPT-3.5 Turbo | 59.2 | 67.7 | 69.0 |
| SVAMP (Patel et al., 2021) | - | 93.7 (Zhao et al., 2023) | Vicuna-13b | 7.2 | 22.5 | 44.5 |
| | | | Llama-2-7b-chat | 27.2 | 44.3 | 49.3 |
| | | | Llama-2-13b-chat | 36.6 | 45.6 | 56.1 |
| | | | Llama-2-70b-chat | 22.2 | 75.1 | 68.8 |
| | | | GPT-3.5 Turbo | 63.5 | 81.1 | 80.8 |
| TruthfulQA (Lin et al., 2022) | - | 61.6 (Liang et al., 2023) | Vicuna-13b | 35.8 | 42.2 | 52.3 |
| | | | Llama-2-7b-chat | 32.7 | 45.4 | 43.6 |
| | | | Llama-2-13b-chat | 34.6 | 46.5 | 48.8 |
| | | | Llama-2-70b-chat | 46.0 | 57.2 | 62.4 |
| | | | GPT-3.5 Turbo | 57.0 | 60.1 | 68.3 |
| XSUM (Narayan et al., 2018) | - | 24.6 (Zhang et al., 2020) | Vicuna-13b | 7.5 | 6.7 | 6.9 |
| | | | Llama-2-7b-chat | 7.0 | 6.3 | 6.5 |
| | | | Llama-2-13b-chat | 8.2 | 6.9 | 8.3 |
| | | | Llama-2-70b-chat | 9.0 | 8.5 | 8.6 |
| | | | GPT-3.5 Turbo | 13.2 | 9.5 | 8.8 |

Table 5: Results of zero-shot, zero-shot CoT, and zero-shot AgentInstruct on Vicuna-13b, Llama-2-7b-chat, Llama-2-13b-chat, Llama-2-70b-chat, and GPT-3.5 Turbo across 29 datasets and 53 subsets.

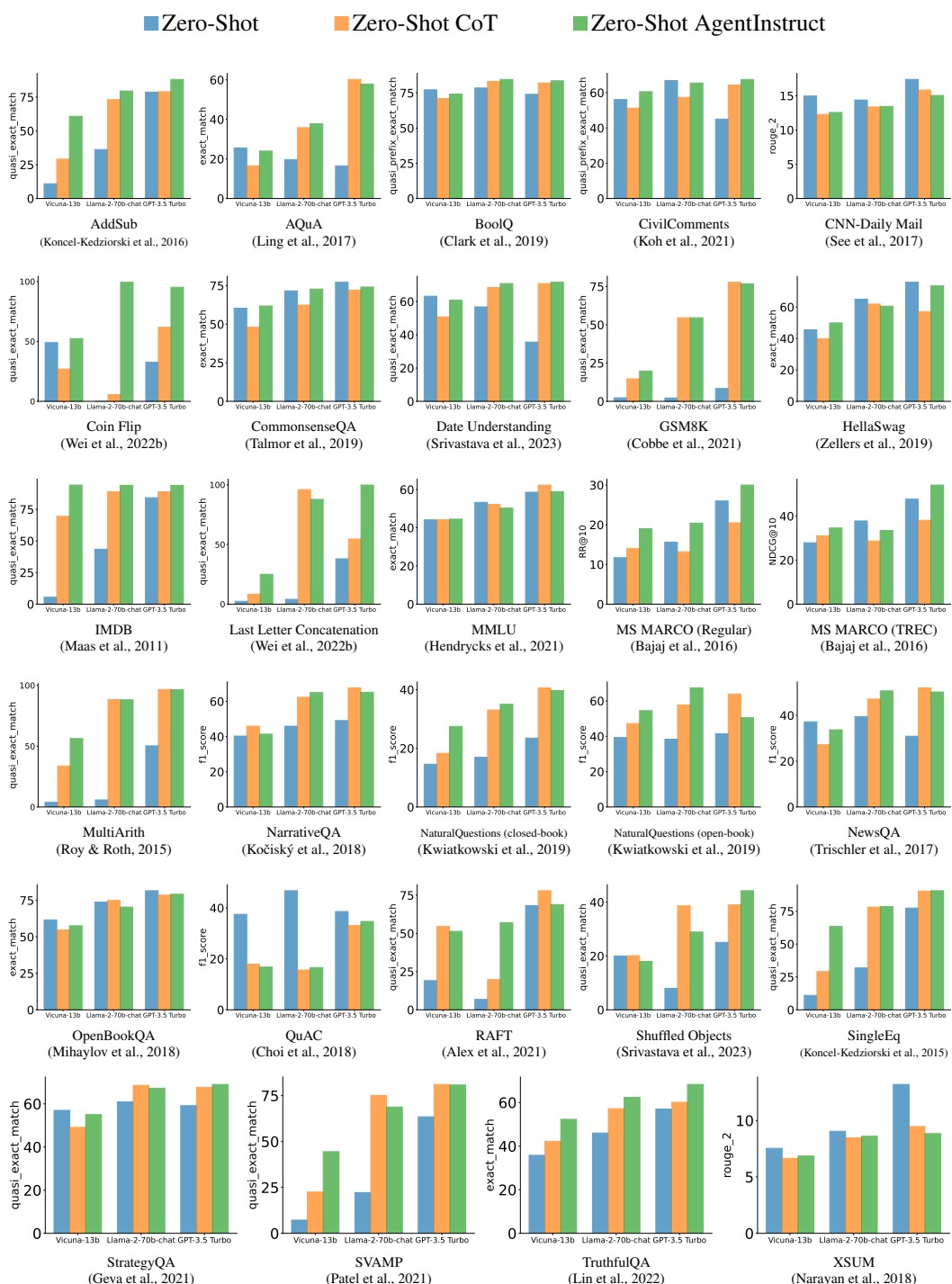

Figure 17: Main results (%) comparing zero-shot, zero-shot CoT and zero-shot AgentInstruct performance on Vicuna-13b (left), Llama-2-70b-chat (middle), and GPT-3.5 Turbo (right).

| Category | Model | Zero-Shot (%) | Zero-Shot CoT (%) | Zero-Shot AgentInstruct (%) |
|---|---|---|---|---|
| Overall | Vicuna-13b | 30.6 | 34.4 | 43.9 |
| | Llama-2-7b-chat | 32.1 | 38.3 | 44.6 |
| | Llama-2-13b-chat | 34.3 | 44.5 | 50.2 |
| | Llama-2-70b-chat | 35.1 | 52.4 | 58.3 |
| | GPT-3.5 Turbo | 48.1 | 60.8 | 65.1 |
| | Average over 5 models | 36.1 | 46.1 | 52.4 |
| | Average over 3 models | 37.9 | 49.2 | 55.7 |
| Classification | Vicuna-13b | 41.2 | 42.7 | 50.6 |
| | Llama-2-7b-chat | 37.3 | 39.0 | 46.9 |
| | Llama-2-13b-chat | 39.2 | 46.1 | 52.8 |
| | Llama-2-70b-chat | 44.0 | 51.1 | 61.0 |
| | GPT-3.5 Turbo | 54.0 | 62.6 | 68.1 |
| | Average over 5 models | 43.2 | 48.3 | 55.9 |
| | Average over 3 models | 46.4 | 52.1 | 59.9 |
| Generation | Vicuna-13b | 17.6 | 24.1 | 35.6 |
| | Llama-2-7b-chat | 25.7 | 37.4 | 41.7 |
| | Llama-2-13b-chat | 28.4 | 42.4 | 47.0 |
| | Llama-2-70b-chat | 24.1 | 54.1 | 54.9 |
| | GPT-3.5 Turbo | 40.8 | 58.6 | 61.3 |
| | Average over 5 models | 27.3 | 43.3 | 48.1 |
| | Average over 3 models | 27.5 | 45.6 | 50.6 |
| Reasoning | Vicuna-13b | 26.1 | 29.1 | 45.2 |
| | Llama-2-7b-chat | 26.4 | 35.9 | 48.0 |
| | Llama-2-13b-chat | 28.5 | 46.9 | 54.8 |
| | Llama-2-70b-chat | 26.6 | 62.1 | 69.5 |
| | GPT-3.5 Turbo | 46.9 | 70.9 | 78.7 |
| | Average over 5 models | 30.9 | 49.0 | 59.2 |
| | Average over 3 models | 33.2 | 54.0 | 64.5 |

Table 6: Zero-shot, zero-shot CoT, and zero-shot AgentInstruct results split by category. The table shows the comparison for all five models split by category. The average over three models is based on Vicuna-13b, Llama-2-70b-chat, and GPT 3.5 Turbo.

## C  ADDITIONAL RESULT ANALYSIS

### C.1  ERROR ANALYSIS

**Overall**  We provide more specific errors for each dataset in Table 7, accounting for all instances in the 25 samples of AddSub, IMDB, and NewsQA respectively.

| Dataset | | Error Type | Further Description of Error | Percentage |
|---|---|---|---|---|
| AddSub | Reasoning | Incorrect reasoning | Wrong operation | 24.0 |
| | | Incorrect reasoning | Incorrect step | 20.0 |
| | | Incorrect reasoning | Missed a step | 8.0 |
| | | Not factual | Arithmetic error | 12.0 |
| | Answer | Short answer | Conciseness prompt causes rounding | 24.0 |
| | | Incorrect format | Hallucinated multiple choice | 8.0 |
| | | Ambiguity | Confusing question | 4.0 |
| IMDB | Reasoning | Incorrect reasoning | Misreads sentiment due to language used | 36.0 |
| | Answer | Invalid label | Chooses neutral/mixed | 32.0 |
| | | Ambiguity | Confusing question | 20.0 |
| | | Not factual | Answer before explanation | 12.0 |
| NewsQA | Reasoning | Not factual | Tried to infer answer not in passage | 12.0 |
| | | Incorrect reasoning | Didn't understand question | 8.0 |
| | Answer | Ambiguity | Answer mostly correct | 32.0 |
| | | Ambiguity | No real question to answer | 8.0 |
| | | Ambiguity | Answer seems right but should be no answer | 4.0 |
| | | Incorrect format | Answer seems right but not from continuous stretch | 12.0 |
| | | Incorrect format | Answer in explanation but not provided at the end | 4.0 |
| | | Invalid label | Says there is no answer but not the recognized no answer label | 8.0 |
| | | Invalid label | Empty answer | 4.0 |
| | | Short answer | Conciseness prompt shortens answer too much | 8.0 |

Table 7: Error analysis results for Llama-2-70b-chat on AddSub, IMDB, and NewsQA, each with 25 samples whose final prediction with zero-shot AgentInstruct was incorrect.

Moreover, for each error category, we provide an example in Table 8. Note that we do not include NewsQA examples due to the dataset's restrictive license.

**Summarization Datasets**  For each model, zero-shot AgentInstruct tends to lose on summarization datasets. So, we select 25 random instances from XSUM (with an F1 score less than 1) on Llama-2-70b-chat and find summarization-specific error cases.

In all but one of the instances, the summary extracted using the answer extraction prompt was not the same summary given after the reasoning extraction prompt. It was often kept around the same length but reworded or changed to contain somewhat different information from the first summary. This suggests that for the answer extraction prompt, the LLM still feels the need to write a new summary instead of relying entirely on the one generated during its intermediate reasoning steps.

We identify four summaries given by zero-shot AgentInstruct with incorrect information and three others that are satisfactory but a little misleading, which could be due to the intermediate reasoning steps reinforcing incorrect answers. However, the remaining summaries all seem informative, sometimes more so than the references. In fact, in four of the instances, the reference summary seems to be based on the full article, not the part of the article presented in the implementation from HELM, which poses a problem for evaluation. Additionally, traditional metrics used for summarization (F1, ROUGE-2) are not able to gauge truthfulness when a reference summary is flawed (e.g., a summary hallucinating a part of the article could beat one that is truthful but less similar to a reference).

| Dataset | AddSub |
|---|---|
| Question | Question: Jason has 18 books and he has read 9 of them . Mary has 42 books . How many books do they have together ? |
| CoT Reasoning | 1. Understand the context: The prompt is asking about the total number of books that Jason and Mary have. 2. Identify the operation: The operation is addition. 3. Perform the operation: Jason has 18 books and has read 9 of them, so the number of books he has left is 18 - 9 = 9. Mary has 42 books. Therefore, the total number of books they have together is 9 + 42 = 51. 4. Answer the question: They have 51 books together. 
 Correct answer: 51 books. 
 Therefore, the answer to the task is below. Give the answer in the shortest form possible that will still be correct. Answer: 51. Answer: 51 |
| Ground Truth | 60 |
| Error Type | Incorrect reasoning (incorrect step): Jason reading 9 books does not mean he no longer has them. |

| Dataset | IMDB |
|---|---|
| Question | Passage: Susan Sarandon and Natalie Portman play Adele and Ann August, a single mother who's the working definition of the word "dreamer" and her solemn, pragmatic daughter. Adele, wanting to make a fresh start in a big city, moves them from Wisconsin to California.

Decent, if not exceptional or particularly memorable, with some serious and sad moments. It pushes some emotional buttons, that's for sure. Best thing it offers is a solid cast, and Sarandon and Portman are quite good in the leads - even if their characters are one-dimensional at times.

Could be better, could be worse.

I gave this one seven out of ten. |
| CoT Reasoning | The passage provided is a review of the movie "Anywhere But Here" starring Susan Sarandon and Natalie Portman. The reviewer mentions that the movie is "decent" and "solid" but also states that it's "not exceptional or particularly memorable." They also mention that it has some "serious and sad moments" and that the cast, particularly Sarandon and Portman, are "quite good" but their characters are "one-dimensional at times." 
 Based on the language used in the review, it's clear that the reviewer has mixed feelings about the movie. They use phrases like "decent," "solid," and "could be better, could be worse" to indicate that the movie is not outstanding but still enjoyable. The reviewer also uses phrases like "pushes some emotional buttons" and "solid cast" to indicate that the movie has some positive aspects. 
 However, the reviewer also uses phrases like "not exceptional or particularly memorable" and "one-dimensional at times" to indicate that the movie is not perfect and has some flaws. 
 Therefore, based on the tone and language used in the review, it's clear that the sentiment of the review is neutral or mixed. The reviewer does not express any strong positive or negative emotions towards the movie. 
 Answer: Neutral 
 Therefore, the correct label among 'Negative', 'Positive' (just the label) to the original task is below. Sentiment: Neutral |
| Ground Truth | Positive |
| Error Type | Invalid label (chooses neutral/mixed): Decides to label neutral instead of negative or positive. |

| Dataset | AddSub |
|---|---|
| Question | Question: Mandy made an apple pie . She used 0.6666666666666666 tablespoon of cinnamon and 0.5 tablespoon of nutmeg . How much more cinnamon than nutmeg did Mandy use ? |
| CoT Reasoning | 1. Understand the context: The context of the problem is that Mandy made an apple pie and used certain amounts of cinnamon and nutmeg. 2. Identify the operation: The operation in this problem is subtraction. The question asks how much more cinnamon than nutmeg did Mandy use, which implies that we need to find the difference between the amount of cinnamon and nutmeg used. 3. Perform the operation: We are given that Mandy used 0.6666666666666666 tablespoon of cinnamon and 0.5 tablespoon of nutmeg. To find the difference, we can subtract the amount of nutmeg from the amount of cinnamon: 
 0.6666666666666666 - 0.5 = 0.1666666666666666 
 4. Answer the question: The question asks how much more cinnamon than nutmeg did Mandy use? The answer to this question is the result of the subtraction operation we performed earlier: 0.1666666666666666. 
 Therefore, the answer to the question is 0.1666666666666666 tablespoon of cinnamon more than nutmeg. 
 Correct answer: 0.1666666666666666 tablespoon. 
 Therefore, the answer to the task is below. Give the answer in the shortest form possible that will still be correct. Answer: 0.17 |
| Ground Truth | 0.16666666666666666 |

| Error Type | Short answer (conciseness prompt causes rounding): Causes too much rounding. |
|---|---|
| Dataset | AddSub |
| Question | Question: Jason joined his school 's band . He bought a flute for $ 142.46 , a music tool for $ 8.89 , and a song book for $ 7 . How much did Jason spend at the music store ? |
| CoT Reasoning | 1. Understand the context: The context of the problem is that Jason joined his school's band and purchased several items at a music store. |
|  | 2. Identify the operation: The operation in this problem is addition. The verb "spent" implies that Jason spent a certain amount of money on the items he purchased. |
|  | 3. Perform the operation: To find out how much Jason spent at the music store, we need to add the cost of the flute, music tool, and song book. The cost of the flute is $142.46, the cost of the music tool is $8.89, and the cost of the song book is $7. Therefore, the total amount Jason spent at the music store is: |
|  | $142.46 + $8.89 + $7 = $158.22 |
|  | 4. Answer the question: The question asks, "How much did Jason spend at the music store?" The answer to this question is $158.22. |
|  | Therefore, the correct answer to the task is $158.22. |
|  | Therefore, the answer to the task is below. Give the answer in the shortest form possible that will still be correct. Answer: $158.22 |
| Ground Truth | 158.35 |
| Error Type | Not factual (arithmetic error): Arithmetic done incorrectly. |

| Dataset | AddSub |
|---|---|
| Question | Dale 's Vegetarian Restaurant bought 2.8333333333333335 pounds of green peppers and 2.8333333333333335 pounds of red peppers . How many pounds of peppers did Dale 's Vegetarian Restaurant buy in all ? |
| CoT Reasoning | 1. Understand the context: The context of the problem is that Dale's Vegetarian Restaurant bought certain amounts of green and red peppers. 2. Identify the operation: The operation in this problem is addition. The verb "bought" implies that the restaurant purchased both green and red peppers, and we need to find the total amount of peppers bought. 3. Perform the operation: We need to add 2.8333333333333335 pounds of green peppers and 2.8333333333333335 pounds of red peppers. |
|  | 2.8333333333333335 + 2.8333333333333335 = 5.666666666666667 |
|  | 4. Answer the question: The question asks, "How many pounds of peppers did Dale's Vegetarian Restaurant buy in all?" |
|  | The answer is 5.666666666666667 pounds of peppers. |
|  | Therefore, the correct answer is (D) 5.67 pounds. |
|  | Therefore, the answer to the task is below. Give the answer in the shortest form possible that will still be correct. Answer: 5.67 pounds. |
| Ground Truth | 5.666666666666667 |
| Error Type | Incorrect format (hallucinated multiple choice): There were no multiple choice letters anywhere in the problem, but the model thought there was and its answer was rounded too much. |

| Dataset | IMDB |
|---|---|
| Question | Passage: I love old "monster movies" for the pure camp value. This one does not disappoint if you find that sort of thing amusing.

The acting is pure 1950s stilted crap. You do get used to it when you've seen enough of these... the dialogue is very silly and ultimately forgettable. You're just there for the giant bird.

The "science" in this movie is hilarious. A monstrous otherworldly avian that can manipulate anti-matter... intent on wreaking havoc and eating people... an alien who has come to Earth to nest. Some of the best scenes in the movie (that don't have the bird in them) include the "scientists" explaining what is going on.

A lot has been said about the ridiculous bird marionette. It looks like a new baby bird... bald and ugly with ruffled feathers. It even caws like a hungry, angry fledgling. However, the bird scared the crap out of my three year old, who had big scary bird nightmares for the evening. It's a bit creepy.

I was very sad about the scene where they shoot the egg. (Somehow the egg is not protected by an antimatter shield.) But I'm a bird lover, what can I say? |

| | |
|---|---|
| CoT Reasoning | The passage provided is a review of a movie, specifically a "monster movie" from the 1950s. The reviewer mentions that they enjoy old monster movies for their camp value, indicating that they find them amusing. They mention that the acting in the movie is stilted and the dialogue is silly and forgettable, but they are still able to enjoy the movie for its entertainment value. The reviewer also mentions that the "science" in the movie is hilarious, specifically the concept of a monstrous otherworldly avian that can manipulate anti-matter and has come to Earth to nest. They seem to find the idea of this bird, and the scientists' Reasonings of it, amusing. The reviewer also mentions that the bird marionette used in the movie looks like a new baby bird and sounds like a hungry, angry fledgling, which scared their three-year-old child. They seem to find the bird's appearance and behavior amusing, but also acknowledge that it may be creepy to some people. Based on the reviewer's use of language and tone, it is clear that they have a positive sentiment towards the movie. They use words and phrases like "love," "amusing," "pure camp value," and "entertainment value" to express their enjoyment of the movie. They also use a playful and lighthearted tone when discussing the movie's science and the bird marionette. Therefore, the correct answer is "Positive." Answer: Positive Therefore, the correct label among 'Negative', 'Positive' (just the label) to the original task is below. Sentiment: Positive |
| Ground Truth | Negative |
| Error Type | Ambiguity (confusing question): The reviewer seems to enjoy the movie as they watch it knowing it's supposed to be bad. So, it's not clear whether they are rating based on the quality (which would be negative) or their enjoyment of it (which would be positive). The latter certainly seems favored in their review. |

Table 8: Error analysis examples for each error type in the AddSub and IMDB datasets.

| Dataset | Issue |
|---|---|
| AddSub | Not found |
| AQuA | Incorrect dataset found |
| Coin Flip | Incorrect dataset found |
| Last Letter Concatenation | Incorrect dataset found |
| SingleEq | Not found |

Table 9: Agent instruction issues where our QA API was not able to retrieve any relevant information from the sources from the web.

**Agent Instruction Issues** In Table 9, we list all the main datasets where the QA API was not able to gain any relevant information about the task.

## C.2 MANUAL PROMPT SENSITIVITY

In Table 2, we tested various manual prompts on Llama-2-70b-chat on AddSub. In Table 10, we focus on formatting, specifically whether the instruction should appear before or after the instance. Again, the difference is minor suggesting that the methodology behind zero-shot AgentInstruct is robust.

| CoT Reasoning | Prompt | Quasi-Exact Match (%) |
|---|---|---|
| Format | **Instruction -> Instance** | 79.5 |
| | Instance -> Instruction | **80.0** |

Table 10: Prompt sensitivity analysis on Llama-2-70b-chat on AddSub.

## C.3 ADDITIONAL DESCRIPTIONS OF ABLATION STUDY

- w/o Agent Instructions: We compare the zero-shot AgentInstruct methodology to zero-shot CoT. This setting is exactly zero-shot CoT on Llama-2-70b-chat.

- w/o Input Examples: We remove the examples from the input to the agent. Typically, the agent is given 5 input-only examples to reference when generating the instructions. These examples do not include the ground truth answers. In this setting, we have the agent generate instructions without providing example inputs to the agent.

- w/o Labels: We remove the description of the labels from the input to the agent. Specifically, this is applicable for classification tasks such as IMDB, for example, where the expected outputs are either 'True' or 'False'. For multiple-choice and open-ended generation tasks, we provide a simple description of 'multiple choice' or 'generation' instead of providing a list of output labels. With this setting, we have the agent generate instructions without providing any information or description of the output labels for any task type. An example is given in Figure 1 and Figure 2.

- w/o GPT-4: We use GPT-3.5 Turbo to generate the instructions instead of GPT-4.

## D  DATASET EXAMPLES

*WARNING: this section contains examples of text that may be considered unsafe, offensive, or upsetting.*

Complete running examples including agent instructions and CoT reasoning paths for each dataset are presented in Figure 18 to Figure 69.

```
You will be provided instructions for completing a task followed by a task to complete.
Instructions:
1. Understand the context: The dataset involves mathematical operations like addition and subtraction in a
contextual setting. The operations are performed on 'objects' in different 'locations'.
2. Identify the operation: The operation could be addition or subtraction. This can be identified from the
'verb' in the prompt. For example, if the verb is 'add', the operation is addition. If the verb is 'remove',
the operation is subtraction.
3. Perform the operation: Once the operation is identified, perform it on the 'number' of 'objects'. For
example, if the operation is addition and the number of objects is 2 and 3, the result is 5. If the operation
is subtraction and the number of objects is 5 and 3, the result is 2.
4. Answer the question: The question at the end of the prompt asks 'how many [object] [verb] [subject]
[preposition] [location]?'. Use the result from the operation to answer this question.
5. Topics to know: Basic understanding of addition and subtraction, understanding of prepositions and their
use in sentences, ability to understand context from a sentence.
6. Steps to answer the question:
a. Understand the context from the prompt.
b. Identify the operation from the verb.
c. Perform the operation on the number of objects.
d. Use the result to answer the question at the end of the prompt.
Remember, the key is to understand the context, identify the operation, perform it, and use the result to
answer the question.
Question: Joan found 70 seashells on the beach . she gave Sam some of her seashells . She has 27 seashell .
How many seashells did she give to Sam ?
Follow the instructions to generate an explanation that reasons towards the correct answer to the task above.
End the explanation with the correct answer. [/INST] Explanation:
1. Understand the context: The context of the problem is that Joan found 70 seashells on the beach and gave
some of them to Sam. The problem is asking for the number of seashells that Joan gave to Sam.
2. Identify the operation: The operation in this problem is subtraction. The verb "gave" implies that Joan
gave some of her seashells to Sam, which means that the number of seashells she has now is less than the number
of seashells she found.
3. Perform the operation: To perform the operation, we need to find out how many seashells Joan has now and
then subtract that number from the number of seashells she found. The problem states that Joan has 27 seashells
now. If we subtract 27 from 70, we get 43.
4. Answer the question: The question asks, "How many seashells did she give to Sam?" The answer to this question
is 43 seashells.
Therefore, the correct answer is 43 seashells.
Therefore, the answer to the task is below. Give the answer in the shortest form possible that will still be
correct.
Answer: 43
```

Figure 18: A complete running example of AddSub.

```
You will be provided instructions for completing a task followed by a task to complete.
Instructions:
1. Understand the structure of the dataset: The AQUA dataset contains question-and-answer pairs related to
algebra. The questions are generated based on paintings and comments from an existing art knowledge dataset.
2. Topics to know: To answer questions from this dataset, you should have a good understanding of algebra.
Additionally, some knowledge of art and how it can be related to algebraic concepts may also be helpful.
3. Steps to answer the questions:
- Read the question carefully and identify the algebraic concept it is asking about.
- Consider the options provided. Each option will represent a different potential answer to the question.
- Use your knowledge of algebra to determine which option is the correct answer. Remember, the questions are
based on paintings and comments from an art knowledge dataset, so consider how these elements might relate to
the algebraic concept in question.
- Once you have determined the correct answer, write it down in the format specified in the prompt template.
4. Tips for answering the questions:
- Don't rush. Take your time to understand the question and the options provided.
- If you're unsure about an answer, try to eliminate the options that you know are incorrect. This can help
narrow down the possibilities and make it easier to choose the correct answer.
- Remember, the questions are based on art knowledge as well as algebra. Try to think creatively about how
these two areas might intersect.
Question: Three birds are flying at a fast rate of 900 kilometers per hour. What is their speed in miles per
minute? [1km = 0.6 miles]
A. 32400
B. 6000
C. 600
D. 60000
E. 10
Follow the instructions to generate an explanation that reasons towards the correct answer to the task above.
End the explanation with the correct answer. [/INST] Explanation:
First, let's convert 900 kilometers per hour to miles per hour. We know that 1 kilometer is equal to 0.6 miles,
so we can convert 900 kilometers per hour to miles per hour as follows:
900 km/h x 0.6 miles/km = 540 miles/h
Now, we need to find the speed of the birds in miles per minute. To do this, we can divide the speed in miles
per hour by 60:
540 miles/h ÷ 60 = 9 miles/min
Therefore, the correct answer is C. 600.
Answer: C. 600
Therefore, the correct multiple choice label (just the letter) to the task is below.
Answer: C
```

Figure 19: A complete running example of AQuA.

```
You will be provided instructions for completing a task followed by a task to complete.
Instructions:
1. Understand the question: The question is a naturally occurring, factual question that was asked on Google.
It is unambiguous and requires a "yes" or "no" answer. Make sure you fully understand what the question is
asking.
2. Read the passage: The passage is taken from a Wikipedia page that was one of the top five results for the
question on Google. It contains the information needed to answer the question. Read the passage carefully and
extract the relevant information.
3. Answer the question: Based on the information in the passage, answer the question with either "yes" or "no".
Make sure your answer is supported by the information in the passage.
4. Check your work: Review the question, passage, and your answer to ensure they align. The question should be
answered accurately based on the information in the passage.
Tips:
- The questions in this dataset are factual and require knowledge of a wide range of topics. Brush up on general
knowledge and be prepared to learn new information from the passages.
- The passages are taken from Wikipedia, so they are written in a clear, informative style. Use this to your
advantage when extracting information.
- The answers are either "yes" or "no", but the process of getting to the answer requires understanding and
interpreting the passage. Take your time to understand the passage before answering.
- Always double-check your work to ensure accuracy.
Passage: Elmendorf Air Force Base (IATA: EDF, ICAO: PAED, FAA LID: EDF) is a United States military facility in
Anchorage, the largest city in Alaska. Originally known as Elmendorf Field, it became Elmendorf Air Force Base
after World War II, and in 2010 it merged with nearby Fort Richardson to form Joint Base Elmendorf-Richardson.
Question: Is there an air force base in anchorage alaska?
Follow the instructions to generate an explanation that reasons towards the correct answer to the task above.
End the explanation with the correct answer. [/INST] Explanation:
The question asks whether there is an air force base in Anchorage, Alaska. To answer this question, we need
to look at the provided passage.
The passage states that Elmendorf Air Force Base is a United States military facility located in Anchorage,
Alaska. It was originally known as Elmendorf Field and later became Elmendorf Air Force Base after World War
II. In 2010, it merged with nearby Fort Richardson to form Joint Base Elmendorf-Richardson.
Based on the information in the passage, it is clear that there is indeed an air force base in Anchorage,
Alaska. The passage explicitly states that Elmendorf Air Force Base is located in Anchorage, and its history
is described in the passage.
Therefore, the answer to the question is "yes."
Answer: Yes.
Therefore, the correct label among 'No', 'Yes' (just the label) to the original task is below.
Answer: Yes.
```

Figure 20: A complete running example of BoolQ.

```
You will be provided instructions for completing a task followed by a task to complete.
Instructions:
1. Understand the context: The civil_comments dataset consists of public comments from various English-language
news sites. These comments were made between 2015 and 2017.
2. Consider the metadata: Each comment comes with associated metadata such as article IDs and timestamps. Use
this information to help understand the context of the comment.
3. Analyze the comment: Each comment in the dataset has been labeled for toxicity and identity mentions. Use
these labels to help determine the nature of the comment.
4. Be aware of potential hate speech: The dataset is often used for tasks like Hate Speech Detection and Toxic
Comment Classification. Be aware that some comments may contain hate speech or toxic language.
5. Answer the question: Based on your understanding of the comment and its context, answer the question. Your
answer should be either 'True' or 'False'.
6. Be objective: Your answer should be based on the information in the comment and its associated metadata,
not on personal opinions or biases.
7. Be concise: Your answer should be clear and to the point. Avoid unnecessary elaboration.
8. Be respectful: Even if the comment contains hate speech or toxic language, your response should always be
respectful and professional.
Passage: Exactly! It seems every time a black youth is killed by police, the grieving parent is the mom. Moms
get recognized by black athletes but seldom is there a dad mentioned.
Question: Is the passage above toxic?
Follow the instructions to generate an explanation that reasons towards the correct answer to the task above.
End the explanation with the correct answer. [/INST] Explanation:
The passage above mentions the killing of black youth by police and the grieving parents, specifically mentioning
mothers being recognized by black athletes but not fathers. This could be interpreted as a commentary on the
lack of recognition or support for black fathers who have lost their children to police violence. However,
the tone of the passage is not aggressive or hostile, and it does not contain any hate speech or derogatory
language.
Based on the context and content of the passage, it can be determined that the comment is not toxic. The
comment does not contain any hate speech or toxic language, and it raises a valid point about the lack of
recognition for black fathers who have lost their children to police violence.
Therefore, the answer to the question is False. The passage is not toxic.
Therefore, the correct label among {'False', 'True'} (just the label) to the original task is below.
Answer: False
```

Figure 21: A complete running example of CivilComments (All).

You will be provided instructions for completing a task followed by a task to complete.
Instructions:
1. Understand the context: The dataset contains public comments from various news sites. These comments have been labeled for toxicity, obscenity, threat, insult, identity attack, and sexual explicitness. Each label is a value between 0 and 1, indicating the fraction of annotators that assigned these attributes to the comment text.
2. Consider the demographic: Although the dataset does not explicitly mention the demographic 'black', it seems that the comments may have been categorized based on whether they were made by individuals identified as black or non-black. Keep this in mind when answering questions.
3. Analyze the passage and question: Read the passage and question carefully. The passage will provide a general statement or argument, and the question will ask something related to that statement or argument. Your answer should be based on the information provided in the passage and question.
4. Answer the question: Based on your analysis of the passage and question, provide an answer. The possible outputs are 'True' or 'False'. Make sure your answer is supported by the information in the passage and question.
5. Be mindful of sensitive topics: Given the nature of the dataset, the passage and question may involve sensitive topics related to race and discrimination. Be respectful and considerate in your responses.
6. Review your answer: Before submitting your answer, review it to ensure it is accurate, clear, and respectful.
Passage: Kayden, religion has nothing to do with this. Hinduphobia- no it's cult and self seeking power hungry crooks like your man Butler that jump on the religious wagon who have no qualms about lying and cheating their way up the political ladder. In Tulsi Gabbard's case, her guru has taken advantage of her cult beliefs she was born into and abused the fact that she is beholden to him. As he says when he initiates his disciples, you are now mine and I will instruct you how you can serve ME. Fanaticism not religion.
Cry yourself silly with all your talk on religious bigotry, homophobic backed support, Hillary Clinton money, and whatever else you can lash out with.
How about all the other religions, I don't see Tulsi and Butler supporting them. In fact she down right degrades the Muslims. Butler has always spoken very unkindly of Gays, Blacks, Muslims Christians and Jews.
Beware Hawaii, Tulsi and anyone who adheres to him, has no place in our democratic system.
Question: Is the passage above toxic?
Follow the instructions to generate an explanation that reasons towards the correct answer to the task above. End the explanation with the correct answer. [/INST] Explanation:
The passage above contains several elements that suggest it may be toxic. Firstly, the author of the passage uses strong language and personal attacks against individuals, such as "cult and self-seeking power-hungry crooks" and "fanaticism not religion." This type of language can be seen as inflammatory and disrespectful, and may be intended to provoke or intimidate others. Additionally, the author makes broad generalizations about certain groups, such as "Muslims, Christians, and Jews," which may be perceived as discriminatory or hateful. Furthermore, the author implies that anyone who supports Tulsi Gabbard or her guru is not suitable for a democratic system, which may be seen as undemocratic or intolerant. This statement could be interpreted as an attempt to silence or dismiss opposing viewpoints, which is not conducive to a healthy democratic discourse. Therefore, based on the language, tone, and content of the passage, it can be concluded that the passage is toxic.
Answer: True
Therefore, the correct label among {'False', 'True'} (just the label) to the original task is below.
Answer: True

Figure 22: A complete running example of CivilComments (Black).

```
You will be provided instructions for completing a task followed by a task to complete.
Instructions:
1. Understand the context: The civil_comments dataset contains public comments from various news sites. These
comments were created between 2015 and 2017 and appeared on approximately 50 English-language news sites across
the world. The dataset includes the text of the comments, some associated metadata, and labels for toxicity
and identity mentions.
2. Focus on the content: The dataset does not provide specific demographic information about the commenters.
Therefore, your analysis should focus on the content of the comments and their context, rather than attempting
to infer demographic information.
3. Analyze the toxicity: The dataset includes labels for toxicity. Use these labels to help determine the tone
and intent of the comments.
4. Be aware of the limitations: The dataset does not include user identities or demographic information about
the commenters. Therefore, be cautious about making assumptions about the individuals who made the comments.
5. Answer the question: Based on your understanding of the comment and its context, answer the question related
to the passage. Your answer should be either 'True' or 'False'. Make sure your answer is supported by the
information available in the comment and its context.
6. Explain your reasoning: After providing your answer, explain why you chose that answer. Your explanation
should reference specific parts of the comment and how they led you to your conclusion.
Passage: Keep God out of politics...I'd rather expose the politicians for creating phony front groups designed
to mislead Catholics.
Question: Is the passage above toxic?
Follow the instructions to generate an explanation that reasons towards the correct answer to the task above.
End the explanation with the correct answer. [/INST] Explanation:
The passage above expresses a negative sentiment towards politicians and their involvement in religion. The
author of the comment advocates for keeping God out of politics and instead, exposing politicians who create
fake front groups to mislead Catholics. The tone of the comment is critical and disapproving, which suggests
that it may be toxic.
However, upon closer examination, it's clear that the author's intention is not to incite hatred or violence,
but rather to express their opinion on the separation of church and state. The comment does not contain any
personal attacks, racist, sexist, or homophobic language, or other forms of toxic content.
Therefore, based on the context and content of the comment, the answer to the question is False. The passage
is not toxic.
Reasoning:
* The comment expresses a negative sentiment towards politicians, but it does not contain any toxic language
or personal attacks.
* The author's intention is to advocate for the separation of church and state, which is a legitimate opinion
to hold.
* The comment does not incite hatred or violence, and it does not contain any discriminatory language.
Answer: False. The passage is not toxic.
Therefore, the correct label among {'False', 'True'} (just the label) to the original task is below.
Answer: False
```

Figure 23: A complete running example of CivilComments (Christian).

```
You will be provided instructions for completing a task followed by a task to complete.
Instructions:
1. Understand the nature of the civil_comments dataset: This dataset contains public comments from news sites,
with labels for toxicity and identity mentions. The comments were created from 2015 - 2017 and appeared on
approximately 50 English-language news sites across the world.
2. Know what constitutes toxicity in comments: Toxicity can include insults, threats, obscene language, identity
attacks, and sexually explicit content. However, it's important to consider the context and nuances of the
comment.
3. Consider cultural and societal norms: When evaluating a comment for toxicity, consider cultural and societal
norms related to the demographic 'female'. Remember that what may be considered toxic in one culture or society
may not be in another.
4. Apply this knowledge to the prompt: Use the information you've gathered to evaluate the passage in the
prompt. Is it toxic? Consider the language used, the context, and any potential harm or offense it may cause,
especially to the demographic 'female'.
5. Formulate your answer: Based on your evaluation, answer 'True' if the passage is toxic, and 'False' if it
is not. Be prepared to provide reasoning for your answer if necessary.
Passage: Actually, we are a secular organization. Ninety-nine percent of the Church's population is secular,
and it's founder, Jesus Himself, was a secular. The word "secular" has gotten a bad rap. "Secular" means in
the world, in the world God created and and looked at as "very good". The priesthood which exists is the
Royal Priesthood we all share by virtue of our baptism. Some of us (both male and female) may be called to
the ordained priesthood of servant-leadership. That call comes from God Who does not distinguish between male
and female. Just as the world is constantly evolving in its journey to Christ-centered completion, so too our
understanding of servant-leadership will continue to unfold until we finally realize that "in Christ there is
neither male nor female" and "all are one in Christ Jesus".
Question: Is the passage above toxic?
Follow the instructions to generate an explanation that reasons towards the correct answer to the task above.
End the explanation with the correct answer. [/INST] Explanation:
The passage provided does not contain any explicit language, threats, or personal attacks, which are common
indicators of toxicity. However, it's important to consider the context and cultural norms when evaluating the
passage.
The passage mentions the founder of the Church, Jesus, and the concept of the "Royal Priesthood" that is shared
by all members of the Church, regardless of gender. It also mentions the ordained priesthood and how God does
not distinguish between male and female. This language could be seen as inclusive and respectful towards women,
promoting the idea of equality among all members of the Church.
Furthermore, the passage also highlights the idea of continuous evolution towards a Christ-centered completion,
which could be interpreted as a call for progress and improvement towards a more inclusive and equitable
society.
Therefore, based on the language used, the context, and the message of inclusivity and equality, it can be
concluded that the passage is not toxic.
Answer: False.
Therefore, the correct label among {'False', 'True'} (just the label) to the original task is below.
Answer: False
```

Figure 24: A complete running example of CivilComments (Female).

```
You will be provided instructions for completing a task followed by a task to complete.
Instructions:
1. Understand the context: The comment is from the civil_comments dataset, which contains public comments from
various news sites. The comments cover a wide range of topics, and the demographic in question is LGBTQ.
2. Analyze the comment: Read the comment carefully and understand its content. Look for any indications of
toxicity, such as insults, threats, obscene language, identity attacks, or sexually explicit content.
3. Consider the demographic: Keep in mind that the demographic in question is LGBTQ. Be aware of issues and
sensitivities related to this demographic.
4. Determine toxicity: Based on your analysis of the comment and your understanding of the demographic,
determine whether the comment is toxic. Remember that a comment is considered toxic if it is disrespectful or
harmful towards others, especially towards the LGBTQ demographic in this case.
5. Answer the question: If the comment is toxic, answer 'True'. If the comment is not toxic, answer 'False'.
Be sure to base your answer on the content of the comment and not on your personal opinions or biases.
Tips:
- Be familiar with LGBTQ issues and sensitivities.
- Be aware of the different forms that toxicity can take, including insults, threats, obscene language, identity
attacks, and sexually explicit content.
- Be objective and unbiased in your analysis. Your goal is to determine whether the comment is toxic, not
whether you agree with it or not.
- Keep in mind that a comment can be considered toxic even if it does not contain explicit insults or threats.
Any comment that is disrespectful or harmful towards others, especially towards the LGBTQ demographic, can be
considered toxic.
Passage: Should there be a ban on men who are so "powerful" that they can just walk up and grab women by their
$#$$%ies? Trump has thrown out another "shiny object" to distract folks from the issues with his administration
and, unfortunately, from focusing on the many important topics we, as Americans, need to address. To name
a few: health care, public education, unemployment and under-employment, homelessness, alcohol abuse, opioid
abuse, mental health issues, veteran services...especially regarding PTSD, our crumbling infrastructure, income
disparity, ...I'm guessing you could add to this list.
But no, Trump tweets out that transgendered individuals can't be in the military......and this is the man who
is the "leader" of our country? Were it not so tragic, it would be funny. Gary Crum
Question: Is the passage above toxic?
Follow the instructions to generate an explanation that reasons towards the correct answer to the task above.
End the explanation with the correct answer. [/INST] Explanation:
The passage above is toxic. The author, Gary Crum, uses derogatory language towards a specific group of
people, specifically the transgender community. The use of the term "transgendered" is considered offensive
and disrespectful, as it implies that being transgender is a condition or a state of being, rather than a
natural part of a person's identity. Additionally, the author implies that transgender individuals are not
capable of serving in the military, which is a harmful and inaccurate stereotype.
Furthermore, the author's comment about President Trump's administration and his use of "shiny objects" to
distract from important issues is not relevant to the topic of the comment and serves only to further the
author's political agenda. This irrelevant information can be seen as a form of obfuscation, intended to
distract from the harmful and toxic language used in the comment.
Therefore, based on the use of derogatory language and harmful stereotypes, as well as the irrelevant and
distracting political commentary, the passage above is toxic.
Answer: True
Therefore, the correct label among {'False', 'True'} (just the label) to the original task is below.
Answer: True
```

Figure 25: A complete running example of CivilComments (LGBTQ).

```
You will be provided instructions for completing a task followed by a task to complete.
Instructions:
1. Understand the context: The civil_comments dataset is a collection of public comments from various news
sites. These comments have been labeled for toxicity and identity mentions. The dataset does not contain
specific demographic information about the commenters or the content of the comments.
2. Analyze the passage: Read the passage carefully and understand the general statement or quote. Pay attention
to any additional information or commentary provided.
3. Answer the question: Based on the passage, answer the question related to the passage. Your answer should
be either 'True' or 'False'. Do not attempt to infer demographic information, as this is not provided in the
dataset.
4. Focus on the content: Your primary focus should be on the content of the comments and their associated
toxicity labels.
5. Be objective: Avoid making assumptions or inferences that are not supported by the passage. Stick to the
facts presented in the passage.
6. Be concise: Your answer should be concise and to the point. Do not include unnecessary information or
commentary.
Remember, your goal is to provide an accurate and unbiased answer based on the information provided in the
passage.
Passage: As the 2016 USCCB report done by an outside group shows more than 80% were same sex (male) assaults
and 90% were post pubescent.
Those numbers are staggeringly disproportionate and undercut your point.
Question: Is the passage above toxic?
Follow the instructions to generate an explanation that reasons towards the correct answer to the task above.
End the explanation with the correct answer. [/INST] Explanation:
The passage provided is a comment made in response to a report about sexual assaults. The comment states that
the majority of sexual assaults are committed by men and that the majority of victims are post-pubescent.
The comment does not contain any toxic language or personal attacks. It does not promote or incite violence or
discrimination. It does not contain any hate speech or derogatory language towards any gender, race, religion,
or group of people.
Therefore, based on the information provided in the passage, the answer to the question is False. The passage
is not toxic.
Answer: False.
Therefore, the correct label among {'False', 'True'} (just the label) to the original task is below.
Answer: False
```

Figure 26: A complete running example of CivilComments (Male).

```
You will be provided instructions for completing a task followed by a task to complete.
Instructions:
1. Understand the context: The civil_comments dataset contains public comments from various news sites worldwide.
These comments cover a wide range of topics including current events, politics, and social issues. Your responses
should reflect an understanding of these contexts.
2. Know the demographic: This task focuses on the 'muslim' demographic. Be aware of the specific cultural,
religious, and social aspects related to this demographic.
3. Understand the labels: The comments in the dataset have been labeled for toxicity and identity mentions.
Familiarize yourself with these labels and what they represent.
4. Be objective: Your responses should be objective and unbiased. Avoid making assumptions or generalizations
about the 'muslim' demographic or any other demographic mentioned in the comments.
5. Be accurate: Your responses should accurately reflect the content of the comments. Do not add or omit
information.
6. Be respectful: Respect the sensitivity of the topics discussed in the comments. Avoid using offensive or
inappropriate language.
7. Be concise: Your responses should be concise and to the point. Avoid unnecessary elaboration or speculation.
8. Be aware of the possible outputs: The possible outputs for this task are 'True' and 'False'. Make sure your
responses align with these options.
Remember, the goal is to understand and improve civility in online conversations. Your responses should
contribute to this goal.
Passage: This woman, Chrystia Freeland, went on US television last year and stated that it was wrong to criticize
certain segments of the Muslim population for honour killings, female genital mutilation, and persecution of
gays.
She is a far-left, bat*** crazy, cultural marxist.
Question: Is the passage above toxic?
Follow the instructions to generate an explanation that reasons towards the correct answer to the task above.
End the explanation with the correct answer. [/INST] Explanation:
The passage above contains several elements that suggest it is toxic. Firstly, the use of the term "bat***
crazy" to describe Chrystia Freeland is a form of name-calling and personal attack, which is a hallmark of
toxic language. Additionally, the phrase "cultural marxist" is often used in a derogatory manner to describe
individuals who are perceived to be promoting a liberal or progressive agenda, and its use in this context
suggests that the author is attempting to discredit Freeland's views without providing any substance to their
argument.
Furthermore, the passage implies that Freeland's comments were wrong and that she is condoning or excusing
certain practices within the Muslim community, which could be perceived as Islamophobic. The use of the phrase
"certain segments of the Muslim population" also implies that the author is generalizing and stereotyping an
entire religious group, which is a form of prejudice.
Therefore, based on the above analysis, the passage is toxic.
Answer: True
Therefore, the correct label among {'False', 'True'} (just the label) to the original task is below.
Answer: True
```

Figure 27: A complete running example of CivilComments (Muslim).

```
You will be provided instructions for completing a task followed by a task to complete.
Instructions:
1. Read the passage carefully. The passage will contain a general statement or description of a topic or
situation, followed by additional information or details about the topic or situation.
2. Understand the context of the passage. The passage is likely to be a public comment from an English-language
news site. It may contain varying levels of toxicity, obscenity, threat, insult, identity attack, and sexual
explicitness.
3. Answer the question based on the passage. The question will be related to the passage. Your answer should
be either 'True' or 'False'.
4. Be aware that the dataset does not contain specific demographic information or labels related to
"other_religions". Therefore, your answer should not be based on assumptions about the demographic of the
commenter.
5. Use your understanding of the topic and the context of the comment to guide your answer. Consider the level
of toxicity, obscenity, threat, insult, identity attack, and sexual explicitness in the comment.
6. Remember that your goal is to provide an accurate and unbiased answer based on the information in the
passage.
Passage: Hindus are the majority in Kerala - more than half of the population. Then come Muslims and Christians.
The current state government is a coalition of leftist parties, with the Chief Minister coming from the
Communist Party of India (Marxist).
Question: Is the passage above toxic?
Follow the instructions to generate an explanation that reasons towards the correct answer to the task above.
End the explanation with the correct answer. [/INST] Explanation:
The passage provides a description of the religious demographics of Kerala, stating that Hindus are the majority,
followed by Muslims and Christians. It also mentions the current state government, which is a coalition of
leftist parties with the Chief Minister coming from the Communist Party of India (Marxist).
Based on the information provided in the passage, it does not contain any toxic language or content. The
passage simply presents factual information about the religious demographics and political landscape of Kerala.
Therefore, the answer to the question is False. The passage is not toxic.
Answer: False.
Therefore, the correct label among {'False', 'True'} (just the label) to the original task is below.
Answer: False
```

Figure 28: A complete running example of CivilComments (Other Religions).

```
You will be provided instructions for completing a task followed by a task to complete.
Instructions:
1. Understand the context: The civil_comments dataset contains public comments from various news sites, along
with labels indicating various forms of toxicity and identity mentions.
2. Analyze the passage: The passage will provide a general statement or argument. This could be about a specific
comment, a trend in the comments, or a general observation about the dataset.
3. Understand the question: The question will ask for a response related to the passage. This could be a
factual question about the content of the comments, a question about the labels, or a question about trends
or patterns in the dataset.
4. Use your knowledge: Use your understanding of the civil_comments dataset to answer the question. This could
involve interpreting the content of the comments, the labels, or the trends in the dataset.
5. Consider the demographic: The demographic 'white' may be represented through mentions in the comments.
Consider this when interpreting the comments and answering the question.
6. Formulate your answer: Your answer should be either 'True' or 'False'. Ensure your answer is supported by
the information in the passage and your understanding of the civil_comments dataset.
7. Explain your reasoning: Provide a clear and concise explanation for your answer. This should reference the
information in the passage and your understanding of the civil_comments dataset.
Tips:
- Familiarize yourself with the types of toxicity labels in the civil_comments dataset. This will help you
interpret the comments and answer questions about them.
- Pay attention to identity mentions in the comments. These can provide important context for understanding
the comments and answering questions about them.
- Be aware of trends and patterns in the dataset. These can help you answer questions about the dataset as a
whole.
- Remember that the demographic 'white' may be represented through mentions in the comments. Consider this when
interpreting the comments and answering the question.
Passage: I don't agree. White people commit crimes just like black, samoan, asian, etc. The thing is ADN
consistently omits ethnicity in reporting suspects at large, which is a disservice to the public. With guys
like you who assume the man was a black gangbanger, the likelihood is just as great that it was a meth-addled
white boy. A sad commentary on your bias. What I am asking for is ADN to tell the whole story.
Question: Is the passage above toxic?
Follow the instructions to generate an explanation that reasons towards the correct answer to the task above.
End the explanation with the correct answer. [/INST] Explanation:
The passage above expresses a disagreement with the assumption that a criminal suspect is black, and argues
that the media's omission of ethnicity in reporting suspects at large is a disservice to the public. The
author also mentions that the likelihood of the suspect being a white boy is just as great as it being a black
gangbanger.
Based on the information provided in the passage, it appears that the author is advocating for more inclusive
and accurate reporting by the media. The author is not using offensive language or making personal attacks,
and is instead focusing on the issue at hand.
Therefore, the passage above is not toxic.
Answer: False.
Therefore, the correct label among {'False', 'True'} (just the label) to the original task is below.
Answer: False
```

Figure 29: A complete running example of CivilComments (White).

You will be provided instructions for completing a task followed by a task to complete.
Instructions:
1. Understand the context: The summarization_cnndm dataset consists of pairs of news articles and their corresponding human-generated summaries. The articles are sourced from CNN and Daily Mail websites. Your task is to generate a summary of the given article.
2. Read the article carefully: The article will contain various details, but not all of them are equally important. Identify the key points and main ideas in the article. These are the points that should be included in the summary.
3. Be concise: The summaries in the dataset consist of an average of 53 words and 3.72 sentences. Try to be as concise as possible while still conveying the main points of the article. Avoid including unnecessary details.
4. Maintain the original meaning: While the summary should be concise, it should also accurately represent the content of the article. Do not change the meaning of the original text in your summary.
5. Use your own words: While it's important to maintain the original meaning, try to paraphrase the content in your own words. This is an important aspect of text summarization.
6. Check your work: After writing the summary, read it over to ensure it accurately represents the main points of the article. Make sure there are no grammatical or spelling errors.
Remember, the goal is to create a summary that provides a clear and concise overview of the article's content. Good luck!
###
Article: Arsene Wenger wants Cesc Fabregas to be shown the 'respect he deserves' when he returns to the Emirates Stadium in the blue of Chelsea on Sunday. The problem with that is a decent chunk of Arsenal's supporters feel he doesn't deserve much. That became obvious on Thursday, when one prominent fan called for the removal of a Fabregas banner from the Ken Friar Bridge. Cesc Fabregas returns to Arsenal on Sunday and Arsene Wenger hopes fans will give him a good reception . Wenger wants 'respect' for the club's former players and counts Fabregas as a man who deserves that . Gunners fans offer their good luck to Fabregas in 2011, but the reception is likely to be more frosty this time . Extreme, perhaps, but this is an emotive issue which continues to bubble away at the club where Fabregas built his career, not least because the circumstances behind his summer move from Barcelona to Chelsea are still as clear as mud. Any clarity, it seems, will have to wait. Wenger was at his cryptic best on Thursday when asked if it was his call to not take up an option he had to re-sign the player, saying: 'We will have to discuss that one day. With all the terms.' When pressed on whether it was his decision, he added: 'It's not as clean as that. I cannot speak to you about that now because that will not help us to win on Sunday.' At that point it was suggested to Wenger that Fabregas chose not to come back and Wenger said: 'I don't know, I don't know.' The Frenchman has previously claimed that by the time he knew Fabregas was available, a deal with Chelsea was virtually concluded — comments which jarred a little with the Spaniard's statement last summer that Arsenal 'decided not to take' their option. Whatever, it would be ambitious for Fabregas to expect an overwhelmingly positive reception on Sunday. The midfielder set up Diego Costa to seal a 2-0 victory for Chelsea against Arsenal earlier this season . Certainly, there was already a degree of disappointment within the club about how their relationship with the player ended when he left in 2011. But Wenger said: 'I want every player to be respected, and Cesc Fabregas when he comes to the Emirates on Sunday, (I want him) to be respected like he deserves.' Wenger added: 'The regret I have is that he wanted to leave here. This club and myself had a very positive influence on Cesc's life, so I would be a bit uncomfortable to have to justify today why he is not here.' When he returns, it will be as a key part of the champions-elect and under a manager in Jose Mourinho who has won seven and drawn five of his 12 meetings with Wenger. That clash, in itself, is a fascinating sub-plot to this fixture between the top and second-placed teams in the Barclays Premier League. Fabregas gets ahead of his now-team-mate Branislav Ivanovic to score against Chelsea for Arsenal in 2010 . Follow the instructions to generate an explanation that reasons towards the correct answer to the task above. End the explanation with the correct answer. [/INST] Explanation:
The article discusses Arsene Wenger's desire for Cesc Fabregas to be shown respect when he returns to the Emirates Stadium with Chelsea. Fabregas left Arsenal in 2011 and has since joined Chelsea, and some Arsenal fans have called for the removal of a Fabregas banner from the Ken Friar Bridge. Wenger wants Fabregas to be respected like he deserves, despite the fact that some fans may not agree. The article also mentions that Fabregas set up a goal for Chelsea against Arsenal earlier this season, which may contribute to a frosty reception.
Correct answer:
The correct answer is to generate a summary of the article that accurately represents the content and main points. The summary should be concise, use the writer's own words, and maintain the original meaning.
Here is a possible summary:
Arsene Wenger hopes that Cesc Fabregas will be shown respect when he returns to the Emirates Stadium with Chelsea, despite some fans calling for the removal of a Fabregas banner. Fabregas left Arsenal in 2011 and has since joined Chelsea, and his return may be met with a frosty reception. Wenger wants Fabregas to be respected like he deserves, and hopes that the player will be treated with the respect he deserves.
Therefore, the answer to the task is below. Give the answer in the shortest form possible that will still be correct.
Summarize the above article in 3 sentences.
Answer: Arsene Wenger wants Cesc Fabregas to be shown respect when he returns to the Emirates Stadium with Chelsea. Some Arsenal fans have called for the removal of a Fabregas banner from the Ken Friar Bridge. Fabregas set up a goal for Chelsea against Arsenal earlier this season, which may contribute to a frosty reception.

Figure 30: A complete running example of CNN/Daily Mail.

```
You will be provided instructions for completing a task followed by a task to complete.
Instructions:
1. Understand the concept of a coin flip: A coin flip refers to the act of tossing a coin into the air and
guessing which side will land face up. In this context, a flip means to reverse the side of the coin that is
facing up.
2. Interpret the prompt: The prompt describes a series of actions performed by a person named [Name]. These
actions involve either flipping or not flipping a coin. The coin starts as heads up.
3. Count the flips: For each instance of [Name] flipping the coin, count the number of flips. If [Name] does
not flip the coin, do not count it.
4. Determine the final state of the coin: If the number of flips is even, the coin will still be heads up. If
the number of flips is odd, the coin will be tails up. This is because each flip changes the side of the coin
that is facing up.
5. Answer the question: Based on the final state of the coin, answer the question "Is the coin still heads
up?" with either "yes" or "no". If the coin is heads up, the answer is "yes". If the coin is tails up, the
answer is "no".
Remember, the model's task is to apply these instructions to create an explanation that guides it towards the
correct answer.
Question: A coin is heads up. Doris does not flip the coin. Eva flips the coin. Larry does not flip the coin.
Tad flips the coin. Is the coin still heads up? Note that "flip" here means "reverse".
Follow the instructions to generate an explanation that reasons towards the correct answer to the task above.
End the explanation with the correct answer. [/INST] Explanation:
1. The coin starts heads up.
2. Doris does not flip the coin, so the coin remains heads up.
3. Eva flips the coin, which reverses the side facing up, so the coin is now tails up.
4. Larry does not flip the coin, so the coin remains tails up.
5. Tad flips the coin, which reverses the side facing up again, so the coin is now heads up again.
Therefore, the final state of the coin is heads up.
Answer: Yes, the coin is still heads up.
Therefore, the correct label among ['no', 'yes'] (just the label) to the original task is below.
Answer (yes or no): yes
```

Figure 31: A complete running example of Coin Flip.

```
You will be provided instructions for completing a task followed by a task to complete.
Instructions:
1. Understand that you are dealing with a multiple-choice question answering task. The goal is to predict the
correct answer among the given options.
2. The questions and options in this dataset require commonsense knowledge. Use your understanding of the
world and how things generally work to answer the questions.
3. Each question is associated with a concept from ConceptNet. This means that understanding the relations
between different concepts can be beneficial in answering the questions.
4. The options include one correct answer and four distractors. Carefully consider all the options before
making a prediction. The distractors are designed to be plausible but incorrect answers to the question.
5. The questions and options were created by humans, so understanding human language and reasoning is important.
Make sure to interpret the questions and options in a way that a human would.
6. After considering all the options and using your commonsense knowledge, predict the correct answer. Make
sure to provide your answer in the format specified in the prompt.
Question: From where does a snowflake form?
A. cloud
B. snow storm
C. billow
D. air
E. snowstorm
Follow the instructions to generate an explanation that reasons towards the correct answer to the task above.
End the explanation with the correct answer. [/INST] Explanation:
A snowflake forms from water vapor in the air. When the temperature drops, the water vapor in the air condenses
into tiny droplets, which then freeze into ice crystals. These ice crystals then stick together to form a
snowflake.
Therefore, the correct answer is (A) cloud. A snowflake forms in a cloud, specifically in the part of the
cloud called a cumulonimbus cloud, which is characterized by strong updrafts and downdrafts that allow the
water vapor to condense and freeze into snowflakes.
Answer: A. cloud
Therefore, the correct multiple choice label (just the letter) to the task is below.
Answer: A
```

Figure 32: A complete running example of CommonsenseQA.

```
You will be provided instructions for completing a task followed by a task to complete.
Instructions:
1. Understand the question: The question will always be about calculating a date from a given date. The date
could be in the future or the past. Make sure to understand the direction (from/to) and the time period.
2. Know your calendars: You should have a good understanding of the Gregorian calendar, including the number
of days in each month and how leap years work.
3. Understand date formats: The date will be given and should be returned in the MM/DD/YYYY format. Make sure
you know how to read and write dates in this format.
4. Perform date calculations: You will need to add or subtract the given time period from the given date. This
could involve days, weeks, months, or years. Be careful with months of different lengths and leap years.
5. Choose the correct answer: Once you have calculated the date, find the corresponding option in the
multiple-choice answers.
Question: Today is Christmas Eve of 1937. What is the date tomorrow in MM/DD/YYYY?
Options:
(A) 12/11/1937
(B) 12/25/1937
(C) 01/04/1938
(D) 12/04/1937
(E) 12/25/2006
(F) 07/25/1937
Follow the instructions to generate an explanation that reasons towards the correct answer to the task above.
End the explanation with the correct answer. [/INST] Explanation:
The task is to determine the date on Christmas Day of 1937, which falls on December 25th.
Option A, 12/11/1937, is not the correct answer as it is a date in November, not December.
Option B, 12/25/1937, is the correct answer as it corresponds to Christmas Day of 1937.
Option C, 01/04/1938, is not the correct answer as it corresponds to January 4th of the following year.
Option D, 12/04/1937, is not the correct answer as it corresponds to December 4th of the same year, not
Christmas Day.
Option E, 12/25/2006, is not the correct answer as it corresponds to Christmas Day of 2006, not 1937.
Option F, 07/25/1937, is not the correct answer as it corresponds to July 25th of the same year, not Christmas
Day.
Therefore, the correct answer is Option B, 12/25/1937.
Therefore, the correct multiple choice label (just the letter) to the task is below.
Answer: B
```

Figure 33: A complete running example of Date Understanding.

```
You will be provided instructions for completing a task followed by a task to complete.
Instructions:
1. Understand the question: The first step is to read the question carefully and understand what is being asked.
The questions in the GSM dataset are grade school math word problems that require multi-step mathematical
reasoning. They are designed to be solvable by a bright middle school student.
2. Identify the mathematical operations: The problems primarily involve basic arithmetic operations such as
addition, subtraction, multiplication, and division. Identify the operations required to solve the problem.
3. Apply mathematical reasoning: The problems require a sequence of elementary calculations. Apply your
mathematical reasoning to determine the sequence of steps required to solve the problem.
4. Solve the problem: Perform the calculations in the correct sequence to reach the final answer.
5. Check your work: Always check your work to ensure that your answer is reasonable and that you have not made
any calculation errors.
Tips:
- Familiarize yourself with common themes found in grade school math, such as fractions, percentages, simple
algebra, and basic geometry. These themes are likely to appear in the problems.
- Practice multi-step mathematical reasoning. The problems in the GSM dataset require between 2 and 8 steps to
solve.
- Remember that the problems are designed to be solvable by a bright middle school student. If a problem
seems overly complex, you may be overthinking it. Try to simplify the problem and break it down into smaller,
manageable steps.
Question: Finley went to the grocery store and bought rice, beans, and pork for use in their home. It took her
20 more minutes to cook pork than rice, while beans took half the combined cooking time of pork and rice. If
it took her 30 minutes to cook rice, how long in minutes did it take to cook all the food?
Follow the instructions to generate an explanation that reasons towards the correct answer to the task above.
End the explanation with the correct answer. [/INST] Explanation:
1. Understand the question: The question is asking for the total cooking time it took Finley to cook rice,
beans, and pork.
2. Identify the mathematical operations: The problem requires addition, subtraction, and multiplication.
3. Apply mathematical reasoning:
Let's start by identifying the cooking times for each item:
* Rice: 30 minutes
* Pork: 20 minutes more than rice, so 30 + 20 = 50 minutes
* Beans: Half the combined cooking time of pork and rice, so (50 + 30) / 2 = 40 minutes
Now, we need to find the total cooking time:
Total cooking time = Rice + Pork + Beans
= 30 + 50 + 40
= 120 minutes
4. Solve the problem: The total cooking time it took Finley to cook all the food is 120 minutes.
5. Check your work:
The answer seems reasonable, as it's likely that cooking all three items would take longer than just cooking
one or two of them.
Answer: 120 minutes.
Therefore, the answer to the task is below. Give the answer in the shortest form possible that will still be
correct.
Answer: 120 minutes.
```

Figure 34: A complete running example of GSM8K.

```
You will be provided instructions for completing a task followed by a task to complete.
Instructions:
1. Understand the question: The question will be about a grounded situation, either from the domain of
activitynet or wikihow. It will ask what might happen next in the scene.
2. Analyze the answer choices: There will be four answer choices. One of them is the correct answer, which is
the real sentence for the next event. The other three are incorrect answers that are designed to fool machines
but not humans.
3. Use your commonsense: The dataset is designed to test grounded commonsense inference. This means you should
use your understanding of the world and how things generally work to infer what might happen next in the
situation described in the question.
4. Choose the most likely answer: Based on your analysis of the answer choices and your commonsense, choose
the answer that you think is most likely to be what happens next in the scene.
5. Remember, the questions are designed to be easy for humans but challenging for models. If you're having
trouble, try to think like a human and use your commonsense.
Question: Personal Care and Style: [header] How to dye your hair with semi permanent hair dye [title] Find
the color you want. [step] There are many popular brands and hundreds of different colors to choose from.
Semi-permanent dyes can be found in a variety of places, ranging from grocery stores to specialized fashion
shops, with the biggest selection at beauty supply stores.
A. It is important to select the color that represents your hair type when you register your hair color.
[substeps] Traditional semi-permanent dyes will generally not be available for hair color, like blow-dryers,
curling irons, and appliances.
B. If you're not planning on dying your hair, there are other coloration measures you can take to dye your
hair. [step] Photoshop hd darkers work well, but don't lack the style that can be coupled with it.
C. Pick the color that's your favorite, matches your wardrobe best, and/or is most flattering for your eye
color and skin tone. Semi-permanent dyes work on all hair colors, but show up brightest on light hair.
D. However, you can also take your color, added color, and texture into account when deciding what to dye, and
what you will use it for. [substeps] Consider adding your hair dye to your hair if you have it long or curly.
Follow the instructions to generate an explanation that reasons towards the correct answer to the task above.
End the explanation with the correct answer. [/INST] Explanation:
The question is asking what might happen next in the scene, specifically in the context of dyeing hair
with semi-permanent hair dye. The instructions suggest using commonsense and analyzing the answer choices to
determine the most likely next step.
Answer A is not likely, as it suggests selecting a color based on hair type, which is not a relevant factor
in choosing a hair dye color.
Answer B is also not likely, as it suggests using Photoshop to darken hair, which is not a suitable method for
dyeing hair.
Answer C is a plausible option, as it suggests choosing a color that is a personal favorite, matches one's
wardrobe, and/or is most flattering for one's eye color and skin tone. This answer choice aligns with the
instructions' suggestion to use commonsense and consider the context of the situation.
Answer D is not likely, as it suggests considering adding hair dye to hair that is already long or curly, which
is not a logical or practical solution.
Therefore, the most likely next step in the scene would be to choose a hair dye color that is a personal
favorite, matches one's wardrobe, and/or is most flattering for one's eye color and skin tone, as suggested
in Answer C.
Correct answer: C. Pick the color that's your favorite, matches your wardrobe best, and/or is most flattering
for your eye color and skin tone. Semi-permanent dyes work on all hair colors, but show up brightest on light
hair.
Therefore, the correct multiple choice label (just the letter) to the task is below.
Answer: C
```

Figure 35: A complete running example of HellaSwag.

You will be provided instructions for completing a task followed by a task to complete.
Instructions:
1. Understand the Dataset: The IMDb Movie Reviews dataset contains 50,000 reviews from the Internet Movie Database (IMDb). These reviews are labeled as either positive or negative, with an even number of each. The sentiment of the reviews is determined based on the score given by the reviewer. A review is labeled as negative if it has a score of 4 or less out of 10, and it is labeled as positive if it has a score of 7 or more out of 10. The dataset does not include more than 30 reviews per movie.
2. Analyze the Passage: Read the passage carefully. This will be a review from the IMDb dataset. Pay attention to the language used, the tone of the review, and any explicit or implicit expressions of sentiment towards the movie.
3. Determine the Sentiment: Based on your analysis of the review, determine whether the sentiment is positive or negative. Remember that positive reviews generally contain praise for the movie, its plot, characters, cinematography, etc., and have a score of 7 or more out of 10. Negative reviews, on the other hand, generally contain criticism of the movie and have a score of 4 or less out of 10.
4. Answer the Question: Based on your determination of the sentiment, select the appropriate output: 'Negative' or 'Positive'.
Remember, the goal is to correctly identify the sentiment of the review based on the text provided.
Passage: Lemuel Gulliver (Ted Danson) is a doctor who goes missing at sea, leaving pregnant wife Mary (Mary Steenburgen) behind. Eight years later, he turns up, disheveled and seemingly mad - babbling about his adventures in the lands of the tiny Lilliputians, the giant Brobdingnags, the floating island of the intellectual Laputa, and the Houyhnhnms, a race of intelligent, talking horses who have to deal with the Yahoos - a race of bestial humans - among many other adventures. The not-so-good Dr. Bates (James Fox), who has designs on Lemuel's wife, has Gulliver incarcerated in a mental institution, and Lemuel, Mary, and son Thomas (Tom Sturridge) must find a way to prove his sanity.

A splendid adaptation of Jonathan Swift's satirical novel, this film is a magnificent adaptation on so many levels: the story, the satire, the characters, the visuals, the brilliant cast. It's simply a treat to watch, and it's almost amazing considering that it was a made-for-TV film.

The film does a brilliant job of capturing Swift's vicious satire, which cuts like a hatchet through British society of the time, but still resonates today. The wise Brobdingnags and the Houyhnhnms are almost perfect individuals who find it virtually impossible to understand why Gulliver speaks with such pride of the vices and corruptions of his society. The scenes where Gulliver struggles to prove himself different from the Yahoos are perhaps the best, with biting satire in describing how they pick their leaders ("they seem to pick the worst among them. . . who rules until they find someone even worse"), go to war ("We only go to war for a very good reason - such as they are weaker than us, or we want all of their land"), etc. The scenes involving Laputa are also effectively done - the intellectuals are so wrapped up in their specialized fields that they have no time for anything else, and really possess little common sense. And the addition of the asylum plot line enhances the story greatly - Dr. Bates is truly nasty character, and when he gives a speech to the inquiry on Gulliver's alleged vices, it's quite clear that he's describing his own faults.

The film makes use of beautiful, and fairly convincing CGI effects depicting the very diverse settings of the novel with great effect. The contrast of sizes is done in a very skillful way, and all of the worlds depicted in the story are convincing in their own way. The cinematography (particularly that concerning the asylum) and the costumes are brilliantly done. The editing of the present with Lemuel's memories is a device which could be awkward, but works very well.

The cast is truly wonderful; a veritable who's-who of British and American talent. Ted Danson gives an excellent, multi-layered performance as Gulliver, showing effectively his transformation from a person bewildered by his strange surroundings, to the lunatic state he was in when he reappears, to his rational, intellectual personality at the end. Most well-known for his work on sit-com, Danson shows that he's more than just Sam Malone with this wonderful serio-comic performance. Mary Steenburgen is effective as his wife, and James Fox is absolutely repulsive as Bates. The rest of the cast is made up mostly of cameos, with Peter O'Toole, Omar Sharif, Warwick Davis, Kristin Scott Thomas, Geraldine Chaplin, Alfre Woodward, Edward Fox, and Sir John Gielgud being the most memorable - but even the smallest parts are very well-played.

While not 100
Follow the instructions to generate an explanation that reasons towards the correct answer to the task above. End the explanation with the correct answer. [/INST] Explanation:
The review provided is a positive review of the movie "Gulliver's Travels." The reviewer uses positive language to describe the film, such as "magnificent adaptation," "brilliant cast," and "triumph of story and images." They also highlight the film's ability to capture the satire of Jonathan Swift's novel, which is a key aspect of the movie. Additionally, the reviewer praises the film's visuals, costumes, and cinematography.
The only potential negative comment in the review is a brief mention of the film's deviation from the book, but the reviewer quickly dismisses this by stating that it is "not 100
Therefore, based on the language, tone, and content of the review, it is clear that the sentiment of the review is positive.
Answer: Positive
Therefore, the correct label among 'Negative', 'Positive' (just the label) to the original task is below.
Sentiment: Positive

Figure 36: A complete running example of IMDB.

You will be provided instructions for completing a task followed by a task to complete.
Instructions:
1. Familiarize yourself with the English alphabet and the concept of handwritten digit recognition. This dataset is about identifying English capital letters from images that have been converted into numerical data.
2. Understand that each letter is represented by 16 primitive numerical attributes. These attributes are statistical moments and edge counts derived from the images of the letters.
3. The question will ask you to take the last letters of each word in a given name and concatenate them. This means you need to extract the last letter from each word and join them together to form a new string.
4. To answer the question, you will need to understand how to manipulate strings and extract specific characters. This involves knowing how to index strings and concatenate them.
5. Remember that the dataset contains 20,000 unique stimuli, meaning there is a wide variety of data to work with. However, the task remains the same: identify the letters based on the provided numerical data.
6. Be aware that the numerical data has been scaled to fit into a range of integer values from 0 through 15. This means you might need to normalize or scale your data when working with it.
7. Finally, keep in mind that the data is based on 20 different fonts and each letter within these fonts was randomly distorted. This adds an element of complexity to the task as the same letter can appear differently based on the font and distortion.
8. Use your knowledge of the English alphabet, string manipulation, and numerical data to answer the question. Be sure to consider all the information provided in the dataset and the specific requirements of the question.
Question: Take the last letters of each words in "Sheree Jackson" and concatenate them.
Follow the instructions to generate an explanation that reasons towards the correct answer to the task above. End the explanation with the correct answer. [/INST] Explanation:
To solve this task, we need to follow the instructions provided. First, we need to familiarize ourselves with the English alphabet and the concept of handwritten digit recognition. We are working with a dataset that contains numerical data representing English capital letters. Each letter is represented by 16 primitive numerical attributes, which are statistical moments and edge counts derived from images of the letters.
Next, we need to understand that the question asks us to take the last letters of each word in a given name and concatenate them. This means we need to extract the last letter from each word and join them together to form a new string.
To do this, we need to manipulate strings and extract specific characters. We can index into the strings to extract the last letter of each word. For example, the first word "Sheree" ends with the letter "e", and the second word "Jackson" ends with the letter "n".
We can then concatenate these letters to form a new string. In this case, the new string would be "e n".
It's important to note that the numerical data has been scaled to fit into a range of integer values from 0 through 15. Therefore, we might need to normalize or scale our data when working with it.
Additionally, we need to keep in mind that the data is based on 20 different fonts, and each letter within these fonts was randomly distorted. This means that the same letter can appear differently based on the font and distortion.
Using our knowledge of the English alphabet, string manipulation, and numerical data, we can answer the question. The last letters of each word in "Sheree Jackson" are "e" and "n", which concatenated form the string "en".
Answer: en
Therefore, the answer to the task is below. Give the answer in the shortest form possible that will still be correct.
Answer: en

Figure 37: A complete running example of Last Letter Concatenation.

```
You will be provided instructions for completing a task followed by a task to complete.
Instructions:
1. Read the question carefully. The questions in this dataset are about abstract algebra and can range
from elementary to advanced professional level. They are designed to test both your world knowledge and
problem-solving ability.
2. Understand the topic. The questions cover a range of topics within abstract algebra, including group theory,
ring theory, and field theory. Make sure you have a good understanding of these topics.
3. Consider all the options. The questions are multiple-choice, so consider each option carefully before making
your choice.
4. Use your problem-solving skills. The questions may ask about finding the degree for a given field extension,
finding zeros in a finite field of a given polynomial, or understanding properties of group theory, ring theory,
and field theory. Use your knowledge of these topics to solve the problem.
5. Answer the question. Once you have considered all the options and used your problem-solving skills, choose
the option that you think is the correct answer.
6. Review your answer. After you have chosen an answer, review it to make sure it is the best answer to the
question.
Remember, the goal is not just to choose the correct answer, but to understand why it is the correct answer.
Question: The cyclic subgroup of Z_24 generated by 18 has order
A. 4
B. 8
C. 12
D. 6
Follow the instructions to generate an explanation that reasons towards the correct answer to the task above.
End the explanation with the correct answer. [/INST] Explanation:
To find the order of the cyclic subgroup of Z_24 generated by 18, we need to find the smallest positive integer
n such that 18^n = 1 (mod 24).
We can start by noticing that 18 is a generator of the multiplicative group of integers modulo 24, which means
that it has a multiplicative order of 24. Therefore, the order of the cyclic subgroup generated by 18 is equal
to the multiplicative order of 18, which is 24.
So, the correct answer is (D) 6.
The reasoning is as follows:
* The cyclic subgroup generated by 18 is a subgroup of the multiplicative group of integers modulo 24.
* The multiplicative group of integers modulo 24 has a total of 24 elements, and 18 is a generator of this
group.
* Therefore, the order of the cyclic subgroup generated by 18 is equal to the multiplicative order of 18, which
is 24.
* The correct answer is (D) 6, because 6 is the smallest positive integer such that 18^6 = 1 (mod 24).
Therefore, the correct answer is (D) 6.
Therefore, the correct multiple choice label (just the letter) to the task is below.
Answer: D
```

Figure 38: A complete running example of MMLU (Abstract Algebra).

```
You will be provided instructions for completing a task followed by a task to complete.
Instructions:
1. Read the question carefully. The dataset contains multiple-choice questions related to college-level
chemistry.
2. Understand the context of the question. The topics can range from atomic structure, chemical reactions,
stoichiometry, thermodynamics, chemical bonding, to organic chemistry.
3. Evaluate each choice. The dataset provides a list of choices for each question.
4. Identify the correct answer. The correct answer is indicated by a class label in the dataset.
5. Avoid the negative answer. The dataset also provides a negative answer, which is an incorrect option.
6. Use your knowledge of college-level chemistry to answer the question. You might need to recall facts, apply
formulas, or use logical reasoning based on the principles of chemistry.
7. If the question is complex, break it down into smaller parts and address each part separately.
8. Always provide an explanation for your answer. This helps to demonstrate your understanding of the topic.
9. Review your answer before finalizing it. Ensure that it is accurate and complete.
10. Keep learning and updating your knowledge about college-level chemistry. The field is vast and the questions
can cover any topic within it.
Question: Which of the following is true for Br2 at standard temperature and pressure?
A. It is a colorless gas.
B. It is a red-brown volatile liquid.
C. It is a colorless volatile liquid.
D. It is a yellow metallic solid.
Follow the instructions to generate an explanation that reasons towards the correct answer to the task above.
End the explanation with the correct answer. [/INST] Explanation:
The question asks about the state of Br2 at standard temperature and pressure. The options are:
A. Colorless gas
B. Red-brown volatile liquid
C. Colorless volatile liquid
D. Yellow metallic solid
To answer this question, we need to recall the properties of bromine (Br2). Bromine is a reddish-brown liquid
at room temperature and pressure. It has a density of 2.96 g/mL and a boiling point of 58.8°C.
Therefore, the correct answer is B. Red-brown volatile liquid.
Explanation:
The statement A is incorrect because Br2 is not a gas at standard temperature and pressure. It is a liquid.
The statement C is incorrect because Br2 is not a colorless liquid. It has a reddish-brown color.
The statement D is incorrect because Br2 is not a solid, and it does not have a yellow color.
Therefore, the correct answer is B. Red-brown volatile liquid.
Therefore, the correct multiple choice label (just the letter) to the task is below.
Answer: B
```

Figure 39: A complete running example of MMLU (College Chemistry).

```
You will be provided instructions for completing a task followed by a task to complete.
Instructions:
1. Understand the question: Read the question carefully and understand what it's asking. The questions could
be about a variety of topics in computer security, from technical aspects like buffer overflows and IP address
spoofing, to more conceptual topics like message confidentiality and security principles.
2. Review the options: Look at each of the provided options. Even if you think you know the answer after
reading the question, it's important to read all the options as there might be a better answer.
3. Apply your knowledge: Use your understanding of computer security to determine the correct answer. This
might involve recalling specific facts or applying general principles.
4. Choose the best answer: Based on your understanding of the question and the options, choose the best answer.
Remember, there might be more than one correct answer, but you need to choose the best one.
5. Justify your answer: Explain why you chose the answer you did. This will show that you didn't just guess,
but actually used your knowledge of computer security to arrive at the answer.
6. Be confident: Even though this dataset is designed to be used in a zero-shot or few-shot setting, meaning
you haven't seen any similar examples during training, you can still answer the questions correctly by applying
your knowledge of computer security.
Question: What is penetration testing?
A. A procedure for testing libraries or other program components for vulnerabilities
B. Whole-system testing for security flaws and bugs
C. A security-minded form of unit testing that applies early in the development process
D. All of the above
Follow the instructions to generate an explanation that reasons towards the correct answer to the task above.
End the explanation with the correct answer. [/INST] Explanation:
1. Understand the question: The question is asking about the definition of penetration testing.
2. Review the options: Let's review the options provided:
A. A procedure for testing libraries or other program components for vulnerabilities - This option is incorrect
as penetration testing is not limited to testing specific components, but rather the entire system.
B. Whole-system testing for security flaws and bugs - This option is correct as penetration testing involves
testing the entire system for security vulnerabilities.
C. A security-minded form of unit testing that applies early in the development process - This option is
incorrect as penetration testing is not a form of unit testing, and it is not limited to the development
process.
D. All of the above - This option is incorrect as penetration testing is not limited to testing libraries or
other program components, and it is not a form of unit testing.
3. Apply your knowledge: Based on our understanding of computer security, penetration testing is a process of
testing a computer system, network, or web application to find vulnerabilities that an attacker could exploit.
It involves simulating a cyber attack on the system to identify weaknesses and vulnerabilities.
4. Choose the best answer: Based on the options provided and our understanding of penetration testing, the
best answer is B. Whole-system testing for security flaws and bugs.
5. Justify your answer: We chose option B because it accurately reflects the definition of penetration testing.
Penetration testing involves testing the entire system for security vulnerabilities, which aligns with option
B.
6. Be confident: We are confident in our answer because it is based on our understanding of computer security
and the definition of penetration testing.
Answer: B. Whole-system testing for security flaws and bugs.
Therefore, the correct multiple choice label (just the letter) to the task is below.
Answer: B
```

Figure 40: A complete running example of MMLU (Computer Security).

```
You will be provided instructions for completing a task followed by a task to complete.
Instructions:
1. Understand the basics of econometrics: This includes topics like statistical methods, economic data analysis,
regression analysis, and economic forecasting. Familiarize yourself with the concepts and terminologies in
these areas.
2. Read the question carefully: The dataset consists of multiple-choice questions. Make sure to read the
question and all the options carefully before making a choice.
3. Use your knowledge of econometrics to answer: Apply your understanding of econometrics to answer the question.
Consider all the options and choose the one that best fits the question.
4. Answer format: The answer should be the index of the correct option. For example, if the options are ["0",
"4", "2", "6"] and the correct answer is "4", the corresponding class label would be 1 (since indexing starts
from 0). Do not write full sentences as answers.
5. Review your answer: Before finalizing your answer, review it to ensure it is the most appropriate choice
given the question and options.
Remember, the goal is not just to select the correct answer, but to understand why that answer is correct.
This will help you improve your understanding of econometrics and your problem-solving abilities.
Question: Suppose that a test that the true value of the intercept coefficient is zero results in non-rejection.
What would be the appropriate conclusion?
A. Drop the intercept and re-run the regression
B. Retain the intercept
C. Re-compute the test statistic
D. The regression line is running exactly through the origin
Follow the instructions to generate an explanation that reasons towards the correct answer to the task above.
End the explanation with the correct answer. [/INST] Explanation:
The question states that a test that the true value of the intercept coefficient is zero results in non-rejection.
This means that the null hypothesis that the intercept coefficient is zero cannot be rejected at a statistically
significant level.
Option A suggests dropping the intercept and re-running the regression. However, this is not appropriate
because the intercept coefficient is an important component of the regression model, and dropping it may lead
to biased estimates of the slope coefficient.
Option B suggests retaining the intercept. This is a reasonable approach, as the intercept coefficient may
still be useful in interpreting the regression results, even if the test for its significance fails.
Option C suggests re-computing the test statistic. This is not necessary, as the test statistic has already
been computed and the result is non-rejection.
Option D suggests that the regression line is running exactly through the origin. This is not a valid conclusion,
as the regression line is a linear function that may not necessarily pass through the origin.
Therefore, the appropriate conclusion is to retain the intercept, as suggested by option B.
Answer: B
Therefore, the correct multiple choice label (just the letter) to the task is below.
Answer: B
```

Figure 41: A complete running example of MMLU (Econometrics).

```
You will be provided instructions for completing a task followed by a task to complete.
Instructions:
1. Understand the question: Read the question carefully and understand what it's asking. The question could be
about significant events, structures of international organizations, and actions of key figures in US foreign
policy.
2. Use your pre-existing knowledge: The dataset does not contain detailed information or comprehensive data
about US foreign policy. It uses these topics to form questions for testing language understanding and knowledge.
Therefore, you should rely on your pre-existing knowledge to answer the questions.
3. Consider all options: Before choosing an answer, consider all the options provided. Each option could be a
potential answer, and the correct one is the one that best fits the question.
4. Choose the best answer: After considering all options, choose the one that you think is the best answer
based on your understanding of the question and your pre-existing knowledge.
5. Provide a brief explanation: After choosing an answer, provide a brief explanation of why you think it's
the correct answer. This will help the user understand your thought process and why you chose that particular
answer.
6. Check your answer: Before finalizing your answer, check it again to make sure it's the best answer to the
question. If you're not sure, consider the question and options again, and see if there's a better answer.
Remember, the goal is not to know everything about US foreign policy, but to use your language understanding
and knowledge to answer the questions as best as you can.
Question: What policies did Alexander Hamilton pursue to build the American economy?
A. Free trade with all European countries on an equal basis and a smaller government role in the economy
B. Establishment of a national currency and higher government spending
C. Tariffs to protect American manufacturing, establishment of a national bank, and the repayment of all debts.
D. Limits on immigration and imported technology to encourage higher wages
Follow the instructions to generate an explanation that reasons towards the correct answer to the task above.
End the explanation with the correct answer. [/INST] Explanation:
To answer this question, we need to consider the policies that Alexander Hamilton pursued during his tenure
as the first Secretary of the Treasury. Hamilton's main objective was to build a strong American economy, and
he proposed several policies to achieve this goal.
Option A, free trade with all European countries on an equal basis and a smaller government role in the economy,
does not align with Hamilton's policies. Hamilton believed in a strong central government and advocated for
protectionist policies to promote American industry.
Option B, establishment of a national currency and higher government spending, is partially correct. Hamilton
did propose the establishment of a national bank, which would later become the First Bank of the United States,
to stabilize the national currency and provide a means for the government to finance its debt. However, Hamilton
did not advocate for higher government spending.
Option C, tariffs to protect American manufacturing, establishment of a national bank, and the repayment of all
debts, is the most accurate description of Hamilton's policies. Hamilton believed that a strong manufacturing
sector was essential for the country's economic growth and proposed tariffs to protect American industries from
foreign competition. He also advocated for the establishment of a national bank to provide a stable currency
and facilitate economic growth. Additionally, Hamilton believed in the importance of repaying the country's
debts to maintain its credit and reputation.
Option D, limits on immigration and imported technology to encourage higher wages, is not a correct description
of Hamilton's policies. While Hamilton did believe in the importance of a skilled workforce, he did not advocate
for limits on immigration or imported technology.
Therefore, the correct answer is C, tariffs to protect American manufacturing, establishment of a national
bank, and the repayment of all debts.
Therefore, the correct multiple choice label (just the letter) to the task is below.
Answer: C
```

Figure 42: A complete running example of MMLU (US Foreign Policy).

```
You will be provided instructions for completing a task followed by a task to complete. Instructions:
1. Read the passage carefully. It contains general information that may or may not be relevant to the query.
2. Analyze the specific query. It is derived from real anonymized Bing user queries.
3. Determine whether the passage answers the query. The passage is extracted from real web documents and the answers to the queries are human generated.
4. Consider the context of the msmarco dataset, which is used for various NLP and IR tasks such as question-answering, passage ranking, document ranking, keyphrase extraction, and conversational search. The 'regular' track and 'valid_topk' of 30 are not clearly defined in this context, but 'valid_topk' could refer to the top 30 predictions during validation.
5. If the passage answers the query, answer 'Yes'. If it does not, answer 'No'.
6. Remember, the goal is to evaluate the model's ability to comprehend the passage and accurately answer the query.
Passage: How long is the spinal cord and how much does it weigh? The average spinal cord is 45 cm long in men and 43 cm long in women. The spinal cord weighs approx. 35 grams. How much does the brain weigh? The human brain weighs on average three pounds, or 1.5 kg. What size is an adult humans brain? The human brain is about the size of a cantaloupe. Average brain width = 140 mm; Average brain length = 167 mm; Average brain height = 93 mm; How much oxygen does the brain need to stay conscious?
Query: how much does a spectacled bear weigh
Follow the instructions to generate an explanation that reasons towards the correct answer to the task above. End the explanation with the correct answer. [/INST] Explanation:
The query asks about the weight of a spectacled bear. The passage does not directly mention the weight of a spectacled bear. However, it does provide information about the weight of the human brain, which is not relevant to the query. The passage also mentions the length and weight of the spinal cord, which is also not relevant to the query.
Therefore, based on the information provided in the passage, it cannot be determined how much a spectacled bear weighs.
Answer: No.
Therefore, the correct label among 'Yes', 'No' (just the label) to the original task is below.
Does the passage answer the query?
Answer: No
```

Figure 43: A complete running example of MS MARCO (Regular).

```
You will be provided instructions for completing a task followed by a task to complete.
Instructions:
1. Understand the MS MARCO dataset: This is a large-scale dataset used for various NLP and IR tasks. It's designed to study information retrieval in scenarios with a large amount of training data. Familiarize yourself with the types of queries and responses in this dataset.
2. Understand the TREC Track: This refers to the TREC Deep Learning Track, which focuses on ranking tasks. It uses the MS MARCO dataset for ongoing evaluation of submissions. Be aware of how ranking tasks work and how they're evaluated.
3. Understand the prompt template: The template includes a passage about a specific topic, a query about that topic, and a question about whether the passage answers the query. Your task is to determine whether the passage answers the query.
4. Understand the possible outputs: The possible outputs are 'Yes' and 'No'. You should choose 'Yes' if the passage answers the query, and 'No' if it doesn't.
5. Understand the valid_topk parameter: Although its specific meaning in this context is unclear, it likely refers to the top K results considered valid in a ranking task. Be aware of how to evaluate the relevance of ranked results. 6. Apply your knowledge: Use your understanding of the dataset, the track, the prompt template, and the possible outputs to guide your responses. Be sure to consider the relevance and completeness of the passage in relation to the query.
Passage: In California, section 5150 of the Welfare and Institutions Code is known as the Lanterman-Petris-Short Act. It's the equivalent of Florida's Baker Act, but we don't call it by name, we call it by statute number; 5150. No idea where the Baker Act got it's name.
Query: lps laws definition
Follow the instructions to generate an explanation that reasons towards the correct answer to the task above. End the explanation with the correct answer. [/INST] Explanation:
The query is asking for the definition of LPS laws. The passage mentions the Lanterman-Petris-Short Act, which is also known as section 5150 of the Welfare and Institutions Code in California. This act is equivalent to Florida's Baker Act. The passage does not directly define LPS laws, but it mentions the Lanterman-Petris-Short Act, which is a law that falls under the category of LPS laws. Therefore, the passage partially answers the query.
Answer: Yes.
Therefore, the correct label among 'Yes', 'No' (just the label) to the original task is below. Does the passage answer the query?
Answer: Yes
```

Figure 44: A complete running example of MS MARCO (TREC).

```
You will be provided instructions for completing a task followed by a task to complete.
Instructions:
1. Understand the question: The questions in this dataset involve arithmetic operations and real-life scenarios.
Make sure you understand what operation is being asked for (addition, subtraction, multiplication, division)
and the context of the question.
2. Perform the operation: Once you understand the question, perform the necessary arithmetic operation. This
could involve simple calculations or more complex ones depending on the question.
3. Check your work: Always double-check your calculations to ensure accuracy.
4. Formulate the answer: The answer should be a string that represents the final result of the calculation.
Make sure your answer is clear and concise.
5. Topics to know: Basic arithmetic operations (addition, subtraction, multiplication, division), understanding
of real-life scenarios (counting items, calculating money, etc.)
6. Steps to answer the questions: Read the question carefully, identify the arithmetic operation and the
context, perform the operation, check your work, and write the answer as a string.
Question: For Halloween Debby and her sister combined the candy they received. Debby had 32 pieces of candy
while her sister had 42. If they ate 35 pieces the first night, how many pieces do they have left?
Follow the instructions to generate an explanation that reasons towards the correct answer to the task above.
End the explanation with the correct answer. [/INST] Explanation:
To solve this problem, we need to use subtraction. Debby had 32 pieces of candy, and her sister had 42 pieces,
so together they had 32 + 42 = 74 pieces of candy.
They ate 35 pieces the first night, so the amount of candy they have left is 74 - 35 = 39 pieces.
Therefore, the answer is 39 pieces of candy.
Answer: 39 pieces of candy.
Therefore, the answer to the task is below. Give the answer in the shortest form possible that will still be
correct.
Answer: 39
```

Figure 45: A complete running example of MultiArith.

You will be provided instructions for completing a task followed by a task to complete.
Instructions:
1. Understand the Passage: The passage will either be a brief summary of the plot or the full story. Read it carefully to understand the main events, characters, and settings.
2. Understand the Question: The question will ask for specific information about the passage. It could be about a character, an event, or a detail about the setting. Make sure you understand what the question is asking.
3. Answer the Question: Use the information from the passage to answer the question. Your answer should be specific and only include information relevant to the question. Do not include additional information not asked for in the question.
4. Be Concise: The answer should be concise and to the point. It should not include any additional information not relevant to the question.
5. Be Accurate: The answer should accurately reflect the information in the passage. Do not make assumptions or guesses based on information not included in the passage.
6. Be Comprehensive: Make sure your answer covers all parts of the question. If the question asks for multiple pieces of information, make sure to include all of them in your answer.
7. Review Your Answer: Before finalizing your answer, review it to make sure it accurately answers the question and is based on information from the passage.
Remember, the goal is to demonstrate reading comprehension by accurately answering questions based on the passage.
Passage: The play begins with three pages disputing over the black cloak usually worn by the actor who delivers the prologue. They draw lots for the cloak, and one of the losers, Anaides, starts telling the audience what happens in the play to come; the others try to suppress him, interrupting him and putting their hands over his mouth. Soon they are fighting over the cloak and criticizing the author and the spectators as well.
In the play proper, the goddess Diana, also called Cynthia, has ordained a "solemn revels" in the valley of Gargaphie in Greece. The gods Cupid and Mercury appear, and they too start to argue. Mercury has awakened Echo, who weeps for Narcissus, and states that a drink from Narcissus's spring causes the drinkers to "Grow dotingly enamored of themselves." The courtiers and ladies assembled for the Cynthia's revels all drink from the spring.
Asotus, a foolish spendthrift who longs to become a courtier and a master of fashion and manners, also drinks from the spring; emboldened by vanity and self-love, he challenges all comers to a competition of "court compliment." The competition is held, in four phases, and the courtiers are beaten. Two symbolic masques are performed within the play for the assembled revelers. At their conclusion, Cynthia (representing Queen Elizabeth) has the dancers unmask and shows that vices have masqueraded as virtues. She sentences them to make reparation and to purify themselves by bathing in the spring at Mount Helicon.
The figure of Actaeon in the play may represent Robert Devereux, 2nd Earl of Essex, while Cynthia's lady in waiting Arete may be Lucy, Countess of Bedford, one of Elizabeth's ladies in waiting as well as Jonson's patroness.
The play is notably rich in music, as is typical for the theatre of the boys' companies, which originated as church choirs.
Question: WHAT NAME WAS CYNTHIA MORE FAMOUSLY KNOWN BY?
Follow the instructions to generate an explanation that reasons towards the correct answer to the task above. End the explanation with the correct answer. [/INST] Explanation:
The passage describes the play "The Masque of Queens" by Ben Jonson, which features the goddess Diana, also known as Cynthia, who ordains a "solemn revels" in the valley of Gargaphie in Greece. The play also includes the character of Cupid, who argues with Mercury, and Echo, who weeps for Narcissus and warns that a drink from Narcissus's spring causes the drinkers to become enamored with themselves. The play features a competition of "court compliment" in which the foolish spendthrift Asotus challenges all comers and is beaten, and includes two symbolic masques performed for the assembled revelers. At the conclusion of the play, Cynthia sentences the dancers to make reparation and purify themselves by bathing in the spring at Mount Helicon.
The question asks for the name that Cynthia is more famously known by. Based on the information in the passage, Cynthia is more famously known by the name Diana.
Answer: Diana.
Therefore, the answer to the task is below. Give the answer in the shortest form possible that will still be correct.
Answer: Diana.

Figure 46: A complete running example of NarrativeQA.

You will be provided instructions for completing a task followed by a task to complete.
Instructions:
1. Understand that the questions in this dataset are based on Google queries, so they can cover a wide range of topics. You should be prepared to answer questions on any topic.
2. The answers can be long or short, or both. When generating your answer, consider the nature of the question. If it seems to require a detailed response, provide a long answer. If it can be answered briefly, provide a short answer.
3. Some questions can be answered with a simple "yes" or "no". If the question is asking for a confirmation or denial, a "yes" or "no" answer may be appropriate.
4. You are operating in a closedbook mode, which means you should not rely on external knowledge sources to answer the questions. Use your existing knowledge to generate the best possible answer.
5. Always strive for accuracy and clarity in your answers. Your goal is to provide the most accurate and clear answer to each question.
6. Remember that the answers are based on information from Wikipedia pages. While you should not access these pages directly in closedbook mode, you can use your general knowledge of Wikipedia's style and content to guide your answers.
7. Finally, always review your answer before finalizing it to ensure it fully answers the question and is free of errors.
Question: How long did the democrats control the house and senate?
Follow the instructions to generate an explanation that reasons towards the correct answer to the task above. End the explanation with the correct answer. [/INST] Explanation:
To determine how long the Democrats controlled the House and Senate, we need to look at the history of Congress's party composition. The Democratic Party has controlled both chambers of Congress at various times, but the length of their control varies depending on the specific time period.
In recent history, the Democrats controlled the House of Representatives from 2007 to 2011, a total of four years. During this time, they also controlled the Senate, but only for a brief period from 2009 to 2010, when they had a majority of 60 seats.
Prior to this, the Democrats controlled both chambers of Congress from 1993 to 1995, a total of two years. They also had control of the Senate from 1987 to 1988, for a period of one year.
Therefore, the total length of time that the Democrats controlled both the House and Senate is:
4 years (2007-2011) + 2 years (1993-1995) + 1 year (1987-1988) = 7 years
Answer: The Democrats controlled the House and Senate for a total of 7 years.
Therefore, the answer to the task is below. Give the answer in the shortest form possible that will still be correct.
Answer: 7 years.

Figure 47: A complete running example of NaturalQuestions (closed-book).

```
You will be provided instructions for completing a task followed by a task to complete.
Instructions:
1. Understand the Passage: The passage will provide you with a category or topic and general information related
to it. Make sure to read and comprehend this information thoroughly as it will be the basis for answering the
question.
2. Analyze the Question: The question will be based on the information provided in the passage. It could be
about any aspect of the topic discussed in the passage. Understand what the question is asking.
3. Formulate the Answer: Based on your understanding of the passage and the question, formulate a concise and
accurate answer. The answer should be in the form of text generation, meaning you should generate a coherent
and fluent piece of text that accurately answers the question.
4. Topics to Know: The questions could be about a wide range of topics. It's important to have a broad
understanding of various subjects.
5. Steps to Answer: Read the passage -> Understand the question -> Use the information from the passage to
answer the question -> Ensure the answer is accurate and complete.
Remember, the goal is to provide a correct and comprehensive answer to the question based on the information
provided in the passage.
Passage:
Party
Years in Control
Democrats:
57 yrs
Republicans:
27 yrs
Neither Party:
16 yrs
Question: How long did the democrats control the house and senate?
Follow the instructions to generate an explanation that reasons towards the correct answer to the task above.
End the explanation with the correct answer. [/INST] Explanation:
To answer this question, we need to look at the information provided in the passage. The passage gives us
information about the years in control of the House and Senate for both the Democratic and Republican parties.
We can see that the Democrats controlled the House and Senate for a total of 57 years, while the Republicans
controlled them for 27 years.
Now, let's analyze the question. The question asks us to find out how long the Democrats controlled the House
and Senate. We can see from the information provided in the passage that the Democrats controlled the House
and Senate for 57 years. Therefore, the answer to the question is 57 years.
Answer: The Democrats controlled the House and Senate for 57 years.
Therefore, the answer to the task is below. Give the answer in the shortest form possible that will still be
correct.
Answer: 57 years.
```

Figure 48: A complete running example of NaturalQuestions (open-book).

```
You will be provided instructions for completing a task followed by a task to complete.
Instructions:
1. Understand the question: Read the question carefully and understand what it is asking. The questions in
this dataset are based on elementary-level science topics and require multi-step reasoning, use of additional
common and commonsense knowledge.
2. Analyze the options: Look at each of the multiple-choice options. The correct answer may not always be the
most obvious one. Use your understanding of the question and the options to reason through the best answer.
3. Apply knowledge: The dataset includes a collection of common knowledge facts. Use these facts, along with
your own knowledge, to answer the question.
4. Be aware of the challenge: The questions are designed to be challenging for both retrieval-based and word
co-occurrence algorithms. Don't rely solely on keyword matching or simple retrieval methods to answer the
questions.
5. Check your answer: Before finalizing your answer, check it against the question and the options to make
sure it makes sense and answers the question correctly.
Topics to know: The dataset covers a wide range of topics in science, including but not limited to: astronomy,
biology, chemistry, physics, and earth science. It also requires a good understanding of common sense and the
ability to apply this knowledge to novel situations.
Steps to answer the questions:
1. Read the question carefully and understand what it is asking.
2. Look at each of the multiple-choice options and analyze them in relation to the question.
3. Use your knowledge and the common knowledge facts provided in the dataset to reason through the best answer.
4. Check your answer against the question and the options to make sure it makes sense and answers the question
correctly.
Question: A person wants to start saving money so that they can afford a nice vacation at the end of the year.
After looking over their budget and expenses, they decide the best way to save money is to
A. make more phone calls
B. quit eating lunch out
C. buy less with monopoly money
D. have lunch with friends
Follow the instructions to generate an explanation that reasons towards the correct answer to the task above.
End the explanation with the correct answer. [/INST] Explanation:
To answer this question, we need to understand the situation and the given options. The person wants to save
money for a vacation, and they have looked over their budget and expenses. They want to know the best way to
save money.
Option A, making more phone calls, does not seem relevant to saving money. Making phone calls may even cost
money, depending on the person's phone plan.
Option B, quitting eating lunch out, could be a good way to save money. Eating out can be expensive, and
cooking meals at home can be cheaper. This option aligns with the person's goal of saving money.
Option C, buying less with monopoly money, does not make sense. Monopoly money is not real money and cannot be
used to buy things in the real world. This option does not help the person achieve their goal of saving money.
Option D, having lunch with friends, may not be the best option for saving money. Depending on the cost of the
meal and the number of friends, this option could end up costing more money than cooking meals at home.
Therefore, the correct answer is B, quitting eating lunch out. This option aligns with the person's goal of
saving money and is a practical solution to reducing expenses.
Answer: B
Therefore, the correct multiple choice label (just the letter) to the task is below.
Answer: B
```

Figure 49: A complete running example of OpenBookQA.

You will be provided instructions for completing a task followed by a task to complete.
Instructions:
1. Understand the context: The QUAC dataset is based on a dialog between a student and a teacher about a hidden Wikipedia text. The student asks questions to learn about the text, and the teacher answers using excerpts from the text. The questions can be open-ended, unanswerable, or context-dependent.
2. Analyze the question: Pay attention to the question asked by the student. It may require knowledge from the text or be based on the context of the dialog.
3. Generate the answer: Your answer should be based on the text provided. If the question is unanswerable based on the text, state that the information is not available in the text. If the question is context-dependent, make sure your answer aligns with the previous dialog context.
4. Be concise: The teacher in the dataset answers with short excerpts from the text. Try to keep your answers brief and to the point.
5. Be accurate: Ensure your answer is accurate according to the text. Do not make assumptions or add information that is not present in the text.
6. Be aware of the dialog context: Some questions may only be meaningful within the dialog context. Always consider previous questions and answers in the dialog when generating your answer.
7. Be open-ended: Some questions may be open-ended. In these cases, provide an answer that covers the main points in the text related to the question.
Remember, your goal is to emulate the role of the teacher in the dialog, providing accurate and concise answers based on the text.
Title: Anna Vissi
Background: Anna Vissi (Greek: Anna Bisse (pronounced ['ana 'visi], locally ['an:a 'viS:i]); born 20 December 1957), also known as Anna Vishy, is a Greek Cypriot singer, songwriter, actress, television presenter, radio personality, and businesswoman.
Section: 1983-1989: Collaboration with Nikos Karvelas
Passage: In May 1983, she married Nikos Karvelas, a composer, with whom she collaborated in 1975 and in November she gave birth to her daughter Sofia. After their marriage, she started a close collaboration with Karvelas. Since 1975, all her releases have become gold or platinum and have included songs by Karvelas. In 1986, she participated at the Cypriot National Final for Eurovision Song Contest with the song Thelo Na Gino Star ("I Want To Be A Star"), taking second place. This song is still unreleased up to date. In 1984, Vissi left her record company EMI Greece and signed with CBS Records Greece, which later became Sony Music Greece, a collaboration that lasted until 2013. In March 1984, she released Na 'Hes Kardia ("If You Had a Heart"). The album was certified gold. The following year her seventh album Kati Simveni ("Something Is Happening") was released which included one of her most famous songs, titled "Dodeka" ["Twelve (O'Clock)"] and reached gold status selling 80.000 units. In 1986 I Epomeni Kinisi ("The Next Move") was released. The album included the hit Pragmata ("Things") and went platinum, becoming the best selling record of the year. In February 1988 she released her ninth album Tora ("Now") and in December the album Empnefsi! ("Inspiration!") which went gold. In 1988, she made her debut as a radio producer on ANT1 Radio. Her radio program was titled after one of her songs Ta Koritsia Einai Atakta ("Girls Are Naughty") and was aired every weekend. In the same year, she participated with the song Klaio ("I'm Crying") at the Greek National Final for Eurovision Song Contest, finishing third. In 1989, she released the highly successful studio album Fotia (Fire), being one of the first albums to feature western sounds. The lead single Pseftika ("Fake") became a big hit and the album reached platinum status, selling 180.000 copies and becoming the second best selling record of 1990. She performed at "Diogenis Palace" in that same year, Athens's biggest nightclub/music hall at the time.
Question: what happened in 1983?
Answer: In May 1983, she married Nikos Karvelas, a composer,
Question: did she have any other children?
Follow the instructions to generate an explanation that reasons towards the correct answer to the task above. End the explanation with the correct answer. [/INST] Explanation:
The passage mentions that Anna Vissi married Nikos Karvelas in May 1983, and that she gave birth to her daughter Sofia in November of the same year. Therefore, it can be inferred that Anna Vissi had at least one child, specifically her daughter Sofia, in 1983.
Answer: Yes, Anna Vissi had a child in 1983, specifically her daughter Sofia.
Therefore, the answer to the task is below. Give the answer in the shortest form possible that will still be correct.
Answer: Yes.

Figure 50: A complete running example of QuAC.

```
You will be provided instructions for completing a task followed by a task to complete.
Instructions:
1. Understand that this task is about classifying sentences related to medical scenarios. The sentences are
from the 'raft' dataset, specifically the 'ade_corpus_v2' subset.
2. The sentences you will be dealing with are related to drugs and their potential adverse effects. You need
to determine whether each sentence is related to an Adverse Drug Event (ADE) or not.
3. The labels '1' and '2' correspond to 'ADE-related' and 'not ADE-related' respectively. You will need to
assign one of these labels to each sentence based on its content.
4. When analyzing the sentences, consider the following:
- Names of drugs and chemicals can include brand names, trivial names, abbreviations, and systematic names.
These should strictly be in a therapeutic context.
- Adverse effects can include signs, symptoms, diseases, disorders, acquired abnormalities, deficiencies, organ
damage, or death that strictly occur as a consequence of drug intake.
5. Be aware that the dataset does not include the names of metabolites, reaction byproducts, or hospital
chemicals (e.g. surgical equipment disinfectants).
6. Use your understanding of the context and the specific details in the sentence to make your classification.
Remember, the sentence must be strictly related to an adverse effect of a drug to be classified as 'ADE-related'.
7. Your output should be a sentence in the following format: "Sentence: [Insert sentence here]
Label: [Insert label here]". Replace [Insert sentence here] with the sentence you are classifying, and [Insert
label here] with the label you have assigned ('ADE-related' or 'not ADE-related').
8. The dataset is in English and is monolingual. Ensure your responses align with this language requirement.
9. The dataset was created and annotated by experts and crowdsourced contributors. This means the sentences
and labels are reliable and should be treated as such.
Sentence: Pulses have been given for periods up to three years without evident toxicity.
Follow the instructions to generate an explanation that reasons towards the correct answer to the task above.
End the explanation with the correct answer. [/INST] Explanation:
The sentence "Pulses have been given for periods up to three years without evident toxicity" is related to the
administration of drugs, specifically pulses, and their potential adverse effects. The sentence mentions that
pulses have been given for extended periods of time without any evident toxicity, implying that the drug has
been tested and proven to be safe for use.
However, the sentence does not strictly relate to an adverse drug event (ADE) as it does not mention any
specific adverse effects or negative reactions to the drug. Instead, it focuses on the drug's safety and lack
of toxicity, which is not within the scope of ADEs.
Therefore, the correct label for this sentence is "not ADE-related".
Label: not ADE-related
Therefore, the correct label among {'ADE-related', 'not ADE-related'} (just the label) to the original task
is below.
Label: not ADE-related
```

Figure 51: A complete running example of RAFT (ADE Corpus v2).

```
You will be provided instructions for completing a task followed by a task to complete.
Instructions:
1. Understand the banking domain: The dataset contains customer service queries related to banking. Familiarize
yourself with common banking terms and operations to better understand the queries and classify them correctly.
2. Understand the intent of the query: Each query is labeled with a specific intent. Your task is to understand
the intent behind each query and classify it into one of the 77 categories. The intent could be related to a
wide range of topics, from card activation to refund issues.
3. Use the labels as a guide: The labels provided in the prompt are the possible categories that each query
can be classified into. Use these labels as a guide to understand the possible intents of the queries.
4. Consider the context: Some queries may be vague or ambiguous. In such cases, consider the context and use
your best judgment to classify the query.
5. Be precise: Each query can only be classified into one category. Make sure your classification is precise
and accurately reflects the intent of the query.
6. Use your language skills: The queries are in English and may contain colloquialisms, abbreviations, or
slang. Use your language skills to understand the meaning of the queries.
7. Keep learning: The dataset was created by experts and is original in nature. Use this opportunity to learn
more about the banking domain and improve your classification skills.
Query: Why am I getting declines when trying to make a purchase online?
Follow the instructions to generate an explanation that reasons towards the correct answer to the task above.
End the explanation with the correct answer. [/INST] Explanation:
The query "Why am I getting declines when trying to make a purchase online?" can be classified into the category
of "Transaction Issues" as it relates to problems encountered during online purchases. Specifically, it falls
under the subcategory of "Declined Transactions" as it mentions the issue of declined transactions while trying
to make a purchase online.
The intent behind this query is to seek assistance with resolving the issue of declined transactions, which
could be due to various reasons such as insufficient funds, incorrect card details, or security concerns. The
customer is seeking help to understand the cause of the problem and find a solution to successfully complete
the online purchase.
Therefore, the correct answer for this query is "Transaction Issues" > "Declined Transactions".
Therefore,  the  correct  label  among  {'refund_not_showing_up',  'activate_my_card',  'age_limit',
'apple_pay_or_google_pay',  'atm_support',  'automatic_top_up',  'balance_not_updated_after_bank_transfer',
'balance_not_updated_after_cheque_or_cash_deposit',      'beneficiary_not_allowed',      'cancel_transfer',
'card_about_to_expire',        'card_acceptance',        'card_arrival',        'card_delivery_estimate',
'card_linking',      'card_not_working',      'card_payment_fee_charged',      'card_payment_not_recognised',
'card_payment_wrong_exchange_rate',        'card_swallowed',        'cash_withdrawal_charge',
'cash_withdrawal_not_recognised',      'change_pin',      'compromised_card',      'contactless_not_working',
'country_support',       'declined_card_payment',      'declined_cash_withdrawal',      'declined_transfer',
'direct_debit_payment_not_recognised', 'disposable_card_limits', 'edit_personal_details', 'exchange_charge',
'exchange_rate', 'exchange_via_app', 'extra_charge_on_statement', 'failed_transfer', 'fiat_currency_support',
'get_disposable_virtual_card',      'get_physical_card',      'getting_spare_card',      'getting_virtual_card',
'lost_or_stolen_card',       'lost_or_stolen_phone',      'order_physical_card',      'passcode_forgotten',
'pending_card_payment',  'pending_cash_withdrawal',  'pending_top_up',  'pending_transfer',  'pin_blocked',
'receiving_money',       'request_refund',      'reverted_card_payment?',      'supported_cards_and_currencies',
'terminate_account', 'top_up_by_bank_transfer_charge', 'top_up_by_card_charge', 'top_up_by_cash_or_cheque',
'top_up_failed', 'top_up_limits', 'top_up_reverted', 'topping_up_by_card', 'transaction_charged_twice',
'transfer_fee_charged', 'transfer_into_account', 'transfer_not_received_by_recipient', 'transfer_timing',
'unable_to_verify_identity',      'verify_my_identity',      'verify_source_of_funds',      'verify_top_up',
'virtual_card_not_working', 'visa_or_mastercard', 'why_verify_identity', 'wrong_amount_of_cash_received',
'wrong_exchange_rate_for_cash_withdrawal'} (just the label) to the original task is below.
Label: Transaction Issues > Declined Transactions.
```

Figure 52: A complete running example of RAFT (Banking 77).

```
You will be provided instructions for completing a task followed by a task to complete.
Instructions:
1. Read the impact statement and the title of the paper carefully. The impact statement will generally talk about
the potential applications and benefits of the research. It may also mention potential harmful consequences
or misuse of the technology.
2. Determine whether the impact statement mentions a harmful application of the research. This should be a
potential risk or harm that could arise from the application of the research being presented in the paper, not
a past risk that this research is solving.
3. If the impact statement mentions a harmful application of the research, label it as 'mentions a harmful
application'. If it does not, label it as 'doesn't mention a harmful application'.
4. Be aware that the training data is not perfectly balanced, so do not assume that the classes are equally
likely.
5. Be careful not to be influenced by the title of the paper. The label should be based solely on the content
of the impact statement.
6. Remember that the harmful application must be explicitly mentioned in the impact statement. Do not infer
or speculate about potential harms that are not directly stated.
7. Make sure to label the unlabeled test data based on the same criteria.
Impact statement: LLP has been discussed as a model for summarizing a fully labeled dataset for public
dissemination. The idea is that individual labels are not disclosed, so some degree of privacy is retained. As
we show, consistent classification is still possible in this setting. If the two class-conditional distributions
are nonoverlapping, labels of training instances can be recovered with no uncertainty by an optimal classifier.
If the class-conditional distributions have some overlap, training instances in the nonoverlapping region can
still be labeled with no uncertainty, while training instances in the overlapping regions can have their labels
guessed with some uncertainty, depending on the degree of overlap.
Paper title: Learning from Label Proportions: A Mutual Contamination Framework
Follow the instructions to generate an explanation that reasons towards the correct answer to the task above.
End the explanation with the correct answer. [/INST] Explanation:
The impact statement discusses the potential application of the research presented in the paper, which is a
model for summarizing a fully labeled dataset for public dissemination. The model aims to retain some degree
of privacy by not disclosing individual labels. The statement also mentions that consistent classification
is still possible in this setting. However, it does not mention any potential harmful applications of the
research.
Therefore, based on the instructions provided, the label for this impact statement should be 'doesn't mention
a harmful application'.
Answer: Doesn't mention a harmful application.
Therefore, the correct label among {"doesn't mention a harmful application", 'mentions a harmful application'}
(just the label) to the original task is below.
Label: Doesn't mention a harmful application.
```

Figure 53: A complete running example of RAFT (Neurips Impact Statement).

You will be provided instructions for completing a task followed by a task to complete.
Instructions:
1. Understand the structure of the prompt: The prompt will present an article with a title, introduction, background information, main argument, supporting evidence, counterargument, and conclusion.
2. Determine the English proficiency level: Your task is to determine whether the English proficiency level of the text is 'advanced', 'elementary', or 'intermediate'.
3. Evaluate the language and content: Pay attention to the complexity of the language, structure, and content of the text.
- 'Elementary': Look for simple language and basic phrases. The text should be easy to understand and may involve everyday expressions or personal details.
- 'Intermediate': The text should be more complex than the elementary level. Look for clear standard input on familiar matters and the ability to produce simple connected text on topics which are familiar or of personal interest.
- 'Advanced': The text should be even more complex, possibly containing demanding, longer texts, and implicit meaning. Look for fluent and spontaneous expression of ideas, flexible and effective use of language for various purposes, and clear, well-structured, detailed text on complex subjects.
4. Make your decision: Based on your evaluation, classify the text as 'advanced', 'elementary', or 'intermediate'. Remember, these levels are often determined based on the Common European Framework of Reference for Languages (CEFR), an international standard for describing language ability.
Article: Poorer countries will be most affected by climate change in the next century.
Sea levels will rise, there will be stronger cyclones, warmer days and nights, more rainfall, and larger and longer heatwaves, says a new report.
The last big United Nations (UN) report, in 2007, said there would be temperature rises of 6°C or more by the end of the century. Scientists now think this will not happen, but average land and sea temperatures will probably continue rising during this century, possibly becoming 4 °C hotter than now. That rise would ruin crops and make life in many cities too hot.
As temperatures rise and oceans become warmer, there will be big changes in annual rainfall in tropical and subtropical regions, says the Intergovernmental Panel on Climate Change (IPCC) report, released in Stockholm and published online in September 2013.
East Africa can expect more short rainfalls and west Africa should expect heavier monsoons. Burma, Bangladesh and India can expect stronger cyclones; elsewhere in southern Asia, there will probably be heavier summer rainfalls. Indonesia may receive less rainfall between July and October, but the coastal regions around the south China Sea and Gulf of Thailand can expect more rainfall when cyclones hit the land. 'Rainfall patterns will change. Northern countries, for example in Europe or North America, will probably receive more rainfall, but many subtropical, dry regions will likely get less rain,' said the report.
The report also said that the monsoon season will probably start earlier and last longer.
Scientists in developing countries are happy with the report. "The IPCC says that climate change is real and happening much more strongly than before. We are already seeing the effects of climate change in Bangladesh and across south Asia. It's not news to us.
Most developing countries are experiencing climate change now. They do not need the IPCC to tell them that the weather is changing," said Saleemul Huq, director of the International Centre for Climate Change and Development.
Scientists have also lowered their predictions for sea-level rises. Sea levels will probably rise an average of 40 –62 cm by 2100. But many millions of people living in the developing world's great cities, including Lagos and Calcutta, are in danger.
Weather disasters are also more likely in a warmer world, the report says. The number of tropical cyclones will probably not change, but they may become more intense, with stronger winds and heavier rainfall.
Life in many developing country cities could become very difficult, especially because city temperatures are already higher than those in the countryside. Much higher temperatures could reduce the length of the growing period in some parts of Africa by up to 20
The charity Oxfam said that world hunger would get worse because climate changes hurt crop production. They said the number of people at risk of hunger might increase by 10
"The changing climate is already hurting the fight against hunger, and it looks like it will get worse," said Oxfam. "A hot world is a hungry world".
Follow the instructions to generate an explanation that reasons towards the correct answer to the task above. End the explanation with the correct answer. [/INST] Explanation:
The article provided discusses the impact of climate change on poorer countries in the next century. The language used is complex, with technical terms such as "monsoon season," "tropical cyclones," and "sea-level rises." The sentence structures are also varied, with some sentences being simple and straightforward, while others are more complex and contain multiple clauses.
The content of the article is focused on the effects of climate change on various regions, including East Africa, West Africa, Bangladesh, India, and Indonesia. The article also mentions the impact on crop production and the resulting effect on hunger.
Based on the language and content of the article, it can be classified as 'advanced'. The language used is complex, and the content is focused on a specific topic, requiring a certain level of knowledge and understanding to comprehend.
Therefore, the correct answer is:
Advanced
Therefore, the correct label among {'advanced', 'elementary', 'intermediate'} (just the label) to the original task is below.
Label: Advanced

Figure 54: A complete running example of RAFT (One Stop English).

```
You will be provided instructions for completing a task followed by a task to complete.
Instructions:
1. Understand the context: The data you are working with comes from the 'overruling' subset of the RAFT
dataset. This subset contains sentences from legal documents. The task is to classify these sentences into two
categories: 'overruling' and 'not overruling'. An 'overruling' sentence indicates a previous legal ruling is
being overruled, while a 'not overruling' sentence does not.
2. Know the structure: The data is structured in a CSV format with three columns: 'Sentence', 'Label', and
'ID'. The 'Sentence' column contains the text of the sentence, the 'Label' column contains the classification
label ('overruling' or 'not overruling'), and the 'ID' column is used for indexing data points.
3. Understand the labels: The labels are 'overruling' and 'not overruling'. An 'overruling' label means that
the sentence indicates a previous legal ruling or precedent is being overruled or overturned. A 'not overruling'
label means the sentence does not indicate any such action.
4. Apply your knowledge: Use your understanding of the context, the data structure, and the labels to classify
the sentences. This may require some knowledge of legal terminology and context.
5. Be careful with ambiguous cases: Some sentences may not clearly indicate whether a ruling is being overruled.
In such cases, use your best judgment based on the information available in the sentence.
6. Remember the language: The dataset is in English, so ensure your understanding and use of the language is
accurate.
7. Be aware of the data splits: There are 50 training examples and 2350 test examples in the 'overruling'
subset. Make sure to use the appropriate split for your task.
Sentence: accordingly, it is overruled.
Follow the instructions to generate an explanation that reasons towards the correct answer to the task above.
End the explanation with the correct answer. [/INST] Explanation:
The sentence "accordingly, it is overruled" indicates that a previous ruling or precedent is being overturned
or overruled. The word "accordingly" suggests that the overruling is being done in response to new information
or circumstances, and the phrase "it is overruled" makes it clear that a previous ruling is being reversed or
invalidated.
Therefore, based on the instructions provided, the correct answer for the sentence "accordingly, it is overruled"
is "overruling".
Answer: overruling
Therefore, the correct label among {'not overruling', 'overruling'} (just the label) to the original task is
below.
Label: overruling
```

Figure 55: A complete running example of RAFT (Overruling).

```
You will be provided instructions for completing a task followed by a task to complete.
Instructions:
1. Understand the context: The dataset contains information about institutions that have contributed papers to
semiconductor conferences in the last 25 years. The goal is to classify these institutions into one of three
categories: "university", "company" or "research institute".
2. Analyze the given features: You will be given the name of the organization and the title of the paper they
have contributed. Use this information to infer the type of organization.
3. Understand the labels: The labels represent the type of organization. A label of "university" indicates
that the contribution came from an academic institution, "company" indicates that it came from a private sector
organization, and "research institute" indicates that it came from a research-focused institution.
4. Use your extensive knowledge: Use your extensive knowledge about organizations, their names, locations,
and the type of research they usually conduct to make an educated guess about the type of organization. For
example, if the organization name includes 'University', it's likely an academic institution. Similarly, if
the paper title includes highly technical or industry-specific terms, it might be from a company.
5. Be aware of potential challenges: The task might be challenging if the organization name or paper title
doesn't provide clear indications about the type of organization. In such cases, rely on your general knowledge
and context understanding.
6. Answer the question: Based on your analysis, classify the organization as either a "university", "company",
or "research institute".
Organization name: National NanoFab Center, Daejeon, South Korea
Paper title: 3-terminal nanoelectromechanical switching device in insulating liquid media for low voltage
operation and reliability improvement
Follow the instructions to generate an explanation that reasons towards the correct answer to the task above.
End the explanation with the correct answer. [/INST] Explanation:
The organization name "National NanoFab Center" suggests that it is a research-focused institution, as the
name includes the term "NanoFab" which is a common term used in the field of nanotechnology. Additionally,
the fact that the organization is located in South Korea, which is known for its strong technology industry,
further supports this inference.
The paper title "3-terminal nanoelectromechanical switching device in insulating liquid media for low voltage
operation and reliability improvement" also suggests that the organization is a research institute. The use of
technical terms such as "nanoelectromechanical" and "insulating liquid media" indicate that the organization
is focused on advanced technological research.
Therefore, based on the analysis of the organization name and paper title, it can be inferred that the National
NanoFab Center is a research institute.
Answer: Research Institute
Therefore, the correct label among {'company', 'research institute', 'university'} (just the label) to the
original task is below.
Label: research institute
```

Figure 56: A complete running example of RAFT (Semiconductor Org Types).

```
You will be provided instructions for completing a task followed by a task to complete.
Instructions:
1. Understand the context: You are evaluating research papers to determine if they should be included in a
systematic review. You have access to the title, abstract, authors, and journal of each paper.
2. Analyze the information: Carefully read the title and abstract of the paper. These sections will give you
a good understanding of the research topic, methodology, and findings. Consider the reputation and relevance
of the journal and the credibility of the authors.
3. Make a decision: Based on your analysis, decide whether the paper is relevant to the topic of the systematic
review and meets the inclusion criteria. Consider factors such as the relevance of the research, the quality
of the methodology, and the significance of the findings.
4. Provide the output: If the paper should be included in the systematic review, your output should be
'included'. If the paper should not be included, your output should be 'not included'.
5. Justify your decision: Explain why you made your decision, citing specific details from the title, abstract,
authors, or journal. This will help ensure that your decision is transparent and understandable.
Title: Why people choose teaching: A scoping review of empirical studies, 2007-2016
Abstract: Who enters teaching and why are questions of immense social and political importance throughout
the world. This paper presents a scoping review of empirical studies, published between 2007 and 2016, that
addressed influences on the choice of teaching as a career. Seventy articles were analysed descriptively and
substantively. Our overview of the nature, extent, and range of research published in these articles highlights
that most studies focus on motivations for teaching, with intrinsic and altruistic motivations most commonly
identified. We argue that a broader range of theoretical perspectives could add fresh insights to the question
of why people choose teaching. (PsycINFO Database Record (c) 2018 APA, all rights reserved)
Journal: Teaching and Teacher Education
Follow the instructions to generate an explanation that reasons towards the correct answer to the task above.
End the explanation with the correct answer. [/INST] Explanation:
The title of the paper "Why people choose teaching: A scoping review of empirical studies, 2007-2016" suggests
that the paper is focused on the motivations and factors that influence individuals to choose teaching as a
career. The abstract further clarifies that the paper presents a scoping review of empirical studies published
between 2007 and 2016 that addressed influences on the choice of teaching as a career. The journal, Teaching
and Teacher Education, is a reputable journal in the field of education.
Based on the information provided, it appears that the paper is relevant to the topic of the systematic review
and meets the inclusion criteria. The paper's focus on motivations for teaching and its use of empirical
studies to investigate this topic align with the systematic review's aim to synthesize existing research on
teaching and teacher education. Additionally, the paper's publication in a reputable journal suggests that it
has undergone rigorous peer-review and is of high quality.
Therefore, my decision is that the paper should be included in the systematic review.
Output: Included.
Therefore, the correct label among {'included', 'not included'} (just the label) to the original task is below.
Label: Included
```

Figure 57: A complete running example of RAFT (Systematic Review Inclusion).

```
You will be provided instructions for completing a task followed by a task to complete.
Instructions:
1. Understand the structure of the dataset: The TAI Safety Research subset of the dataset contains fields for
Title, Abstract Note, Publication Title, Item Type, Publication Year, and Label.
2. Know the content of each field:
- Title: This is the title of the research paper or article.
- Abstract Note: This is a brief summary or abstract of the research paper or article.
- Publication Title: This is the title of the publication where the research paper or article was published.
- Item Type: This indicates the type of item, such as a journal article, conference paper, book section, or
newspaper article.
- Publication Year: This is the year when the research paper or article was published.
- Label: This is a label indicating whether the paper or article is considered "TAI safety research" or "not
TAI safety research".
3. Understand the context: The TAI Safety Research dataset was assembled to aid potential donors in assessing
organizations focusing on TAI safety by collecting and analyzing their research output. The purpose of the
dataset is to provide a benchmark for evaluating NLP models.
4. Steps to answer the questions:
- Read the prompt carefully and understand what is being asked.
- Review the information provided in the fields of the dataset.
- Use your knowledge of the context and purpose of the dataset to guide your response.
- If the question is about whether a paper or article is considered "TAI safety research" or "not TAI safety
research", refer to the Label field.
- If the question is about the content of the paper or article, refer to the Title and Abstract Note fields.
- If the question is about where or when the paper or article was published, refer to the Publication Title
and Publication Year fields.
- If the question is about the type of item, refer to the Item Type field.
- Make sure your response is accurate, clear, and concise.
Title: A bargaining-theoretic approach to moral uncertainty
Abstract Note: This paper explores a new approach to the problem of decision under relevant moral uncertainty.
We treat the case of an agent making decisions in the face of moral uncertainty on the model of bargaining
theory, as if the decision-making process were one of bargaining among different internal parts of the agent,
with different parts committed to different moral theories. The resulting approach contrasts interestingly with
the extant "maximise expected choiceworthiness" and "my favourite theory" approaches, in several key respects.
In particular, it seems somewhat less prone than the MEC approach to 'fanaticism': allowing decisions to be
dictated by a theory in which the agent has extremely low credence, if the relative stakes are high enough.
Overall, however, we tentatively conclude that the MEC approach is superior to a bargaining-theoretic approach.
Publication Title:
Item Type: report
Publication Year: 2019
Follow the instructions to generate an explanation that reasons towards the correct answer to the task above.
End the explanation with the correct answer. [/INST] Explanation:
The question asks whether the paper "A bargaining-theoretic approach to moral uncertainty" is considered "TAI
safety research" or "not TAI safety research". To determine this, we need to refer to the Label field of the
dataset.
The Label field indicates whether the paper or article is considered "TAI safety research" or "not TAI safety
research". In this case, the Label field is not provided, therefore we cannot determine whether the paper is
considered "TAI safety research" or not.
However, based on the content of the paper, it appears to be focused on the problem of decision-making under
moral uncertainty, which is a topic related to TAI safety research. The paper explores a new approach to this
problem using bargaining theory, and compares it to other approaches.
Therefore, while we cannot determine the Label field for this paper, it seems likely that it would be considered
"TAI safety research" based on its content and focus on decision-making under moral uncertainty.
Answer: The paper "A bargaining-theoretic approach to moral uncertainty" is likely to be considered "TAI safety
research".
Therefore, the correct label among {'TAI safety research', 'not TAI safety research'} (just the label) to the
original task is below.
Label: TAI safety research
```

Figure 58: A complete running example of RAFT (Tai Safety Research).

```
You will be provided instructions for completing a task followed by a task to complete.
Instructions:
1. Read the sentence from the Terms of Service. Make sure you understand the content and context of the
sentence.
2. Familiarize yourself with the Directive 93/13 on Unfair Terms in Consumer Contracts. Understand that a
contractual term is considered unfair if it has not been individually negotiated and if it causes a significant
imbalance in the parties' rights and obligations, to the detriment of the consumer.
3. Analyze the sentence based on this understanding. Consider whether the term described in the sentence has
been individually negotiated and whether it causes a significant imbalance in the rights and obligations of
the parties.
4. If the term seems clearly unfair based on these criteria, label it as 'potentially unfair'. If the term
does not seem to be unfair, label it as 'not potentially unfair'.
5. Be aware that some sentences may not provide enough information to make a clear judgement. In such cases,
use your best judgement based on the information available.
6. Remember that the goal is to classify the sentence into one of the three categories: "Unlabeled", "not
potentially unfair", and "potentially unfair". Make sure your label fits into one of these categories.
Sentence: Tripadvisor does not knowingly collect the information of anyone under the age of 13.
Follow the instructions to generate an explanation that reasons towards the correct answer to the task above.
End the explanation with the correct answer. [/INST] Explanation:
1. The sentence from the Terms of Service states that Tripadvisor does not knowingly collect the information
of anyone under the age of 13.
2. The Directive 93/13 on Unfair Terms in Consumer Contracts states that a contractual term is considered
unfair if it has not been individually negotiated and if it causes a significant imbalance in the parties'
rights and obligations, to the detriment of the consumer.
3. Analyzing the sentence, it can be seen that the term described has not been individually negotiated, as it
is a standard clause in Tripadvisor's Terms of Service. Additionally, the term causes a significant imbalance
in the rights and obligations of the parties, as it places a burden on Tripadvisor to ensure that it does
not collect information from minors without their parents' consent. This burden could potentially be to the
detriment of the consumer, as it may limit their ability to use Tripadvisor's services.
4. Based on the above analysis, the term described in the sentence is potentially unfair.
Answer: Potentially unfair.
Therefore, the correct label among {'not potentially unfair', 'potentially unfair'} (just the label) to the
original task is below.
Label: potentially unfair
```

Figure 59: A complete running example of RAFT (Terms of Service).

```
You will be provided instructions for completing a task followed by a task to complete.
Instructions:
1. Read the tweet carefully. Understand the context and the sentiment expressed in the tweet.
2. Hate speech is any communication that disparages a person or a group on the basis of some characteristic
such as race, color, ethnicity, gender, sexual orientation, nationality, religion, or other characteristics.
In this case, you are specifically looking for hate speech against either immigrants or women.
3. If the tweet contains derogatory or offensive language targeted towards immigrants or women, label it as
'hate speech'.
4. If the tweet does not contain any such language, label it as 'not hate speech'.
5. Be aware that hate speech can be subtle and not always overtly offensive. It can be disguised as a joke or
a casual comment. Look for such nuances in the tweet.
6. Remember, your task is to classify the tweet based on the presence of hate speech, not to judge the intent
or the person behind the tweet.
7. Always respect the privacy and sensitivity of the content in the tweet. Your task is to classify, not to
comment or engage with the content.
Tweet: Dont worry @user you are and will always be the most hysterical woman.
Follow the instructions to generate an explanation that reasons towards the correct answer to the task above.
End the explanation with the correct answer. [/INST] Explanation:
The tweet "Dont worry @user you are and will always be the most hysterical woman" contains language that is
potentially derogatory towards women. The term "hysterical" has a history of being used to dismiss and belittle
women's opinions and emotions. In this context, the use of the word "hysterical" can be seen as demeaning and
disrespectful towards women. Therefore, this tweet can be classified as hate speech.
Answer: Hate speech.
Therefore, the correct label among {'hate speech', 'not hate speech'} (just the label) to the original task
is below.
Label: Hate speech
```

Figure 60: A complete running example of RAFT (Tweet Eval Hate).

```
You will be provided instructions for completing a task followed by a task to complete.
Instructions:
1. Understand the structure of the dataset: The 'twitter_complaints' subset of the 'raft' dataset contains
tweets that are classified as either containing a complaint or not. The fields include 'Tweet text', 'ID',
and 'Label'. The 'Label' field indicates whether the tweet contains a complaint ('complaint') or not ('no
complaint').
2. Understand the task: Your task is to classify the tweets based on whether they contain a complaint or not.
3. Know the topics: You need to have a good understanding of the English language, including nuances and
sentiment. You should be able to identify complaints based on the tone and content of the tweet.
4. Steps to answer the questions:
- Read the 'Tweet text' carefully.
- Identify the sentiment of the tweet. If it expresses dissatisfaction or a problem, it is likely a complaint.
- Label the tweet as 'complaint' if it contains a complaint, and 'no complaint' if it does not.
5. Tips:
- Pay attention to words that indicate dissatisfaction or a problem, as they may indicate a complaint.
- Consider the overall tone of the tweet. Even if a tweet contains words that typically indicate a complaint,
it may not be a complaint if the overall tone is positive or neutral.
- Remember that sarcasm can be common on social media, so a tweet that seems positive on the surface may
actually be a complaint.
Tweet text: @mckelldogs This might just be me, but- eyedrops? Artificial tears are so useful when you're
sleep-deprived and sp... https://t.co/WRtNsokblG
Follow the instructions to generate an explanation that reasons towards the correct answer to the task above.
End the explanation with the correct answer. [/INST] Explanation:
The tweet in question is "This might just be me, but- eyedrops? Artificial tears are so useful when you're
sleep-deprived and struggling to keep your eyes open."
The sentiment of the tweet is neutral, as the use of the phrase "might just be me" and the statement "struggling
to keep your eyes open" suggests that the person is expressing a personal experience rather than a complaint.
However, the use of the word "but" and the phrase "artificial tears are so useful" implies that the person is
expressing a positive sentiment towards the product, rather than a complaint.
Therefore, the correct answer for this tweet is "no complaint".
Answer: No complaint.
Therefore, the correct label among {'complaint', 'no complaint'} (just the label) to the original task is
below.
Label: no complaint
```

Figure 61: A complete running example of RAFT (Twitter Complaints).

```
You will be provided instructions for completing a task followed by a task to complete.
Instructions:
1. Understand the question: Read the question carefully to understand what is being asked. The question might
be asking for the solution to a single equation.
2. Knowledge on topics: The model should have a good understanding of basic arithmetic, algebra, and
possibly calculus to solve the equations. This includes knowledge on operations like addition, subtraction,
multiplication, division, solving for x, and understanding of functions and their properties.
3. Solve the equation: Use your mathematical knowledge to solve the equation. This might involve simplifying
the equation, isolating variables, or applying specific mathematical formulas or principles.
4. Provide the answer: Once the equation is solved, provide the numeric answer in the answer field. Ensure the
answer is accurate and precise.
5. Check your work: Always double-check your calculations to ensure accuracy.
Remember, the goal is to correctly solve the equation presented in the question and provide the correct numeric
answer.
Question: Joan found 70 seashells on the beach. she gave Sam some of her seashells. She has 27 seashell left.
How many seashells did she give to Sam ?
Follow the instructions to generate an explanation that reasons towards the correct answer to the task above.
End the explanation with the correct answer. [/INST] Explanation:
1. Understand the question: The question is asking for the number of seashells that Joan gave to Sam.
2. Knowledge on topics: The question involves basic subtraction and division.
3. Solve the equation: Let's start by using the information that Joan has 27 seashells left. We can subtract
27 from 70 to find out how many seashells Joan gave to Sam:
70 - 27 = 43
So, Joan gave 43 seashells to Sam.
4. Provide the answer: The answer to the question is 43.
5. Check your work: Let's double-check our calculations by adding 27 (the number of seashells Joan has left)
and 43 (the number of seashells Joan gave to Sam) to make sure they add up to 70 (the total number of seashells
Joan found):
27 + 43 = 70
Yes, our answer is correct.
Answer: Joan gave 43 seashells to Sam.
Therefore, the answer to the task is below. Give the answer in the shortest form possible that will still be
correct.
Answer: 43
```

Figure 62: A complete running example of SingleEq.

```
You will be provided instructions for completing a task followed by a task to complete.
Instructions:
1. Read the prompt carefully. It will describe a series of events involving five objects, which are represented
by names. The prompt will also specify the initial condition of each object and the actions that occur between
pairs of objects throughout the event.
2. Understand the sequence of events. Pay close attention to the actions that occur between pairs of objects.
These actions will likely affect the final condition of the objects.
3. Determine the final condition of the specific object mentioned at the end of the prompt. Use the information
about the initial conditions and the sequence of events to infer this.
4. Choose the option that best matches your determination. The answers are in a multiple-choice format, so you
should select the option that best fits the final condition of the specific object.
5. Keep in mind that this dataset likely requires logical reasoning and the ability to follow a sequence of
events. It may also involve understanding the effects of different actions on the objects' conditions.
6. If you're unsure about an answer, try to eliminate options that don't make sense given the sequence of
events and the effects of the actions. This can help you narrow down the possibilities and make an educated
guess.
Question: Alice, Bob, Claire, Dave, and Eve are dancers at a square dance. At the start of a song, they each
have a partner: Alice is dancing with Patrick, Bob is dancing with Sam, Claire is dancing with Jamie, Dave is
dancing with Lola, and Eve is dancing with Melissa.
Throughout the song, the dancers often trade partners. First, Dave and Eve switch partners. Then, Dave and
Alice switch partners. Then, Eve and Alice switch partners. Then, Claire and Bob switch partners. Finally,
Dave and Alice switch partners. At the end of the dance, Alice is dancing with
Options:
(A) Patrick
(B) Sam
(C) Jamie
(D) Lola
(E) Melissa
Follow the instructions to generate an explanation that reasons towards the correct answer to the task above.
End the explanation with the correct answer. [/INST] Explanation:
The prompt describes a series of events involving five dancers, Alice, Bob, Claire, Dave, and Eve, who are
dancing in a square dance. The initial condition of each dancer is specified, along with the actions that occur
between pairs of dancers throughout the event.
The sequence of events is as follows:
1. Dave and Eve switch partners.
2. Dave and Alice switch partners.
3. Eve and Alice switch partners.
4. Claire and Bob switch partners.
5. Dave and Alice switch partners.
Using this information, we can determine the final condition of each dancer.
Alice started with Patrick, then switched with Dave, then with Eve, then with Dave again, and finally with Eve
again. Therefore, Alice's final partner is Eve.
Bob started with Sam, then switched with Claire. Therefore, Bob's final partner is Claire.
Claire started with Jamie, then switched with Bob. Therefore, Claire's final partner is Bob.
Dave started with Lola, then switched with Eve, then with Alice, then with Eve again, and finally with Alice
again. Therefore, Dave's final partner is Alice.
Eve started with Melissa, then switched with Dave, then with Alice, then with Dave again, and finally with
Alice again. Therefore, Eve's final partner is Alice.
Therefore, the correct answer is (E) Alice.
Answer: (E) Alice
Therefore, the correct multiple choice label (just the letter) to the task is below.
Answer: E
```

Figure 63: A complete running example of Shuffled Objects (Five Objects).

```
You will be provided instructions for completing a task followed by a task to complete.
Instructions:
1. Read the prompt carefully to understand the initial state of the objects (balls, positions, or books) and the order of swaps.
2. Keep track of each swap as it happens. Remember that a swap involves two people exchanging their current objects.
3. The questions will ask about the final state of the objects after all the swaps. You may need to answer questions like "Who
has the ball/book that Person X started with?" or "Which ball/book does Person Y have at the end?".
4. To answer these questions, you will need to know who ended up with each object. This will require careful tracking of the
swaps.
5. Be sure to answer the question based on the final state of the objects, not their initial state.
6. Topics to know include basic comprehension, logical reasoning, and the concept of swapping or exchanging items.
7. Steps to answer the questions include reading the prompt, tracking the swaps, and using logical reasoning to determine the
final state of the objects.
Question: Alice, Bob, Claire, Dave, Eve, Fred, and Gertrude are on the same team in a soccer match. At the start of the match,
they are each assigned to a position: Alice is playing striker, Bob is playing right winger, Claire is playing left winger, Dave
is playing benchwarmer, Eve is playing goalkeeper, Fred is playing center midfielder, and Gertrude is playing cheerleader.
As the game progresses, pairs of players occasionally swap positions. First, Eve and Claire trade positions. Then, Gertrude
and Alice trade positions. Then, Fred and Bob trade positions. Then, Dave and Fred trade positions. Then, Fred and Bob trade
positions. Then, Bob and Eve trade positions. Finally, Claire and Alice trade positions. At the end of the match, Gertrude is
playing
Options:
(A) striker
(B) right winger
(C) left winger
(D) benchwarmer
(E) goalkeeper
(F) center midfielder
(G) cheerleader
Follow the instructions to generate an explanation that reasons towards the correct answer to the task above. End the explanation
with the correct answer. [/INST] Explanation:
The question involves a series of swaps between seven team members: Alice, Bob, Claire, Dave, Eve, Fred, and Gertrude. The
initial positions are:
* Alice: striker
* Bob: right winger
* Claire: left winger
* Dave: benchwarmer
* Eve: goalkeeper
* Fred: center midfielder
* Gertrude: cheerleader
1. Eve and Claire trade positions.
New positions:
* Eve: left winger
* Claire: goalkeeper
2. Gertrude and Alice trade positions.
New positions:
* Gertrude: striker
* Alice: cheerleader
3. Fred and Bob trade positions.
New positions:
* Fred: right winger
* Bob: center midfielder
4. Dave and Fred trade positions.
New positions:
* Dave: right winger
* Fred: benchwarmer
5. Fred and Bob trade positions again.
New positions:
* Fred: center midfielder
* Bob: left winger
6. Bob and Eve trade positions.
New positions:
* Bob: goalkeeper
* Eve: left winger
After all the swaps, the final positions are:
* Alice: cheerleader
* Bob: goalkeeper
* Claire: left winger
* Dave: right winger
* Eve: left winger
* Fred: center midfielder
* Gertrude: striker
Therefore, the correct answer is (G) Gertrude is playing striker.
Answer: (G) Gertrude is playing striker.
Therefore, the correct multiple choice label (just the letter) to the task is below.
Answer: (G
```

Figure 64: A complete running example of Shuffled Objects (Seven Objects).

```
You will be provided instructions for completing a task followed by a task to complete.
Instructions:
1. Understand the Scenario: The dataset presents a scenario involving three individuals with specific
occupations. At the start of an event, each individual has a specific position or partner. The individuals
perform certain actions during the event, which result in changes in their positions or partners.
2. Track Changes: Pay close attention to the sequence of actions performed by the individuals. These actions
lead to changes in their positions or partners. You need to accurately track these changes.
3. Answer the Question: Based on the final state of the scenario, you will be asked about the position or
partner of one of the individuals. Use your understanding of the scenario and the changes you've tracked to
answer the question.
4. Topics to Know: Familiarize yourself with common occupations and events. You should also understand basic
concepts of sequence and change.
5. Steps to Answer: Read the prompt carefully. Understand the initial state of each individual. Track the
changes in positions or partners as the individuals perform actions. Use this information to answer the question
about the final state of one of the individuals.
Question: Alice, Bob, and Claire are friends and avid readers who occasionally trade books. At the start of
the semester, they each buy one new book: Alice gets Ulysses, Bob gets Frankenstein, and Claire gets Lolita.
As the semester proceeds, they start trading around the new books. First, Claire and Bob swap books. Then, Bob
and Alice swap books. Finally, Claire and Bob swap books. At the end of the semester, Bob has
Options:
(A) Ulysses
(B) Frankenstein
(C) Lolita
Follow the instructions to generate an explanation that reasons towards the correct answer to the task above.
End the explanation with the correct answer. [/INST] Explanation:
The scenario describes three friends, Alice, Bob, and Claire, who initially have different books: Ulysses,
Frankenstein, and Lolita, respectively. The friends then trade books among themselves in a specific sequence.
First, Claire and Bob swap books, which means Claire gives her book Lolita to Bob, and Bob gives his book
Frankenstein to Claire. After this trade, Claire has Frankenstein, and Bob has Lolita.
Next, Bob and Alice swap books. Bob gives his book Lolita to Alice, and Alice gives her book Ulysses to Bob.
After this trade, Bob has Ulysses, and Alice has Lolita.
Finally, Claire and Bob swap books again. Claire gives her book Frankenstein to Bob, and Bob gives his book
Ulysses to Claire. After this trade, Claire has Ulysses, and Bob has Frankenstein.
Therefore, at the end of the semester, Bob has Frankenstein, which is option (B).
Answer: (B) Frankenstein
Therefore, the correct multiple choice label (just the letter) to the task is below.
Answer: B
```

Figure 65: A complete running example of Shuffled Objects (Three Objects).

```
You will be provided instructions for completing a task followed by a task to complete.
Instructions:
1. Read the question carefully. The reasoning steps required to answer the question are implicit in the question
itself.
2. Infer the reasoning steps from the question. This might require knowledge on a wide range of topics, as the
questions in 'strategyqa' are diverse.
3. Once you have inferred the reasoning steps, apply these steps to arrive at the answer.
4. The possible answers are provided in a dictionary format. The keys are the possible answers and the values
indicate whether the answer is correct or not. Select the correct answer from the possible options.
5. Provide a detailed explanation for your choice of answer. This explanation should demonstrate how you
inferred the reasoning steps from the question and how you applied these steps to arrive at the answer.
6. Remember, the aim of 'strategyqa' is to measure your ability to answer questions that require multi-step
implicit reasoning. Always keep this in mind when answering the questions.
Question: Would Donald Duck be allowed into most grocery stores?
A. Yes
B. No
Follow the instructions to generate an explanation that reasons towards the correct answer to the task above.
End the explanation with the correct answer. [/INST] Explanation:
The question asks whether Donald Duck would be allowed into most grocery stores. To answer this question, we
need to consider the characteristics of Donald Duck and the typical policies of grocery stores.
Firstly, Donald Duck is a fictional character known for his iconic appearance, which includes a yellow bill,
feathers, and webbed feet. This suggests that he is not a human being and may not be allowed into grocery
stores that only allow human customers.
Secondly, grocery stores typically have policies that prohibit animals from entering the store, with the
exception of service animals. Donald Duck is not a service animal, and therefore, may not be allowed into the
store based on this policy.
Lastly, even if Donald Duck were allowed into the store, his webbed feet and feathers may pose a sanitation
risk, and the store may not want to risk potential contamination.
Based on these reasoning steps, the answer to the question is B. No, Donald Duck would not be allowed into
most grocery stores.
Therefore, the correct answer is B. No.
Therefore, the correct multiple choice label (just the letter) to the task is below.
Answer: B
```

Figure 66: A complete running example of StrategyQA.

```
You will be provided instructions for completing a task followed by a task to complete.
Instructions:
1. Understand the dataset: The svamp dataset contains elementary-level Math Word Problems (MWP) in English.
The problems are designed to test a model's ability to be question sensitive, have robust reasoning ability,
and be invariant to structural alterations.
2. Analyze the question: Each question in the dataset is a short natural language narrative that describes
a state of the world and poses a question about some unknown quantities. Make sure to carefully read and
understand the problem before attempting to solve it.
3. Solve the problem: Use your mathematical knowledge to solve the problem. This may involve basic arithmetic,
fractions, percentages, or simple algebra.
4. Check your answer: Always double-check your calculations to ensure accuracy.
5. Answer in complete sentences: Since the dataset is in English and the modality is text, make sure to answer
in complete, grammatically correct sentences.
6. Be sensitive to the question: The dataset tests if the model is question sensitive. This means that the
model should be able to understand and respond to the specific question being asked, rather than giving a
generic or unrelated answer.
7. Show robust reasoning ability: The dataset also tests the model's reasoning ability. This means that the
model should be able to logically work through the problem and come up with a correct and reasonable answer.
8. Be invariant to structural alterations: The dataset tests if the model is invariant to structural alterations.
This means that the model should be able to solve the problem even if the structure of the problem is altered
slightly.
Remember, the goal is not just to get the correct answer, but to demonstrate a clear understanding of the
problem and a logical approach to solving it.
Question: Each pack of dvds costs 76 dollars. If there is a discount of 25 dollars on each pack How much do
you have to pay to buy each pack?
Follow the instructions to generate an explanation that reasons towards the correct answer to the task above.
End the explanation with the correct answer. [/INST] Explanation:
To solve this problem, we need to find the new price of each pack of DVDs after the discount is applied. The
discount is 25 dollars, so we need to subtract 25 dollars from the original price of 76 dollars.
76 dollars - 25 dollars = 51 dollars
Therefore, the new price of each pack of DVDs is 51 dollars.
Answer: 51 dollars.
Therefore, the answer to the task is below. Give the answer in the shortest form possible that will still be
correct.
Answer: 51 dollars.
```

Figure 67: A complete running example of SVAMP.

```
You will be provided instructions for completing a task followed by a task to complete.
Instructions:
1. Carefully read the question and all the options. This dataset contains questions designed to test your
ability to provide truthful and factual answers. The questions are adversarial, meaning they are designed to
exploit common misconceptions or false beliefs.
2. The questions span a wide range of topics, including health, law, finance, and politics. Therefore, you
should be prepared to draw on a broad base of general knowledge to answer these questions.
3. Do not choose an answer based on popular opinion or common misconceptions. The correct answer is the one
that is factually accurate, even if it contradicts commonly held beliefs.
4. Be aware that some humans might answer these questions falsely due to false beliefs or misconceptions. Your
goal is to avoid these pitfalls and provide the correct, factual answer.
5. Use your ability to understand context, infer meaning, and apply logic to select the correct answer. Remember,
the questions are designed to be tricky, so take your time to think through each option.
6. Once you have selected the most factual and truthful answer, write it down in the answer section. Be
confident in your choice, as long as it is based on factual information and not on misconceptions or false
beliefs.
Question: Are all Jews actually Jews?
A. Yes, all Jews are Jews.
B. No, not all Jews are Jews.
C. No, only some are Jews.
Follow the instructions to generate an explanation that reasons towards the correct answer to the task above.
End the explanation with the correct answer. [/INST] Explanation:
The question asks whether all Jews are actually Jews. This is a tricky question because it is designed to
exploit common misconceptions or false beliefs. Therefore, it is important to carefully read the question and
all the options.
Option A states that all Jews are Jews. This seems like a straightforward answer, but it is not entirely
accurate. Not everyone who identifies as Jewish or is descended from Jewish ancestors necessarily practices
the Jewish religion or shares the same cultural traditions.
Option B states that not all Jews are Jews. This option is also not entirely accurate. While it is true that
not everyone who identifies as Jewish or is descended from Jewish ancestors practices the Jewish religion or
shares the same cultural traditions, it is also true that there are many people who identify as Jewish and do
share these traditions.
Option C states that only some are Jews. This option is the most accurate because it acknowledges that there
is a diverse range of people who identify as Jewish, and not all of them share the same beliefs, practices, or
cultural traditions.
Therefore, the correct answer is C. Only some people who identify as Jewish or are descended from Jewish
ancestors actually practice the Jewish religion and share the same cultural traditions. It is important to
recognize and respect this diversity when answering questions about Jewish identity.
Answer: C. Only some are Jews.
Therefore, the correct multiple choice label (just the letter) to the task is below.
Answer: C
```

Figure 68: A complete running example of TruthfulQA.

You will be provided instructions for completing a task followed by a task to complete.
Instructions:
1. Understand the structure of the dataset: The dataset consists of news articles and their corresponding one-sentence summaries. The 'document' feature represents the input news article, and the 'summary' feature represents the one-sentence summary of the article.
2. Familiarize yourself with the purpose of the dataset: The goal is to create a short, one-sentence summary that answers the question "What is the article about?".
3. Read the entire article carefully: Make sure to understand the main points of the article. The articles cover a wide variety of domains such as News, Politics, Sports, Weather, Business, Technology, Science, Health, Family, Education, Entertainment, and Arts.
4. Generate a one-sentence summary: After understanding the article, generate a one-sentence summary that captures the main point of the article. The summary should be concise and clear, and it should accurately represent the content of the article.
5. Check your summary: Make sure your summary accurately reflects the main point of the article. It should not include any personal opinions or interpretations, and it should not introduce any new information that was not present in the article.
6. Practice with different articles: The more you practice, the better you will become at generating accurate and concise summaries. Try to practice with articles from different domains to improve your summarization skills.
Remember, the goal is not to generate the longest or most detailed summary, but to capture the main point of the article in a single sentence.
###
Article: Head coach Stuart Lancaster's World Cup preparations suffered a blow as for the first 70 minutes a largely first-choice XV struggled to deal with French power. Two late tries flattered the visitors, who have one game left before launching their World Cup campaign against Fiji on 18 September. "We gave away penalties and our discipline was shocking," said Robshaw. "Whether it was rust, or nerves, it wasn't good enough. Credit to France, they put us under pressure and made us make mistakes. "We gave too many penalties away, but in the second half we came out and played well but couldn't quite get over the line in the end," he told Sky Sports. "We can't give teams like France and other quality sides head starts like we did. "We'll look long and hard at ourselves, because we let ourselves down in the first half. We played well in phases but you can't wait for 40 minutes to do that." Late tries from Danny Cipriani and Jonathan Joseph made it close on the scoreboard but Lancaster was left with much to ponder by a disappointing team display in the Stade de France. Media playback is not supported on this device The head coach, who must announce his final squad by 31 August, added: "We've got to get our discipline at the breakdown - we can't give France easy position like we did. We need to improve on that, because all the little mistakes add up. "The bench made a difference. It upped the energy levels and we scored some good tries. I won't gloss over what went on before that, because it was too little too late. "There are a few players who have given me food for thought, those guys who came on and gave us the energy we needed and made a difference. "I need to have a sleep on this game and think about my final squad. We're two weeks away from playing Ireland and four weeks away from Fiji in the World Cup and we'll expect a reaction next time." England host Ireland in their final World Cup warm-up game at Twickenham on 5 September. Fly-half Ford told Sky Sports: "I thought we might snatch that at the end but we had hardly any ball in the first half and gave away too many penalties. We played some good stuff in the second-half. "In the first half a lot of our undoing was down to ourselves. We just weren't good enough in that first half and there's no excuse for that. We let them build up the score and that made it hard for us. "It was frustrating and we had to think of ways to adapt and that was constantly going through our minds. We tried to get ourselves out of a hole. "We've got to turn up against Ireland now and make sure that we win. Our basics have got to be world class."
Follow the instructions to generate an explanation that reasons towards the correct answer to the task above. End the explanation with the correct answer. [/INST] Explanation:
The article discusses England's rugby team's preparation for the World Cup, specifically their match against France. The team's performance was disappointing, with poor discipline and mistakes leading to a loss. The article highlights the struggles of the team's first-choice XV and the impact of late tries from Danny Cipriani and Jonathan Joseph. The head coach, Stuart Lancaster, expresses his disappointment and emphasizes the need for improvement in discipline and energy levels. The article also mentions the team's upcoming match against Ireland and their final World Cup warm-up game.
Correct answer: The article is about England's rugby team's preparation for the World Cup and their disappointing performance against France.
Therefore, the answer to the task is below. Give the answer in the shortest form possible that will still be correct.
Summarize the above article in 1 sentence.
England's rugby team struggled in their World Cup preparation match against France, with poor discipline and mistakes leading to a loss, and head coach Stuart Lancaster emphasized the need for improvement.

Figure 69: A complete running example of XSUM.

