# OpenReview forum: "Agent Instructs Large Language Models to be General Zero-Shot Reasoners"
_ICLR.cc/2024/Conference — Submitted to ICLR 2024_

### Official Review · Reviewer_tm2h · 2023-10-28

**Soundness:** 3 good
**Presentation:** 3 good
**Contribution:** 3 good
**Rating:** 6
**Confidence:** 5

**Summary:**

This paper introduces a novel approach where Large Language Models (LLMs) are repurposed as agents during reasoning, enhancing the autonomy of the question-answer process. Specifically, the proposed agent-based reasoner autonomously generates task-specific instructions without the need for training, which subsequently aids the LLM in reasoning over the question. The methodology's efficacy is showcased across various tasks (generation, classification, and reasoning) and on models of differing scales, such as Vicuna-13b, Llama-2-70b-chat, and GPT-3.5 Turbo. Notably, the results are commendable.

**Strengths:**

1. The innovative AgentInstruct autonomously streamlines prompt engineering. It significantly reduces the human effort required in designing chain-of-thought exemplars while maintaining high reasoning prowess.

2. AgentInstruct exhibits versatility, as evidenced by its robust performance across LLMs of various scales (13b, 70b, and ~200b), indicating that it isn't limited to a specific scale.

3. The method's adaptability extends to different tasks, including generation, classification, and reasoning, suggesting it isn't merely task-specific.

4. The community stands to benefit immensely from the authors' decision to release the code, as it paves the way for effortless result reproduction and broader application of the method.

**Weaknesses:**

1. Although the methodology promotes a higher degree of autonomy in the question-answer process, its implementation might be deemed intricate by some, which could be a potential shortcoming (on the contrary, the implementation of simple methods such as CoT is easier).

2. Implementing the method could demand higher computational throughput from the LLM, leading to increased computational costs or API fees.

3. While there's a notable enhancement in the performance of cutting-edge LLMs — Vicuna-13b by 13.3%, Llama-2-70b-chat by 23.2%, and GPT-3.5 Turbo by 17.0% — the performance increment isn't as pronounced with larger models like GPT-3.5 Turbo as it is with Llama-2-70b-chat. This raises concerns about the method's scalability with even larger models. The absence of data on its application to GPT-4, which is larger than GPT-3.5, further intensifies this curiosity about the method's scalability.

**Questions:**

Please see weaknesses. I would like to update my evaluation after the discussion.

**Details Of Ethics Concerns:**

N.A.

---

> ### Author Response · Authors · 2023-11-18
> **Response to Reviewer tm2h**
>
> We feel excited that you like our paper in general! Our response to your questions is below:
>
> **Implementation**: We acknowledge that there are more steps in the zero-shot AgentInstruct pipeline than chain-of-thought. But one of the motivations to develop an agent-based approach is that the recent language agents (ReAct [1], LangChain [2], AutoGPT [3], GPTs [4], AutoGen [5]) are able to automate task planning and process. So, we have made it as easy to use as possible, which is one of our major internal goals. For now, AgentInstruct only requires the input consisting of basic dataset information such as the name of the dataset (Sec. 2). To facilitate the following research (as you have also mentioned), we already released our code and plan on further increasing the simplicity of our method.
>
> **Computational costs**: One goal of zero-shot AgentInstruct is to improve the small models (like Llama 2) to perform competitively with large models (e.g., GPT 3.5). The result is shown in Figure 6. Since we only generate instructions once per dataset and the instructions do not add much to the context window, we believe the slightly increased cost is worthwhile for the increased performance. Notably, we observe more performance improvement with AgentInstruct on smaller models like Vicuna-13b.
>
> **Performance on larger models**: Because zero-shot AgentInstruct uses GPT-4 (within an agent) to generate the instructions, we would expect better performance where the difference between that and the inference model’s size is larger. This is because we’d expect more
> information to be passed to the smaller model that it doesn’t inherently have access to, so the instructions would be more helpful in this situation. To address your review, we run an additional dataset, IMDB, with the following settings: Zero-Shot GPT-4, Zero-Shot CoT, and Zero-Shot AgentInstruct using GPT-4 for both instruction generation and inference, and the results are 87.4, 96.1, and 96.6 respectively. We will add the results to the paper. These experimental results seem to match up with this intuition, suggesting that as the performance gap between the agent model and inference model narrows, so does the impact of zero-shot AgentInstruct. This aligns with the conclusion from knowledge distillation work in LLMs like Alpaca [6], Vicuna [7], Self-Instruct [8].
>
> References:
>
> [1] ReAct: Synergizing Reasoning and Acting in Language Models. ICLR 2023.
>
> [2] LangChain: GitHub 2022.
>
> [3] AutoGPT: GitHub 2023.
>
> [4] Introducing GPTs. OpenAI 2023.
>
> [5] AutoGen: Enabling Next-Gen LLM Applications via Multi-Agent Conversation Framework. CoRR 2023.
>
> [6] Alpaca: A Strong, Replicable Instruction-Following Model. CRFM Blog Post 2023.
>
> [7] Vicuna: An Open-Source Chatbot Impressing GPT-4 with 90%* ChatGPT Quality. LMSYS Org Blog Post 2023.
>
> [8] Self-Instruct: Aligning Language Models with Self-Generated Instructions. ACL 2023.

---

### Official Review · Reviewer_LvbF · 2023-10-28

**Soundness:** 1 poor
**Presentation:** 3 good
**Contribution:** 1 poor
**Rating:** 3
**Confidence:** 5

**Summary:**

The paper introduces AgentInstruct which generalizes zero-shot reasoning abilities of LLMs. Here, the agent generates instructions which optimize the reasoning process of LLMs. The proposed approach is compared with zero-shot and zero-shot-CoT as baselines and reports 17.8%  and 6.5% average improvement across the tasks.

**Strengths:**

The originality of the paper is mainly rooted in its  proposed approach being a cost-effective alternative to zero-shot CoT as the instructions can be generated using a bigger more costly model while the reasoning LLM can be a cheaper alternative. This is a valid argument highlighted by the authors.

**Weaknesses:**

The agents used a much bigger and powerful LLM, GPT-4, as the default agent to generate the instructions. However, the models which were evaluated on the proposed approach were limited to Vicuna, Llama-2-chat, and GPT-3.5 Turbo. As GPT-4 is missing from the list of models, the evaluation against baseline approaches (zero-shot and zero-shot-CoT) is not a fair comparison because they have not utilized the power of GPT-4. This flaw in experiment design, unfortunately, invalidates the reported results.

Similarly, the agent is given access to search engines to get the "top releant web pages containing information about the dataset". This added information provided to the agent (GPT-4 model) is an extra context which the baseline approaches (zero-shot and zero-shot-CoT) don't have access to. Consequently, the comparison between the proposed AgentInstruct and the baseline methods is not a fair comparison.

In other words while the approach decouples the "instruction generation" from "reasoning", the "instruction generation" step is utilizing extra context and more powerful model which reasonably contributes to the higher end-to-end performance of the approach.

In the "Ablation Study" the authors have evaluated the impact of removing each of the components of the Zero-Shot AgentInstruct to assert that they are all effective. At the same time this ablation study further highlights the impact of GPT4 in the overall performance of the method (Table 1).

Furthermore as demonstrated in Figure 6, the ZeroShot GPT-4 only marginally lacks behind the Zero-Shot Agent-Instruct (79.5 vs 88.1 or 0.6% improvement) with lower cost ($1 vs. $2). This is a very important observation which further invalidates the sanity and effectiveness of the proposed method because a much simpler approach (i.e., ZeroShot which is the simplest of approaches) is demonstrating almost similar results as the very complicated Zero-Shot AgentInstruct. In the same section the authors mention that "Though ReAct narrowly outperforms
zero-shot AgentInstruct" where narrow corresponds to 0.8% improvement which is larger than the 0.6% improvement of Zero-Shot AgentInstruct over the Zero-Shot GPT-4.

**Questions:**

In table 1, please elaborate on the first 3 settings: w/o Agent Instructions, w/o Input Examples and w/o Labels. Reading the manuscript and the paragraph that follows Table 1, it is not fully clear what each of these components represent. Particularly the "Labels" have not been discussed in the manuscript.

---

> ### Author Response · Authors · 2023-11-18
> **Response to Reviewer LvbF**
>
> Thank you for reviewing and for the feedback! We find there is a severe misunderstanding of our proposed method and hope our response is able to clarify this.
>
> **Fundamental misunderstanding**: A key component of zero-shot AgentInstruct, compared to other agent-based methods, is that the agent is only leveraged at the task/dataset level. In practice for a dataset like IMDB, this means that the agent generates a single instruction, and the same instruction is then prepended to all 1000 test instances. For example, given an arbitrary movie review with an unknown sentiment, the inference pipelines of zero-shot AgentInstruct and zero-shot CoT are nearly identical, with the only difference that zero-shot AgentInstruct prepends an agent-generated instruction, which is the same between this movie review and the 999 other movies reviews we evaluate on. So in total, we only run the agent 53 times, resulting in 53 agent-generated instructions, as we evaluate on 53 subsets. This was described in the introduction, Sec. 2, and Appendix A.3.1. And this is the major motivation behind decoupling the instruction generation and reasoning steps.
>
> Zero-shot AgentInstruct is a method. A key part of the method is using a powerful model with external capabilities embedded in the agent to instruct the reasoning process of LLMs on a dataset level. We compare our method to other existing methods (zero-shot and zero-shot CoT), and the results suggest that 1) intermediate reasoning is generally helpful, and 2) intermediate reasoning is most helpful when instructed/grounded by providing a set of well-defined, step-by-step instructions to follow. Therefore we don’t think our method given access to GPT-4 nor the web invalidates the reported results. In contrast, it is a new way to automatically generate prompts according to a task compared to manual prompt engineering.
>
> Another angle to understand zero-shot AgentInstruct is through the lens of knowledge distillation, where a larger, more powerful model (GPT-4) distills task-specific (but again not instance-specific) knowledge to a smaller, less powerful model (Vicuna, Llama-2-chat, GPT-3.5 Turbo). For example, Alpaca [1], Vicuna [2], Self-Instruct [3] are using the data generated from large models like GPT-3.5 to train small models like LLaMA to minimize the performance gap between the large and small models. Along this line, the advantage of our method is that we do not need task-specific training. When compared to these literatures (there is often a single model used for evaluation), our results have been evidenced by three models of different sizes.
>
> **Figure 6 misunderstanding**: With respect to Figure 6, the performance difference between ZeroShot GPT-4 and Zero-Shot Agent-Instruct is not marginal (79.5 vs 88.1 is an 8.6% improvement, **NOT** a 0.6% improvement). Granted, our proposed zero-shot AgentInstruct methodology introduces additional complexity, but does so at the task/dataset level (instead of data instance level) which we believe is acceptable given the overall performance increases (17.8% and 6.5% relative to zero-shot and zero-shot CoT respectively). It’s true that other agent-based methods such as ReAct can narrowly outperform zero-shot AgentInstruct (0.8%), but again zero-shot AgentInstruct is unique in the sense that the agent is only leveraged once for each dataset, instead of on each instance (for example, there are 395 test instances in the AddSub dataset), which explains the monumental cost difference (100 times less). This underperformance is quite narrow compared to the performance difference between Zero-Shot GPT-4 and Zero-Shot AgentInstruct, which is 8.6% and not 0.6%.
>
> **GPT-4 experiment**: As far as not including full evaluation results on GPT-4, the limiting factor was cost. Running all 29 datasets of 53 subsets on 3 modes (zero-shot, zero-shot CoT, and zero-shot AgentInstruct) would use an estimated 193,874,856 input tokens and 70,424,217 output tokens for a total cost of around $10,041 with current GPT-4 prices. Instead, we limited our experimental results on GPT-4 to a limited study on only a couple of datasets. AddSub results are shown in Figure 6, and results on IMDB are as follows: Zero-Shot GPT-4 87.4, Zero-Shot CoT GPT-4 96.1, Zero-Shot AgentInstruct (GPT-4 as agent, GPT-3.5 Turbo for inference) 93.9, and Zero-Shot AgentInstruct (GPT-4 as agent, GPT-4 for inference) 96.6. Even though the sample is relatively small, it seems zero-shot AgentInstruct applies to GPT-4 as well.
>
> **Ablation study**: Regarding the ablation study, intuitively a larger, more powerful model as the agent will lead to better downstream performance. The last setting confirms this, where using GPT-3.5 Turbo to generate the instructions leads to worse performance when compared to GPT-4, but still significantly outperforms the standard zero-shot and zero-shot CoT (Table 1 vs Table 5 and Table 6).

---

> ### Author Response · Authors · 2023-11-18
> **Response to Reviewer LvbF (Cont.)**
>
> **Table 1 (description of settings)**:
>
> w/o Agent Instructions: We compare the zero-shot AgentInstruct methodology to zero-shot CoT. This setting is exactly zero-shot CoT on Llama-2-70b-chat.
>
> w/o Input Examples: We remove the examples from the input to the agent. Typically, the agent is given 5 input-only examples to reference when generating the instructions. These examples do not include the ground truth answers. In this setting, we have the agent generate instructions without providing example inputs to the agent.
>
> w/o Labels: We remove the description of the labels from the input to the agent. Specifically, this is applicable for classification tasks such as IMDB, for example, where the expected outputs are either ‘True’ or ‘False’. For multiple-choice and open-ended generation tasks, we provide a simple description of ‘multiple choice’ or ‘generation’ instead of providing a list of possible outputs. With this setting, we have the agent generate instructions without providing any information or description of the output labels for any task type. An example is given in Figure 1 and Figure 2. We will add more details in the new version.
>
> References:
>
> [1] Alpaca: A Strong, Replicable Instruction-Following Model. CRFM Blog Post 2023.
>
> [2] Vicuna: An Open-Source Chatbot Impressing GPT-4 with 90%* ChatGPT Quality. LMSYS Org Blog Post 2023.
>
> [3] Self-Instruct: Aligning Language Models with Self-Generated Instructions. ACL 2023.

---

> > ### Comment · Reviewer_LvbF · 2023-11-21
> >
> > Thank you authors for your detailed response.
> >
> > In light of your futher explanation on Figure 6, I realize that I must have misread the results and that the reported improvement is considerable. Hence, I take back the last paragraph of Weaknesses section starting with "Furthermore as demonstrated in Figure 6..." and upgrade my vote.
> >
> > We had rightfully highlighted the main contribution of the work on proposing a high-performing low cost solution and we continue to highlight this strength of the paper. However, the impact of the GPT4 agent in instruction generation and the extra context provided for it to generate those instructions (e.g. access to web) makes the comparisons. None of the baselines (Zero-shot, Zero-shot CoT, etc) have access to the extra context.
> >
> > Thank you for further elaborating on the contents of table 1. Regardless of the outcomes for NeurIPS, please consider including those added instructions for your readers particularly for row 3, the w/o Labels.

---

> > > ### Author Response · Authors · 2023-11-22
> > > **Response to Reviewer LvbF**
> > >
> > > Thank you for your continued engagement. Unfortunately, it seems there is still a misunderstanding regarding the contribution of our method. We hope our below response can help clarify things.
> > >
> > > **Contribution misunderstanding**: We want to continue to emphasize that AgentInstruct is essentially the combination of instructions crafted by an agent with zero-shot CoT reasoning. **Similarly, zero-shot CoT [1] is an extension to standard zero-shot prompting: it is the combination of manual instructions with standard zero-shot prompting. Zero-shot AgentInstruct is an extension to zero-shot CoT, by further replacing the manual instructions in zero-shot CoT with the agent generated instructions, as described in Sec. 2.** These instructions are generated automatically (instead of manual prompt engineering) and at a low cost (once per dataset instead of per dataset instance) and follow a trend in the research community to use powerful models to make smaller models more powerful (e.g., Alpaca [2], Vicuna [3], or Self-Instruct [4]). Therefore, our approach should rightfully be compared to zero-shot and zero-shot CoT. In addition, few-shot performance or finetuning can be compared to zero-shot, where even additional task examples/training are used. Therefore, adding extra information should not necessarily impose extra constraints on the evaluation of the models, or else this would invalidate almost every recent advance in LLMs. For example, when comparing Alpaca [2], Vicuna [3], or Self-Instruct [4] based models with a separate model, researchers do not further finetune the separate model on the instruction-tuning data used in Alpaca or Vicuna. When comparing large models to small models, let’s say comparing GPT-3 [5] and GPT-2 [6], researchers do not add billions of parameters to GPT-2, as the addition of the parameters themselves is largely the reason performance is boosted.
> > >
> > > **Table 1 description**: Following your suggestion, we have added a further description of the contents of Table 1 in the appendix.
> > >
> > > References:
> > >
> > > [1] Large Language Models are Zero-Shot Reasoners. NeurIPS 2022.
> > >
> > > [2] Alpaca: A Strong, Replicable Instruction-Following Model. CRFM Blog Post 2023.
> > >
> > > [3] Vicuna: An Open-Source Chatbot Impressing GPT-4 with 90%* ChatGPT Quality. LMSYS Org Blog Post 2023.
> > >
> > > [4] Self-Instruct: Aligning Language Models with Self-Generated Instructions. ACL 2023.
> > >
> > > [5] Language Models are Few-Shot Learners. NeurIPS 2020.
> > >
> > > [6] Language Models are Unsupervised Multitask Learners. OpenAI 2019.

---

### Official Review · Reviewer_JX3F · 2023-11-02

**Soundness:** 3 good
**Presentation:** 3 good
**Contribution:** 2 fair
**Rating:** 5
**Confidence:** 4

**Summary:**

The paper presents a method to improve the zero-shot performance of LLMs on language tasks. The proposed approach involves automatically generating task-specific instructions (using an LLM based agent) that are provided to the reasoning LLM to improve its performance. The task-specific instructions might include steps to break down the task into simpler steps, information about the format of the expected answer. To generate these task-specific instructions, the agent takes as input information about the dataset (sourced from the web) and a few examples from that task. Using this extra context, the authors show that LLMs can reason better about the task and show improvement across several generation and classification based benchmarks. On average they show 6.5% improvement over a chain-of-thought baseline, and 17.5% average improvement over state-of-the-art LLMs like Vicuna, LLaMa2, and GPT-3.5-Turbo.

**Strengths:**

- The paper demonstrates the effectiveness of their approach on a fairly exhaustive evaluation. The approach is evaluated on 29 datasets spanning generation, classification and reasoning. The evaluation is run using three LLM baselines.
- The authors provide exhaustive details about the prompts, evaluation dataset and experiments in the supplementary for easily replicating their experimental setup. The authors also provide code to run the experiments done in the paper.
- The authors also attempt to do error analysis to better understand failure modes of their approach. I specially enjoyed reading Section 3.6 and Appendix C.1

**Weaknesses:**

- Evaluation vs “In-the-wild” generaliztion: While the paper shows strong improvements across various benchmarks, I am not convinced of the fundamental assumption made in the paper. To improve a performance on the task, the authors assume that the task is one of the “standard” tasks LLMs are typically evaluated on. While this is great for evaluating, its unclear how the proposed approach will work “in-the-wild”. In realistic usecases, the task might be completely different than one of the standard usecases. Additionally, we might not know beforehand which benchmark does the task belong to? In that case, it’s not going to always be possible to generate task-specific instructions.
- Second, while the authors claim that the performance of the method is zero-shot, I believe that showing a few labeled examples in the prompt to the agent responsible for generating task-specific instruction is a bit unfair. A fair approach would be to make these examples be available even to other zero-shot baselines. Interestingly, without Input Examples, the approach takes a severe hit (see Table 1, Row 1 vs Row 3).
- Finally, I also believe that cost comparisons done in Figure 6 are a bit misleading. Here, the authors show that Zero-shot AgentInstruct cost ~2$ vs 20$ (ZS-COT GPT4). This is true when running evaluation on standard benchmarks because we have to generate task-specific instructions once per benchmark. But like I said before, this is not going to be true in-the-wild. In that setup, a new instruction will have to be generated for every unique user query. For completeness, and to be consistent with a realistic use-case, it would be more accurate to include the costs of creating task-specific instruction to each instance of the task, and then show the overall cost.

**Questions:**

Related to points raised in the weaknesses section:

- How will the approach work in-the-wild, when the source of task is not known apriori. That is we don’t know which dataset does the task belong to, or if the task belongs to any dataset at all.
- What is the cost of creating the task-specific instruction. Can you add a column to Figure 6, which adds this cost multiplied by number of instances in the dataset plus the additional cost of reasoning using GPT-3.5 Turbo?
- I also didn’t understand the point of experiments in Section 3.3? What is the insight from that experiment?
- I also didn’t fully understand how the agent uses the question-answering API (Section A.3.1) to add context about the task? I think it’s one of the most crucial steps of the pipeline, and it’d be great to explain how it’s implemented in more detail (and perhaps in the main manuscript). Concretely, how are the retrieved documents to generate the instruction? Are the retrieved documents added as prompt to the agent to generate task-specific instruction? Is that in addition to the name of the dataset, task information and few input examples? Why are they added to a vector database?

---

> ### Author Response · Authors · 2023-11-18
> **Response to Reviewer JX3F**
>
> We are happy that you enjoy reading our paper!
>
> **In-the-wild generalization**: Thanks for pointing this out. In fact, in-the-wild tasks are among 29 tasks (53 sub-datasets) that our approach was evaluated on. For example, CivilComments and GSM8K are in-the-wild benchmark datasets as evidenced by HELM [1] and WILDS [2]. Zero-shot AgentInstruct outperforms standard zero-shot and CoT on these datasets. As shown in Table 6, when a task is completely new or out of domain (e.g., AQuA and SingleEq), the performance basically degrades to zero-shot CoT given there is no high-quality task knowledge on the web (Table 9). Of course, we agree that there are directions to enhance our method’s generalization ability. First, zero-shot AgentInstruct would be also suitable for in-the-wild tasks, if, for example, a set of many instances were seen on-the-fly that were drawn from a particular task. Second, given minimal task information, zero-shot AgentInstruct can be expanded to identify which task an instance belongs to, then choose from its previously generated instructions which instruction to use similar to [3]. Third, we can expand our method to use less information if it is not available, e.g., the labels. The requirement for the dataset name could be removed and replaced with more advanced prompting and querying methods. For example, we could imagine solving AddSub; instead of searching for information for “AddSub” explicitly, we could query for help on solving basic arithmetic problems using addition and subtraction and incorporate that general information into solving a specific task. This approach will allow for the utilization of our method and great expansion to more datasets.
>
> **Zero-shot setting**: As shown in the introduction and Sec. 2, the input examples are input-only without ground truth labels. An example is shown in Figure 1 and Figure 2. So, AgentInstruct is zero-shot. Based on our experiments (in Table 1 as you mentioned), we found these were necessary to ensure the instructions were applicable for broad tasks over the instances. We will further clarify this point in the new version.
>
> **Cost of instructions**: We will add the suggested cost to Figure 6. So if the agent is called to generate instructions for each data instance instead of for the AddSub dataset once, the total cost will be approximately 50 cents per instance for 395 instances, costing about $197 overall. Currently, our methodology is the necessity of a repeated application of LLM reasoning steps on instances that fall under the same task. This is because instructions are only generated once per dataset, so the cost of using a large model is small, at most 1 dollar per dataset. When the number of tasks equals the number of instances, the costs would be similar. Still, as the number of instances increases, there will be more savings from using a smaller model for reasoning with zero-shot AgentInstruct regardless of the number of tasks.
>
> **Section 3.3 clarification**: Section 3.3 details the drop in performance when context length is expanded, which is largely based on the lengths of instances in the dataset. This is relevant for users who may be choosing which reasoning model to use; since the instructions take up space in the context window, they may decrease performance. Recent LLMs with longer context support will help further improve the performance.
>
> **Question answering API**: We will move some technical details (currently in Appendix A.3.1) to the main body based on your suggestion. For the question answering API, it is used as a tool for the agent. Since the documents are long, they are added to a vector database so they do not expand past the context length of the language model powering the agent. During the agent instruction generation step, another instance of the powerful language model receives questions about the document and interfaces with the vector database to provide an answer. Further information can be seen in our code.
>
> References:
>
> [1] Holistic Evaluation of Language Models. TMLR 2023
>
> [2] Wilds: A benchmark of in-the-wild distribution shifts. ICML 2021
>
> [3] Large Language Models as Tool Makers. CoRR 2023

---

### Author Response · Authors · 2023-11-22
**An Updated Version has been Uploaded**

We express our sincere appreciation to the reviewers for your insightful feedback on our paper. The opportunity to receive such valuable input has allowed us to thoroughly enhance our work. We have given serious consideration to all the reviews received and have made appropriate revisions to our paper as a result. A summary of changes can be found at the beginning of the Appendix. We are committed to further improving the quality and impact of our research.

---

### Meta-Review · Area_Chair_Zdav · 2023-12-05

**Metareview:**

This collection of reviews for the paper presents a range of perspectives on the proposed method's effectiveness and limitations. While there is consensus on the potential of the approach in enhancing zero-shot reasoning capabilities of large language models (LLMs), concerns are raised about its real-world applicability, fairness in comparisons with baseline models, and cost-effectiveness. Reviewer JX3F appreciates the extensive evaluation across various datasets but questions the assumption of standard tasks and the cost analysis in real-world scenarios. Reviewer LvbF criticizes the use of a more powerful GPT-4 model for instruction generation, which is not so new, and the added context from search engines not available to baseline methods. Reviewer tm2h highlights the innovative aspect of AgentInstruct in streamlining prompt engineering and its versatility across different tasks and LLM scales. However, they note potential implementation complexities and increased computational demands. The varied feedback indicates that while the method shows promise in improving LLM performance, its practicality and fair assessment against other models need further scrutiny.

**Justification For Why Not Higher Score:**

A significant part of the paper is not so new.

**Justification For Why Not Lower Score:**

NA

---

### Decision · Program_Chairs · 2024-01-16

Reject